# Left-right side-specific endocrine signaling complements neural pathways to mediate acute asymmetric effects of brain injury

Nikolay Lukoyanov[1†], Hiroyuki Watanabe[2†], Liliana S Carvalho[1†], Olga Kononenko[2†], Daniil Sarkisyan[2†], Mengliang Zhang[3,4], Marlene Storm Andersen[4], Elena A Lukoyanova[1], Vladimir Galatenko[5‡], Alex Tonevitsky[6,7], Igor Bazov[2], Tatiana Iakovleva[2], Jens Schouenborg[3], Georgy Bakalkin[2]*

[1]Departamento de Biomedicina da Faculdade de Medicina da Universidade do Porto, Instituto de Investigação e Inovação em Saúde, Instituto de Biologia Molecular e Celular, Porto, Portugal; [2]Department of Pharmaceutical Biosciences, Uppsala University, Uppsala, Sweden; [3]Neuronano Research Center, Department of Experimental Medical Science, Lund University, Lund, Sweden; [4]Department of Molecular Medicine, University of Southern Denmark, Odense, Denmark; [5]Faculty of Mechanics and Mathematics, Lomonosov Moscow State University, Moscow, Russian Federation; [6]Faculty of Biology and Biotechnology, National Research University Higher School of Economics, Moscow, Russian Federation; [7]Shemyakin–Ovchinnikov Institute of Bioorganic Chemistry RAS, Moscow, Russian Federation

*For correspondence: georgy.bakalkin@farmbio.uu.se

[†]These authors contributed equally to this work

Present address: [‡]Evotec International GmbH, Göttingen, Germany

**Abstract** Brain injuries can interrupt descending neural pathways that convey motor commands from the cortex to spinal motoneurons. Here, we demonstrate that a unilateral injury of the hindlimb sensorimotor cortex of rats with completely transected thoracic spinal cord produces hindlimb postural asymmetry with contralateral flexion and asymmetric hindlimb withdrawal reflexes within 3 hr, as well as asymmetry in gene expression patterns in the lumbar spinal cord. The injury-induced postural effects were abolished by hypophysectomy and were mimicked by transfusion of serum from animals with brain injury. Administration of the pituitary neurohormones β-endorphin or Arg-vasopressin-induced side-specific hindlimb responses in naive animals, while antagonists of the opioid and vasopressin receptors blocked hindlimb postural asymmetry in rats with brain injury. Thus, in addition to the well-established involvement of motor pathways descending from the brain to spinal circuits, the side-specific humoral signaling may also add to postural and reflex asymmetries seen after brain injury.

## Introduction

Brain lesions can interrupt descending neural pathways that convey motor commands from the cerebral cortex to motoneurons located in the brain stem and anterior horn of the spinal grey matter (*Cai et al., 2019*; *Kuypers, 1981*; *Lemon, 2008*; *Purves et al., 2001*; *Smith et al., 2017*; *Tan et al., 2012*; *Zörner et al., 2014*). Brain injury-induced motor deficits typically develop on the contralateral side of the body and include motor weakness, loss of voluntary movements, spasticity, asymmetric limb reflexes, and abnormal posture. Evidence suggests that these deficits are related to the impaired signaling through the descending motor tracts and to deafferentation-induced spinal neuroplasticity (*Cai et al., 2019*; *Deliagina et al., 2014*; *Kanagal and Muir, 2009*; *Küchler et al., 2002*; *Li and Francisco, 2015*; *Morris and Whishaw, 2016*; *Tan et al., 2012*; *Whishaw et al., 1998*;

**eLife digest** Brain trauma or a stroke often lead to severe problems in posture and movement. These injuries frequently occur only on one side, causing asymmetrical motor changes: damage to the left brain hemisphere triggers abnormal contractions of the right limbs, and vice-versa.

The injuries can disrupt neural tracts between the brain and the spinal cord, the structure that conveys electric messages to muscles. However, research has also shed light on new actors: the hormones released into the bloodstream by the pituitary gland. Similar to the effects of brain lesions, several of these molecules cause asymmetric posture in healthy rats. In fact, a group of hormones can trigger muscle contraction of the left back leg, and another of the right one. Could pituitary hormones mediate the asymmetric effects of brain injuries?

To investigate this question, Lukoyanov, Watanabe, Carvalho, Kononenko, Sarkisyan et al. focused on rats in which the connection between the brain and the spinal cord segments that control the hindlimbs had been surgically removed. This stopped transmission of electric messages from the brain to muscles in the back legs.

Strikingly, lesions on one side of the brain in these animals still led to asymmetric posture, with contraction of the leg on the opposite side of the body. These effects were abolished when the pituitary gland was excised. Postural asymmetry also emerged when blood serum from injured rats was injected into healthy animals. The findings suggest that hormones play an essential role in signalling from the brain to the spinal cord.

Further experiments identified that two pituitary hormones, β-endorphin and Arg-vasopressin, induced contraction of the right but not the left hindlimb of healthy animals. In addition, small molecules that inhibit these hormones could block the deficits seen on the right side after an injury on the left hemisphere of the brain. Taken together, these results show that neurons in the spinal cord are not just controlled by the neural tracts that descend from the brain, but also by hormones which have left-right side-specific actions. This unique signalling could be a part of a previously unknown hormonal mechanism that selectively targets either the left or the right side of the body. This knowledge could help to design side-specific treatments for stroke and brain trauma.

*Zelenin et al., 2016*; *Zörner et al., 2014*; *Zörner et al., 2010*). Mechanisms of these impairments are not well understood.

In animal experiments, a unilateral brain injury (UBI) induces hindlimb postural asymmetry (HL-PA) with contralesional limb flexion and asymmetry of the hindlimb withdrawal reflexes (*Watanabe et al., 2021*; *Watanabe et al., 2020*; *Zhang et al., 2020*). Consistently, lateral hemisection of the spinal cord impairs postural functions (*Zelenin et al., 2016*) and enhances monosynaptic and polysynaptic hindlimb reflexes on the ipsilesional side (*Hultborn and Malmsten, 1983a*; *Hultborn and Malmsten, 1983b*; *Malmsten, 1983*; *Rossignol and Frigon, 2011*). Asymmetry in posture and reflexes persists after complete transection of the spinal cord (*Rossignol and Frigon, 2011*; *Watanabe et al., 2020*; *Zhang et al., 2020*). This may be due to neuroplastic changes in the lumbar spinal cord induced by brain or spinal cord injury through the descending neural tracts. In addition, the side-specific signals from a lesion site to the lumbar domains may be conveyed by endocrine messengers. Signaling that is not mediated by the descending neural tracts has been proposed but generally has been disregarded (*Bakalkin et al., 1986*; *Cope et al., 1980*; *Wolpaw and Lee, 1989*).

Analysis of neurotransmitter mechanisms demonstrates that opioid peptides and Arg-vasopressin may induce the formation of HL-PA in rats with intact brains (*Bakalkin et al., 1981*; *Bakalkin et al., 1986*; *Chazov et al., 1981*; *Klement'ev et al., 1986*; *Watanabe et al., 2020*). Either the left or the right hindlimb can be flexed, depending on the compound injected. The κ-opioid agonists dynorphin, bremazocine and U-50,488, as well as the preferential endogenous μ-opioid agonist Met-enkephalin, induce flexion of the left hindlimb, whereas the δ-opioid agonist Leu-enkephalin and Arg-vasopressin cause the right limb to flex. These effects mimic the UBI-induced formation of the HL-PA, and suggest that these neurohormones may be involved in the left-right side specific postural and sensorimotor effects of the UBI. The left-right side-specific neurohormonal effects may be induced through lateralized receptors. Indeed, in the rat spinal cord, the expression of the opioid

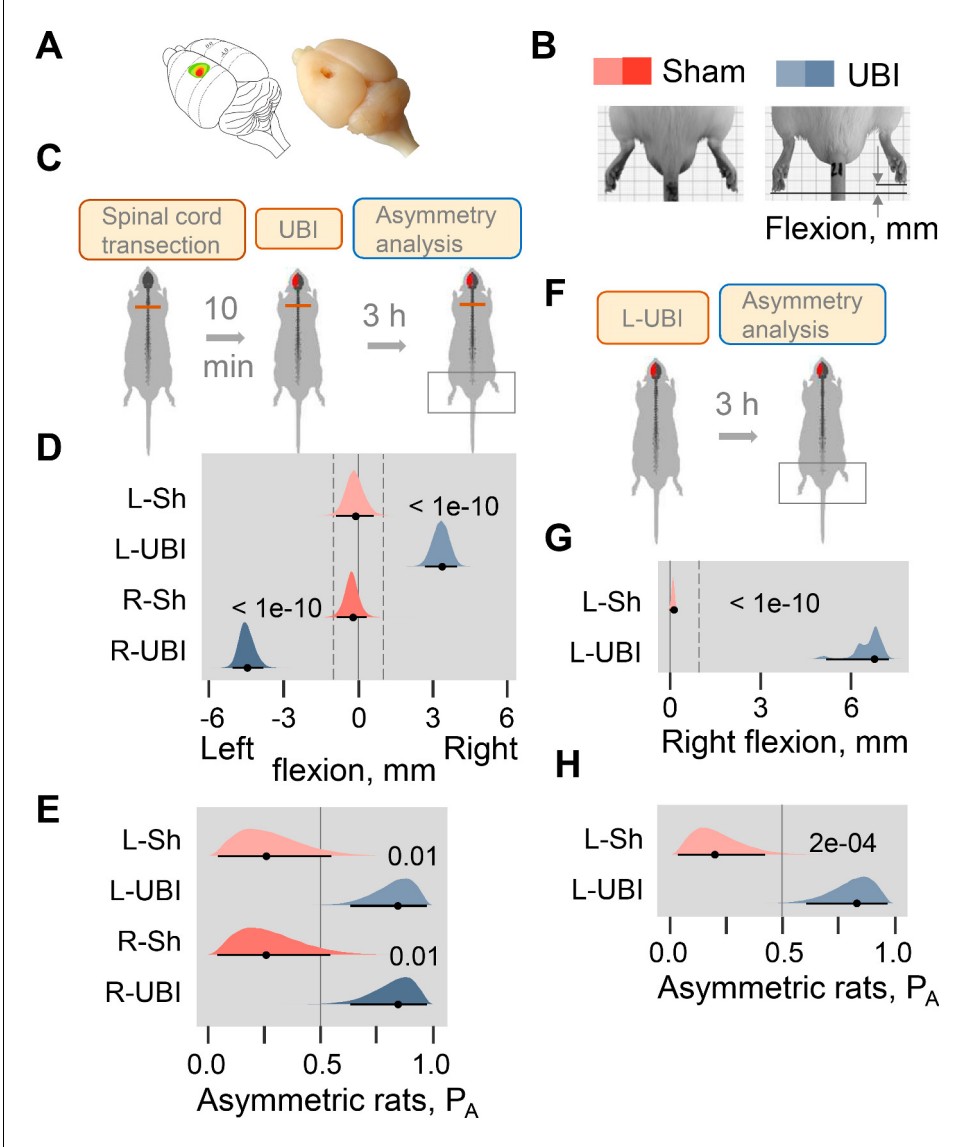

**Figure 1.** Postural asymmetry of hindlimbs induced by the unilateral ablation of the hindlimb representation area of sensorimotor cortex in rats with completely transected and intact spinal cord analyzed for comparison. (**A**) Location of the right hindlimb representation area on the rat brain surface (adapted from **Frost et al., 2013**) and a representative UBI. (**B**) HL-PA analysis. (**C–E**) HL-PA 3 hr after left UBI (n = 9) or right UBI (n = 9), and left (n = 4) or right (n = 4) sham surgery, all performed after complete spinal cord transection. (**F–H**) HL-PA 3 hr after left UBI (n = 8) or left sham surgery (n = 7) in animals with intact spinal cord. (**C,F**) Experimental design. (**D,G**) The HL-PA in millimeters (mm), and (**E,H**) the probability to develop HL-PA (P$_A$) above 1 mm threshold. Threshold is shown by vertical dotted lines in (**D,G**). The HL-PA and the probability are plotted as median (black circles), 95% HPDC intervals (black lines), and posterior density (colored distribution) from Bayesian regression. Negative and positive HL-PA values are assigned to rats with the left and right hindlimb flexion, respectively. Effects on asymmetry and differences between the groups: 95% HPDC intervals did not include zero, and adjusted p-values were ≤ 0.05. Adjusted p is shown for differences identified by Bayesian regression.

The online version of this article includes the following video, source data, source code and figure supplement(s) for figure 1:

**Source code 1.** Source code for **Figure 1D,E**.

**Source data 1.** The EXCEL source data file contains HL-PA data for panels D, E, G and H of **Figure 1**.

**Figure supplement 1.** Histological verification and UBI-produced HL-PA and motor deficits in the beam-walking and ladder rung tests in rats with intact spinal cord.

**Figure supplement 2.** Analysis of HL-PA by the hands-off method of hindlimb stretching followed by photographic recording of the asymmetry.

*Figure 1 continued on next page*

*Figure 1 continued*

**Figure supplement 3.** Time-course of HL-PA formation after the left-side (L-UBI) and right-side (R-UBI) UBI in Wistar rats with transected spinal cord.

**Figure supplement 4.** UBI induced formation of HL-PA in Sprague Dawley (**B–E**) and Wistar (**F–I**) rats with transected spinal cord.

**Figure supplement 5.** Induction of HL-PA by the left UBI in Sprague Dawley rats with dissected spinal cord segment.

**Figure 1—video 1.** Video recording of HL-PA by the hands-off method (see Materials and methods) in intact rat (episode 1), and rat subjected to the left UBI 1 day before the analysis (episodes 2 and 3).

https://elifesciences.org/articles/65247#fig1video1

receptors and their co-expression patterns are different between the left and right sides (*Kononenko et al., 2017*; *Watanabe et al., 2021*).

In this study, we tested the hypothesis that the effects of UBI on the hindlimb posture and sensorimotor functions are mediated by a side-specific neuroendocrine pathway that operates in parallel with the descending neural tracts. Our strategy was to disable the descending neural influences in order to reveal the endocrine signaling. For this purpose, the spinal cord was completely transected before the brain injury was performed. The HL-PA and asymmetry of withdrawal reflexes that are regulated by neurohormones were studied as the readouts of UBI effects. Hypophysectomy and pharmacological antagonists of opioid and vasopressin receptors were used to disable hormonal signaling. We also tested whether the administration of serum collected from animals with UBI, as well as the administration of pituitary neurohormones β-endorphin and Arg-vasopressin, may replicate the effects of brain injury by inducing HL-PA in rats with intact brain.

## Results

### Brain injury induces postural asymmetry in rats with transected spinal cord

The hypothesis that a unilateral brain lesion may induce HL-PA through a pathway that bypasses the descending neural tracts was tested in rats that had complete transection of the spinal cord before the UBI was performed (*Figure 1A–E*; *Figure 1—figure supplements 1–5*). The spinal cord was transected at the T2-T3 level and then the hindlimb representation area of the sensorimotor cortex was ablated (*Figure 1A*; *Figure 1—figure supplement 1A*). HL-PA was analyzed within 3 hr after the UBI by both the hands-on and hands-off methods of hindlimb stretching followed by photographic and / or visual recording of the asymmetry in animals under pentobarbital anesthesia (for details, see 'Materials and methods' and *Figure 1—figure supplement 2*). HL-PA data are presented as the median values of HL-PA in mm (HL-PA size), and the probability to develop HL-PA (denoted as $P_A$ on the figures) that depicts the proportion of rats with HL-PA above the 1 mm threshold. The analysis was generally blind to the observer (for details, see 'Materials and methods'). Control experiments demonstrated that this injury produced HL-PA with contralesional hindlimb flexion within 3 hr after the UBI in rats with intact spinal cord (*Figure 1F–H*; *Figure 1—figure supplement 1B–D*), and contralesional hindlimb motor deficits in the beam-walking and ladder rung tests (*Figure 1—figure supplement 1E,F*).

Strikingly, in the rats with transected spinal cords the UBI also induced HL-PA (*Figure 1C–E*). The HL-PA developed within 3 hr after the brain injury. Its size and probability were much greater than in rats with sham surgery (Left UBI, n = 31; Right UBI, n = 15; sham surgery, n = 29). An unanticipated observation was that in rats with HL-PA, the hindlimb was flexed on the contralesional side. The left or right hindlimb flexion was induced by the right and left UBI, respectively (*Figure 1D*; *Figure 1—figure supplement 3B,C*; *Figure 1—figure supplement 4B,C,F,G*). Both Wistar rats (*Figure 1D–E*; *Figure 1—figure supplement 3*, 4F-I) and Sprague Dawley rats (*Figure 1—figure supplements 4B–E* and *5*) that were used in further molecular and electrophysiological experiments, respectively, developed HL-PA with hindlimb flexion on the contralesional side. To ensure the completeness of the transection, a 3–4 mm spinal segment was excised at the T2-T3 level in a subset of rats (*Figure 1—figure supplement 5*). After the excision, the left-side UBI-induced hindlimb postural asymmetry with the right limb flexion that replicated the other findings. The HL-PA size and probability,

the time course of HL-PA development and formation of contralesional hindlimb flexion in rats with transected spinal cords that received UBI (*Figure 1D,E*; *Figure 1—figure supplement 3*, *Figure 1—figure supplement 4*) were similar to those of the UBI animals with intact spinal cords (*Figure 1G,H*; *Figure 1—figure supplement 1C,D*). We conclude that HL-PA formation in animals with transected spinal cord is mediated through a pathway that operates in parallel with the descending neural tracts and assures the development of contralesional flexion.

## Brain injury induces asymmetry in withdrawal reflexes in rats with transected spinal cord

The withdrawal reflexes are instrumental in the investigation of brain injury-induced functional changes in hindlimb neural circuits activated by afferent input (*Dewald et al., 1999*; *Schouenborg, 2002*; *Serrao et al., 2012*; *Spaich et al., 2006*; *Zhang et al., 2020*). We next sought to determine whether UBI in rats with transected spinal cords produces changes in the hindlimb withdrawal reflexes, and whether these changes are asymmetric. Special care was taken to ensure that EMG recordings obtained from the left and right hindlimbs were quantitatively comparable. To achieve this, a number of strict technical criteria, such as maximally symmetrical positioning of the stimulation and recording electrodes, were applied. The criteria used in this study are described in details in 'Materials and methods', and are similar to those proposed by Hultborn and Malmsten (*Hultborn and Malmsten, 1983a*; *Hultborn and Malmsten, 1983b*; *Malmsten, 1983*). Furthermore, to minimize inter-individual variations, the asymmetry indices were used instead of the absolute values of the reflex size. This allowed double assessment: first, within both the UBI and control groups that identified asymmetric reflexes in each group, and, second, between these groups that revealed the effects of UBI *vs.* sham surgery. Because multiple responses were measured for the same animal, including two of its limbs, four muscles, and the varying stimulation conditions, and because they were analyzed within an animal group and between the groups, we applied mixed-effects models using Bayesian inference. Only strong and significant UBI effects were considered as biologically relevant.

Electromyographic responses were recorded from the extensor digitorum longus, interosseous, peroneus longus, and semitendinosus muscles of the contra- and ipsilesional hindlimbs in the rats with UBI (n = 18) or sham surgery (n = 11) performed after complete spinal transection and analyzed as the asymmetry index (AI = $\log_2$[Contra / Ipsi], where Contra and Ipsi were values for muscles of the contralesional and ipsilesional limbs) (*Figure 2*; *Figure 2—figure supplement 1*; *Figure 2—figure supplement 2*). When reflexes on both sides are equal (i.e. the Contra / Ipsi ratio equals 1), the asymmetry index is zero; if reflexes are doubled in size on the Contra or Ipsi side (i.e. the Contra / Ipsi ratio equals 2.0 or 0.5) the asymmetry index is +1 or –1, respectively.

Analysis of the electrically evoked electromyographic responses revealed that the asymmetry index was different from zero in the current threshold for the semitendinosus muscle (3.6-fold lower on the contra- *vs.* ipsilesional side), and in the number of spikes for the extensor digitorum longus (3.5-fold higher on the contra- *vs.* ipsilesional side) and semitendinosus (5.9-fold higher on the contra- *vs.* ipsilesional side) muscles in UBI rats (*Figure 2C,D*). No contra- *vs.* ipsilesional asymmetry was evident in the sham surgery group. Representative UBI-induced asymmetry in the number of spikes for the semitendinosus muscle is shown in *Figure 2A,B* (for those of extensor digitorum longus, interosseous and peroneus longus muscles, see *Figure 2—figure supplement 1*).

When compared to sham surgery, UBI substantially decreased the asymmetry index for the current threshold of the semitendinosus (4.0-fold), and the asymmetry index for the number of spikes of the interosseous (2.8-fold) that may be due to the decline in the responses on the contralesional side and/or their elevation on the ipsilesional side. Concomitantly, UBI elevated the asymmetry index for the number of spikes of the extensor digitorum longus (5.2-fold) and semitendinosus (6.7-fold) suggesting activation of the responses on the contralesional side and/or their inhibition on the ipsilesional side (*Figure 2E,F*). No changes in peroneus longus were revealed. Each group consisted of rats with left and right sided surgeries (*Figure 2—figure supplement 2*). Analysis of the asymmetry index for the four groups (i.e. the left UBI, left sham surgery, right UBI and right sham surgery groups) revealed virtually the same asymmetries in the UBI group, and the same UBI *vs.* sham differences in the asymmetry index, but not for all comparisons (*Figure 2—figure supplement 3*). Thus, the right UBI produced higher responses of the left (contralesional) extensor digitorum longus (4.1-fold) and the left (contralesional) semitendinosus (53.8-fold) compared to those on the right side,

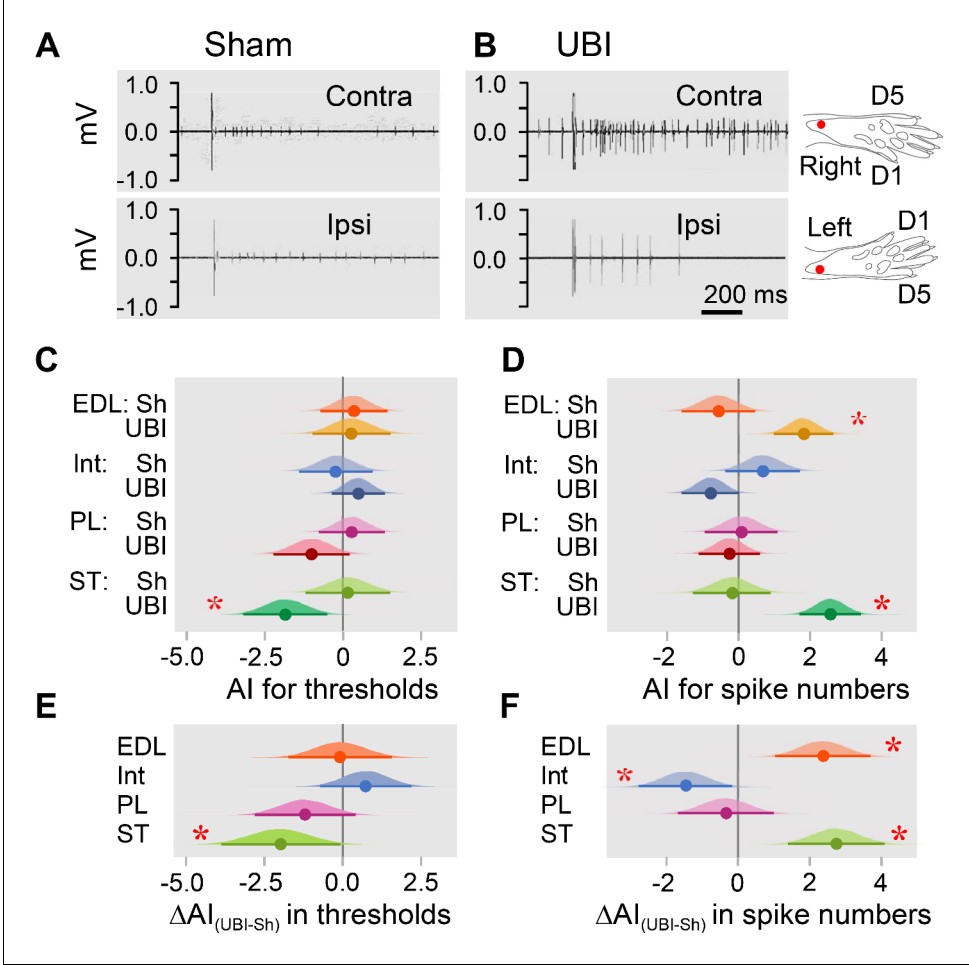

**Figure 2.** Hindlimb withdrawal reflexes in rats with UBI performed after complete spinal cord transection. EMG activity of left and right extensor digitorum longus (EDL), interosseous (Int), peroneus longus (PL) and semitendinosus (ST) muscles were evoked by electrical stimulation of symmetric paw sites. The number of observations for both UBI and sham group is shown in *Figure 2—figure supplement 2*. The rats were subjected to left (n = 6) or right (n = 5) sham surgery, or to the left (n = 9) or right (n = 9) UBI. (**A,B**) Representative semitendinosus responses. (**C,D**) Asymmetry index (AI=log$_2$[Contra/Ipsi]) for threshold and spike number. Differences in the asymmetry index from zero (AI = 0 when the ratio [Contra/Ipsi] = 1, that corresponds to a symmetric pattern) in UBI rats (**C**) in the current threshold for the semitendinosus muscle {median of the posterior distribution (median) = −1.840, 95% highest posterior density continuous interval (HPDCI) = [−3.169,–0.477], adjusted p-value (p) = 0.015, fold difference = 3.6}; and (**D**) in the number of spikes for the extensor digitorum longus (median = 1.818, HPDCI = [0.990, 2.655], p = 4×10$^{−5}$, fold difference = 3.5) and semitendinosus (median = 2.560, HPDCI = [1.691, 3.415], p = 1×10$^{−8}$, fold difference = 5.9) muscles. (**E,F**) Differences in the asymmetry index between the UBI and sham surgery (Sh) groups [ΔAI$_{(UBI − Sh)}$]. Differences in the asymmetry index between the UBI and sham surgery groups for (**E**) the current threshold of the semitendinosus (median = −1.992, HPDCI = [−3.911,–0.106], p = 0.040, fold difference = 4.0); and (**F**) the number of spikes of the interosseous (median = −1.463, HPDCI = [−2.782,–0.159], p = 0.028, fold difference = 2.8), extensor digitorum longus (median = 2.379, HPDCI = [1.080, 3.743], p = 4×10$^{−4}$, fold difference = 5.2), and semitendinosus (median = 2.745, HPDCI = [1.419, 4.128], p = 6×10$^{−5}$, fold difference = 6.7). Medians, 95% HPDC intervals and densities from Bayesian sampler are plotted. * Asymmetry and differences between the groups: 95% HPDC intervals did not include zero, and adjusted p-values were ≤ 0.05.

The online version of this article includes the following source data, source code and figure supplement(s) for figure 2:

**Source code 1.** Source code for *Figure 2C,E*.

**Source code 2.** Source code for *Figure 2D,F*.

**Source data 1.** The EXCEL source data file contains EMG data for panels C and D of *Figure 2*.

**Figure supplement 1.** Effects of the UBI on hindlimb withdrawal reflexes.

*Figure 2 continued on next page*

*Figure 2 continued*

**Figure supplement 2.** The number of rats analyzed in EMG experiments.
**Figure supplement 3.** The UBI effects on the hindlimb withdrawal reflexes: separate analysis of left or right brain injury.

while responses of the left *vs.* right interosseous were decreased (5.5-fold) (*Figure 2—figure supplement 3D*). Additionally, the interosseous muscle was found to be asymmetric after the right side UBI. No effects on thresholds were identified.

Thus, in rats with transected spinal cord, UBI, but not sham surgery, induced asymmetry in withdrawal reflexes. The number of spikes of both flexor muscles, the extensor digitorum longus and semitendinosus was higher on the contra *vs.* ipsilateral side in the UBI rats. Consistently, the threshold was lower for the contra *vs.* ipsilesional semitendinosus. These effects may be due to (i) higher sensitivity of the afferent system reflected in a lower threshold on the contra *vs.* ipsilesional side for the semitendinosus; and (ii) an increased excitability of efferent systems for both muscles reflected in the increased number of spikes on the contra *vs.* ipsilesional side. At the cellular level, the increased excitatory drive may develop due to changes in local spinal circuits including those in presynaptic afferent inhibition, and/or changes in intrinsic membrane properties of motoneurons. Regardless of mechanism, robust differences in the asymmetry index between the UBI and sham groups suggested that the UBI markedly elevated both the sensitivity of semitendinosus afferents and the excitability of the extensor digitorum longus and semitendinosus efferents, all on the contralesional *vs.* ipsilesional side. The UBI also inhibited the contralesional interosseous. The UBI effects on the hindlimb withdrawal reflexes in rats with the transected spinal cords were similar in their range and contra-ipsilesional patterns to those of the UBI animals with intact spinal cords (*Watanabe et al., 2021*; *Zhang et al., 2020*). These effects corroborate clinical findings showing contralateral facilitation of withdrawal reflexes in stroke patients (*Dewald et al., 1999*; *Serrao et al., 2012*; *Spaich et al., 2006*).

## Brain injury produces molecular changes in the lumbar spinal cord

We examined whether the UBI performed after complete spinal transection produced molecular changes in the lumbar spinal segments. Expression of 20 neuroplasticity-related, opioid and vasopressin genes, and the levels of three opioid peptides were analyzed in the ipsilesional and contralesional lumbar spinal cord of the rats with transected spinal cord that also had the left UBI (n = 12) or left sham surgery (n = 11). Genes coding for regulators of axonal sprouting, synapse formation, neuronal survival and neuroinflammation (*Arc*, *Bdnf*, *Dlg4*, *Homer-1*, *Gap43*, *Syt4*, and *Tgfb1*), transcriptional regulators of synaptic plasticity (*cFos*, *Egr1*, and *Nfkbia*), and essential components of the glutamate system critical for neuroplastic responses and regulation of spinal reflexes (*GluR1*, *Grin2a*, and *Grin2b*) were selected as neuroplasticity genes (for detailed description, see 'Materials and methods, Neuroplasticity-related genes'). Genes of the opioid and vasopressin systems were included because of their involvement in asymmetric spinal responses to brain injury (see next section).

First, the mRNA levels and their median asymmetry index (AI = $\log_2$[Contra/Ipsi], where Contra and Ipsi were the levels in the contralesional and ipsilesional lumbar spinal cord) were compared between the UBI and sham surgery groups. Gene expression was either elevated on the ipsilesional side (*Syt4*, *Grin2a*, *Grin2b*, and *Oprk1*; *Figure 3—figure supplement 1A–D*) or decreased on the contralesional side (*Gap43* and *Penk*; *Figure 3—figure supplement 1E,F*) (for all six genes, $P_{unadjusted}$ < 0.05). Consistently, the gene expression asymmetry index was decreased for *Syt4* ($P_{adjusted}$ = 0.004), and for *Oprk1*, *Oprm1*, *Dlg4*, and *Homer1* (for all four genes, $P_{unadjusted}$ < 0.05) (*Figure 3—figure supplement 1G–K*). These differences were subtle, between 0.13 and 0.38-fold, and therefore were discarded as evidence of the UBI effects.

These differences, however, pointed to the different direction in responses of the left and right spinal cord to the injury. Therefore, in the second step, we assessed whether the proportion of genes with lower expression on the contralesional *vs.* ipsilesional side was different between the UBI and sham surgery groups. The median gene expression asymmetry index of 19 out of 20 genes at the pairwise comparison was lower in the UBI rats compared to sham surgery group (sign-test: p =

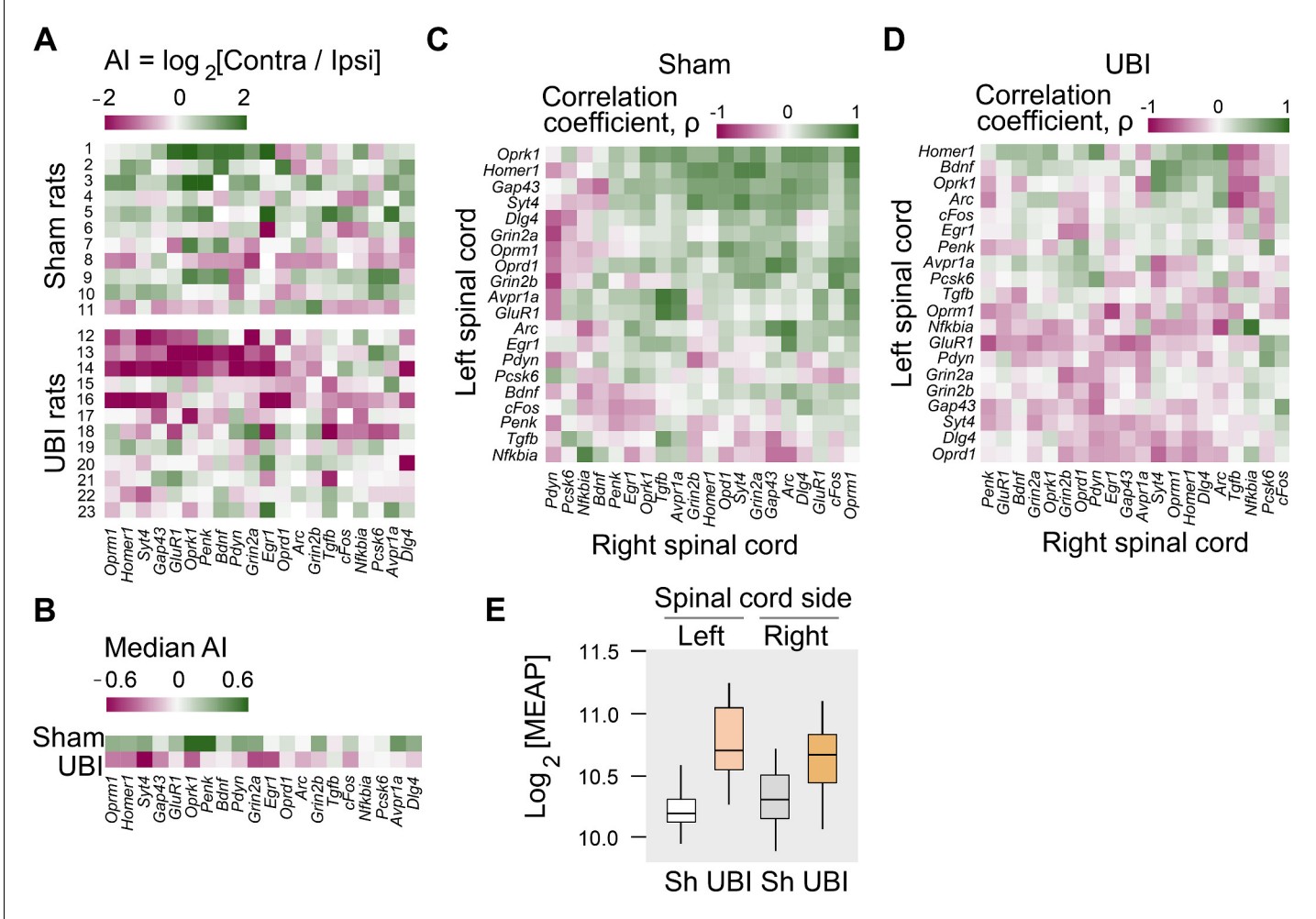

**Figure 3.** Expression of neuroplasticity-related and neuropeptide genes in the lumbar spinal cord of rats with left UBI performed after complete spinal cord transection. The mRNA and peptide levels were analyzed in the ipsi- and contralesional halves of lumbar spinal cord isolated from rats 3 hr after the left UBI (n = 12) or left sham surgery (Sh; n = 11). (A,B) Heatmap for the (0,1)-standardized expression asymmetry index (AI=log$_2$[Contra/Ipsi]) for each gene denoted for each rat individually, and as medians for rat groups. (C,D) Heatmap for Spearman's rank correlation coefficients of expression levels between the left- and right lumbar halves for all gene pairs (inter-area correlations) in rats with transected spinal cord that were subjected to sham surgery or UBI. (E) UBI effects on the Met-enkephalin-Arg-Phe (MEAP) levels in the left ($P_{adjusted}$ = 9×10$^{-4}$; fold change: 1.4×) and right ($P_{unadjusted}$ = 0.020; fold change: 1.3×) halves. Data are presented in fmol/mg tissue in the log$_2$ scale as boxplots with median and hinges representing the first and third quartiles, and whiskers extending from the hinge to the highest/lowest value that lies within the 1.5 interquartile range of the hinge.

The online version of this article includes the following source data and figure supplement(s) for figure 3:

**Source data 1.** The EXCEL source data file contains data for panels A,B of *Figure 3*.

**Source data 2.** The EXCEL source data file contains data for panel E of *Figure 3*.

**Figure supplement 1.** UBI effects on expression of neuroplasticity-related and neuropeptide genes (A–F), and the expression asymmetry index (G–K) in the lumbar spinal cord of rats with transected spinal cord.

**Figure supplement 2.** Heatmap for expression levels (A–D) and intra-area correlations (E–H) in the left (ipsilesional) (A,C,E,G) and right (contralesional) (B,D,F,H) lumbar spinal cord.

**Figure supplement 3.** UBI effects on the levels of opioid peptides Dynorphin B (DynB) and Leu-enkephalin-Arg (LER) in the lumbar spinal cord of the rats with transected spinal cord.

4×10$^{-5}$) (*Figure 3A,B*). Changes in the gene expression asymmetry index were consistent with decreased expression of 17 genes (sign test: p = 0.003) in the contralesional half (*Figure 3—figure supplement 1E,F*; *Figure 3—figure supplement 2A–D*) concomitantly with elevated expression of 15 genes (sign test: p = 0.041) in the ipsilesional half (*Figure 3—figure supplement 1A–D*; *Figure 3—figure supplement 2A–D*).

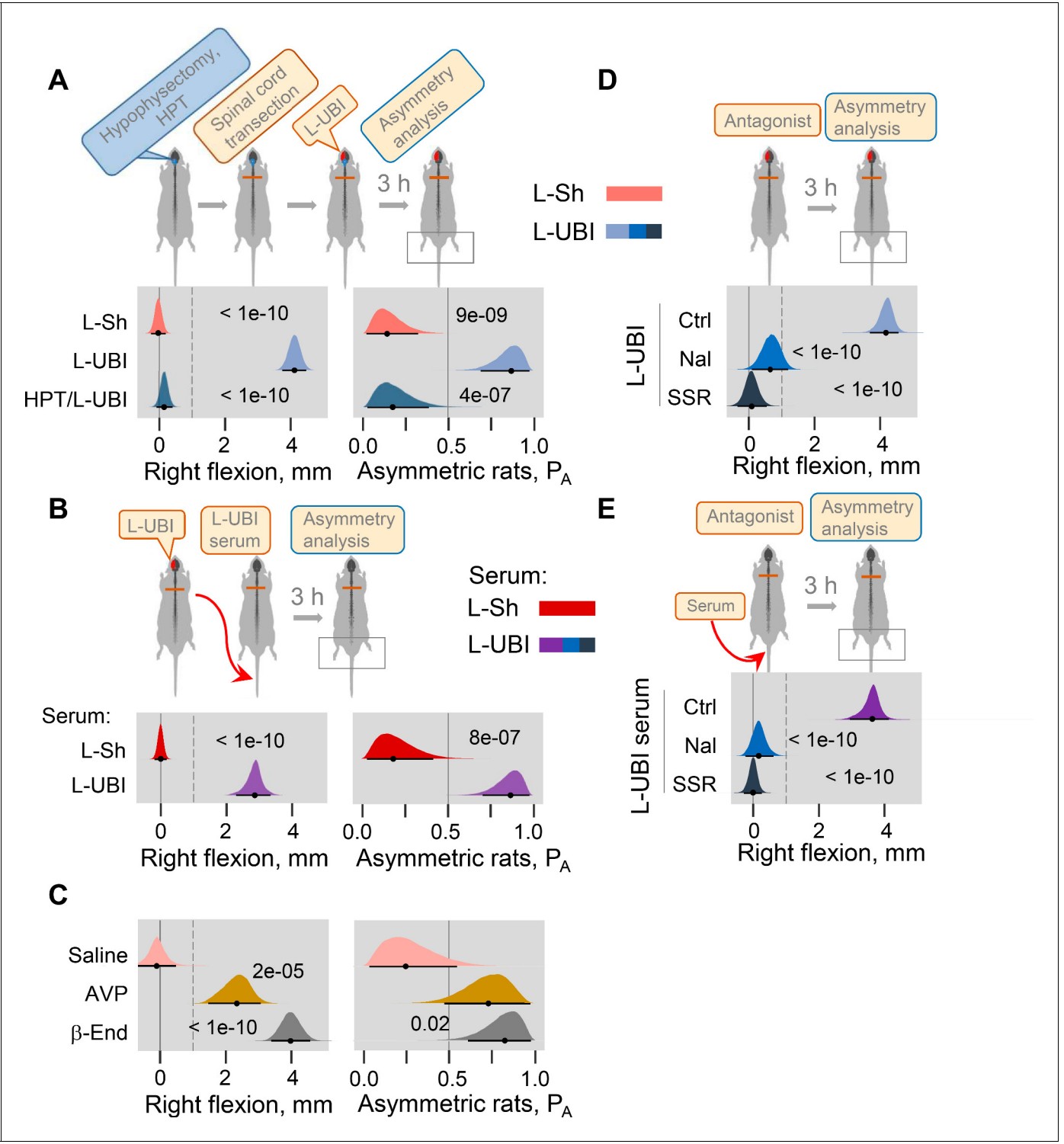

**Figure 4.** Neuroendocrine pathway mediating postural asymmetry formation in rats with transected spinal cord. (**A**) HL-PA in hypophysectomized (HPT; n = 8) and control (n = 12) rats with transected spinal cord 3 hr after left UBI (L-UBI). Left sham surgery (L–Sh): n = 8. (**B**) HL-PA after intravenous administration of serum from rats with either left UBI (L-UBI serum) or left sham surgery (L-Sh serum) to rats with transected spinal cord (n = 13 and 7, respectively). (**C**) Induction of HL-PA by Arg-vasopressin (AVP) and β-endorphin (β-End) in rats with transected spinal cord. Synthetic β-endorphin or Arg-vasopressin (1 microgram and 10 nanogram / 0.3 ml saline / animal, respectively), or saline was administered intravenously to rats (n = 8, 7, and 4 rats, respectively) after spinal cord transection. The HL-PA was analyzed in rats 60 min after the injection and under pentobarbital anesthesia. (**D**) Effect of naloxone (Nal, n = 6) or saline (n = 6), and SSR-149415 (SSR, n = 6) or vehicle (n = 5) on HL-PA 3 h after left UBI in rats with transected spinal cord. Vehicle and saline groups were combined into a single control group (Ctrl; n = 11). (**E**) Effect of naloxone (n = 6) or saline (n = 3) and SSR-149415 (n = 6)

*Figure 4 continued on next page*

*Figure 4 continued*

or vehicle (n = 3) on HL-PA 3 hr after intravenous administration of the left UBI serum to rats with transected spinal cord. Ctrl = saline + vehicle; n = 6. In (D,E), naloxone (or saline) and SSR-149415 (or vehicle) were administered 0.5 and 3 hr before HL-PA analysis, respectively. HL-PA values in millimeters (mm) and probability ($P_A$) to develop HL-PA above 1 mm threshold (denoted by vertical dotted lines) are plotted as median, 95% HPDC intervals, and posterior distribution from Bayesian regression. Negative and positive HL-PA values are assigned to rats with the left and right hindlimb flexion, respectively. Asymmetry and differences between the groups: 95% HPDC intervals did not include zero, and adjusted p-values were ≤ 0.05. Adjusted p is shown for differences identified by Bayesian regression.

The online version of this article includes the following source data and figure supplement(s) for figure 4:

**Source data 1.** The EXCEL source data file contains data for panels A-E of *Figure 4*.
**Figure supplement 1.** HL-PA formation in hypophysectomized rats with UBI, and in rats with intact brain after administration of serum of the UBI animals.
**Figure supplement 2.** Induction of HL-PA by Arg-vasopressin (AVP) administered intracisternally to rats with intact brain.

Gene co-expression patterns characterize regulatory interactions within and across tissues (*Dobrin et al., 2009*; *Erola et al., 2020*; *Major Depressive Disorder Working Group of the Psychiatric Genomics Consortium et al., 2019*; *Zhang et al., 2020*). Third, we examined whether the UBI induced changes in mRNA–mRNA correlations within the left and right half of the lumbar spinal cord (intra-area correlations), and between these halves (inter-area correlations). The proportion of intra-area positive correlations, which dominated in rats with sham surgery, was reduced after the UBI (Fisher's Exact Test: all correlations in the right half, $p = 3 \times 10^{-5}$; significant correlations in the left and right areas, $p = 0.008$ and $0.009$, respectively) (*Figure 3—figure supplement 2E–H*). The inter-area gene-gene coordination strength was decreased after the UBI (Wilcoxon signed-rank test; all and significant correlations: $p = 4 \times 10^{-7}$ and $3 \times 10^{-4}$, respectively) (*Figure 3C,D*). Positive inter-area correlations were predominant in rats with sham surgery (68%) in contrast to the UBI rats (42%) (Fisher's Exact Test: all and significant correlations, $p = 6 \times 10^{-14}$ and $0.004$, respectively). Thus, the UBI robustly impairs coordination of expression of neuroplasticity-related and neuropeptide genes within and between the left and right halves of the lumbar spinal cord.

Fourth, analysis of opioid peptides demonstrated that the UBI substantially elevated the levels of the proenkephalin marker Met-enkephalin-Arg-Phe in the ipsilesional ($P_{adjusted} = 9 \times 10^{-4}$) and contralesional ($P_{unadjusted} = 0.020$) spinal halves (*Figure 3E*), and the prodynorphin-derived Dynorphin B and Leu-enkephalin-Arg in the ipsilesional spinal cord (for both, $P_{unadjusted} < 0.05$) (*Figure 3—figure supplement 3*).

Altogether, the analysis of gene expression and of opioid peptides adds strong molecular evidence for the lateralized signaling from the injured brain to the lumbar neural circuits in rats with transected spinal cord.

## UBI effects are mediated by neuroendocrine pathway

The left-right side specific mechanism that does not engage the descending neural tracts may operate through the neuroendocrine system by a release of pituitary hormones into the blood. Consistent with this hypothesis, no HL-PA developed in hypophysectomized animals that received left UBI after spinal transection (n = 8); the HL-PA median values and $P_A$ were nearly identical to those in sham operated rats (n = 8) (*Figure 4A*; *Figure 4—figure supplement 1A–E*). We next examined whether left UBI stimulates the release of chemical factors that may induce the development of HL-PA, into the blood. Serum that was collected 3 hr after performing a left UBI in rats with transected spinal cord was administered either centrally (into the cisterna magna; UBI serum, n = 13; sham serum, n = 7; *Figure 4—figure supplement 1F–J*) or intravenously (UBI serum, n = 13; sham serum, n = 7; *Figure 4B*; *Figure 4—figure supplement 1K–O*) to rats after their spinalization. Serum administration by either route resulted in formation of HL-PA with its values and its probability similar to those induced by the UBI in rats with intact and transected spinal cords. Remarkably, animals injected with serum from rats with left UBI displayed hindlimb flexion on the right side, which was the same as the flexion side in the donor rats (*Figure 4B*; *Figure 4—figure supplement 1F–O*). No HL-PA developed after administration of serum collected from rats with the left sham surgery. We conclude that the left UBI stimulates a release of chemical factors from the pituitary gland into the blood that induce HL-PA with contralesional flexion.

Previous studies demonstrated that multiple peptide factors extracted from the brain, pituitary gland and serum may induce a side-specific hindlimb motor response. Several of them were identified as peptide neurohormones including opioid peptides (*Bakalkin and Kobylyansky, 1989*; *Bakalkin et al., 1986*; *Chazov et al., 1981*) and Arg-vasopressin (*Klement'ev et al., 1986*). It was found that Arg-vasopressin or Leu-enkephalin administered centrally induced HL-PA with right hindlimb flexion. The pituitary gland is the main source of the opioid neurohormone β-endorphin and the antidiuretic hormone Arg-vasopressin in the body. Here, we first replicated the effects of Arg-vasopressin that, consistent with a previous study (*Klement'ev et al., 1986*), produced flexion of the right hindlimb after its intracisternal administration (*Figure 4—figure supplement 2*; peptide, n = 22, and saline, n = 9 at the 180 min time point). We then tested if β-endorphin and Arg-vasopressin may evoke asymmetric motor response after intravenous administration. Injection of these neurohormones, but not saline, to rats with transected spinal cords resulted in development of HL-PA with right hindlimb flexion (*Figure 4C*; β-endorphin, n = 8; Arg-vasopressin, n = 7; saline, n = 4).

We next investigated whether opioid receptors and the vasopressin receptor V1B, that is expressed in the pituitary gland (*Roper et al., 2011*), mediate formation of HL-PA in UBI rats or in animals treated with serum from UBI rats. Naloxone and SSR-149415, the opioid and vasopressin V1B receptor antagonists, respectively, administered to animals with transected spinal cord that also received a left UBI (naloxone, n = 6; SSR-149415, n = 6; saline and vehicle, n = 11) inhibited HL-PA formation (*Figure 4D*). Similarly, the HL-PA induced by serum from animals with left UBI was abolished by administration of either naloxone (n = 6) or SSR-149415 (n = 6) (*Figure 4E*). Thus, the pituitary neurohormones β-endorphin and Arg-vasopressin released into the systemic circulation may serve as side-specific signals that mediate UBI effects on hindlimb motor circuits.

## Discussion

### The left and right hemispheres in top-down control of the endocrine system

During embryonic development, the left–right asymmetry of the body is generated by multiple paracrine signaling molecules that enable communications between the left and right halves of the embryo (*Hamada et al., 2002*). In the adult brain, functional lateralization is an organizing principle (*Duboc et al., 2015*; *MacNeilage et al., 2009*) and lateralized functions may be regulated by paracrine signaling molecules including peptide neurohormones (*Deliagina et al., 2000*; *Hussain et al., 2012*; *Kononenko et al., 2017*; *Marlin et al., 2015*; *Nation et al., 2018*; *Phelps et al., 2019*; *Watanabe et al., 2015*; *Watanabe et al., 2021*; *Zink et al., 2011*). Thus, oxytocin enables retrieval behavior by enhancing responses of the left, but not right, auditory cortex through its receptors expressed on the left side (*Marlin et al., 2015*). Arg-vasopressin targets the left but not right hemispheric areas to modulate social recognition-related activity (*Zink et al., 2011*). In the human brain, asymmetric distribution of the μ-opioid receptor along with opioid peptides that elicit euphoria and dysphoria may provide a basis for the lateralized processing of positive and negative emotions (*Kantonen et al., 2020*; *Watanabe et al., 2015*).

Top-down control of the hypothalamic-pituitary-adrenal and hypothalamic-pituitary-gonadal axes, as well as the immune system, is left-right hemisphere specific (*Bakalkin et al., 1984*; *Inase and Machida, 1992*; *Kiss et al., 2020*; *Lueken et al., 2009*; *Madsen et al., 2012*; *Meador et al., 2004*; *Sullivan and Gratton, 2002*; *Xavier et al., 2013*). The right-sided injuries to the brain or spinal nerves compared to those on the left side produce stronger effects on the neuroendocrine systems, including hormonal levels in the peripheral circulation (*Hussain et al., 2012*; *Inase and Machida, 1992*; *Kononenko et al., 2017*; *Lueken et al., 2009*). Cortisol (corticosterone) secretion is under excitatory control of the right hemisphere and, consistently, its phasic response is diminished in patients with right but not left sided stroke (*Lueken et al., 2009*). These features may be related to the asymmetry of the hypothalamic – pituitary axis. Thus, basal and corticotropin-releasing hormone–induced secretion of Arg-vasopressin and ACTH by the pituitary gland is lateralized to the right petrosal sinus (*Kalogeras et al., 1996*). Conversely, non-directional stimuli such as stress and stress-induced pain produce lateralized responses in the CNS (*Bakalkin et al., 1982*; *Bakalkin et al., 1984*; *Nation et al., 2018*; *Sullivan and Gratton, 2002*). Both left- and right-sided nerve or body injuries elicit functional and molecular responses that are often lateralized to the right, but not to the

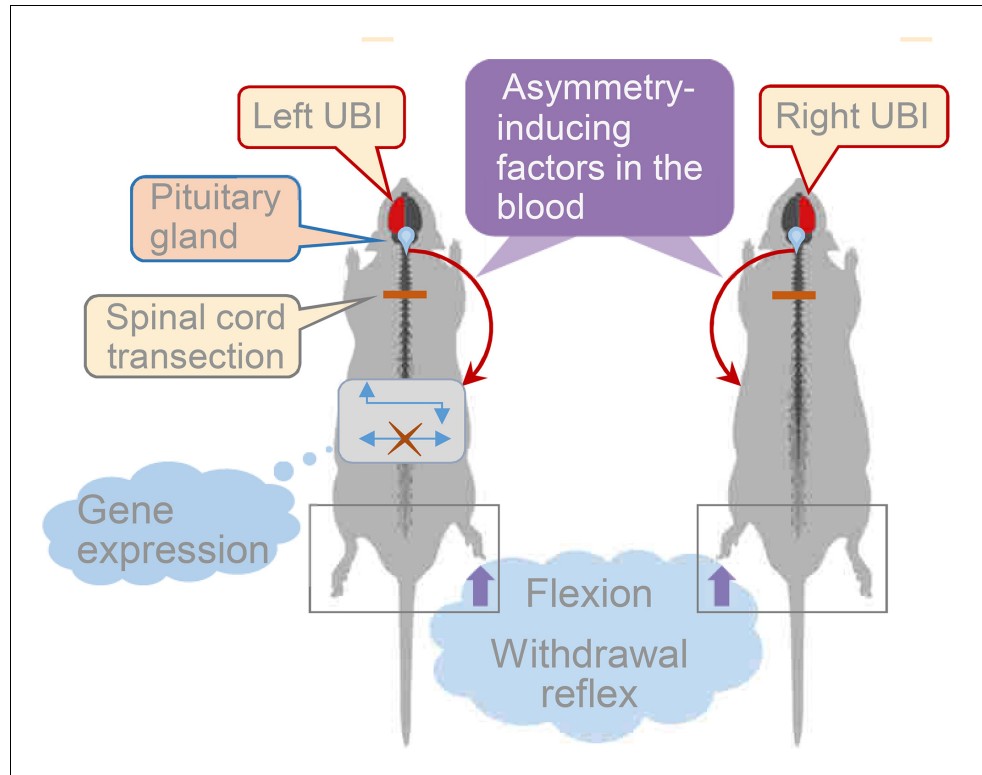

**Figure 5.** Model for the humoral neuroendocrine side-specific signaling from the unilaterally injured brain to the lumbar spinal cord. In the rats with transected spinal cords, the UBI stimulates the release of the asymmetry inducing factors (neurohormones) from the pituitary gland into the circulation. They are transported to their target sites and induce flexion of the contralesional hindlimb and asymmetric, contra *vs*. ipsilesional side specific changes in withdrawal reflexes and spinal gene expression patterns.

left, in the brain and spinal cord (*Bakalkin et al., 1984*; *Hussain et al., 2012*; *Inase and Machida, 1992*; *Kononenko et al., 2017*; *Phelps et al., 2019*). Neuropathic pain induced by either the left- or right-sided nerve injuries is controlled through κ-opioid receptors in the right amygdala (*Phelps et al., 2019*).

Taken together, these findings suggest that feedforward and feedback interactions between the lateralized CNS features and the endocrine system controlling peripheral processes are mediated either by neural circuits with unusual, asymmetric organization, or by left-right sided neuroendocrine pathways that may be similar to the left-right sided paracrine mechanism operating in the development. An alternative pathway that does not engage descending neural tracts may convey side-specific signals from the brain to the paired endocrine glands, the left and right spinal cord, and the left and right extremities had been suggested (*Bakalkin et al., 1984*; *Cope et al., 1980*; *Wolpaw and Lee, 1989*), and supported by preliminary evidence (*Bakalkin et al., 1986*) but not elaborated.

## The left-right side-specific humoral signaling

This study provides evidence for left-right side-specific humoral signaling that mediates the effects of UBI on the formation of HL-PA, asymmetry in withdrawal reflexes, and asymmetric changes in gene expression patterns in the lumbar spinal cord (*Figure 5*). Encoding of information about the injury side in a hormonal message, humoral transmission of this message to its target sites on peripheral nerve endings or spinal neurons, and translation of this message into the left-right side-specific response, are the key stages of this phenomenon.

The left- and right-side-specific responses evoked by hormonal molecules circulating in the blood is a core of the humoral signaling pathway. Together with previous reports (*Bakalkin et al., 1981*; *Bakalkin and Kobylyansky, 1989*; *Bakalkin et al., 1986*; *Chazov et al., 1981*; *Watanabe et al., 2020*), this study demonstrates that peptide neurohormones and opioids administered

intravenously, intrathecally or intracisternally induce HL-PA in rats with intact brain. The critical finding is that the side of the flexed limb depends on the compound administered. Endogenous and synthetic κ-opioid agonists dynorphin, bremazocine, and U-50,488, along with the endogenous μ/δ-opioid agonist Met-enkephalin, induce flexion of the left hindlimb (*Bakalkin et al., 1981*; *Bakalkin and Kobylyansky, 1989*; *Bakalkin et al., 1986*; *Chazov et al., 1981*; *Watanabe et al., 2020*). In contrast, β-endorphin and Arg-vasopressin, and the δ-agonist Leu-enkephalin, cause the right limb to flex [the present study and *Bakalkin et al., 1981*; *Chazov et al., 1981*; *Klement'ev et al., 1986*]. Thus, topographical information conveyed by the 'non-directional' molecular messengers circulating in the blood is converted into side-specific motor responses. Hypophysectomy disables the endocrine pathway including its opioid and Arg-vasopressin components and abolishes the HL-PA. Consistent with this finding, serum from rats with left UBI induces HL-PA with contralesional hindlimb flexion in rats with intact brain. The pituitary gland is the main source of Arg-vasopressin and β-endorphin in the body that are secreted into the bloodstream (*Autelitano et al., 1989*; *Day and Akil, 1989*). Naloxone and SSR-149415 block the left UBI-induced formation of HL-PA. Altogether, these findings demonstrate that opioid and Arg-vasopressin neurohormones transmit side-specific signals from the injured brain to the spinal neural circuits.

Theoretically, the paravertebral chain of sympathetic ganglia, which is the remaining neural connection after complete spinal cord transection, may convey supraspinal signals to the muscle vasculature and through this mechanism may differentially affect ipsi- and contralesional muscles. However, the sympathetic ganglia likely do not mediate control of lumbar neural circuits by the supraspinal structures (*Brodal, 1981*; *Wolpaw and Lee, 1989*). Furthermore, the sympathetic system has a limited capacity to independently regulate blood flow to the left and right hindlimbs (*Lee et al., 2007*). Our present findings could not rule out a role of the sympathetic pathway. However, experiments with hypophysectomized rats, 'pathological' serum and neurohormones inducing the HL-PA strongly suggest the dominance of the humoral pathway in rats with transected spinal cord.

## Functional and mechanistic implications

Clinical studies revealed robust functional changes induced by stroke or traumatic brain injury (TBI) in contralateral withdrawal reflexes that also control posture and locomotion (*Dewald et al., 1999*; *Sandrini et al., 2005*; *Serrao et al., 2012*). Patients with post-stroke motor deficits lose their ability to modulate the withdrawal reflexes that affects spatiotemporal interaction among joints and causes movement abnormalities during motor activities (*Bohannon and Smith, 1987*; *Serrao et al., 2012*).

Spinal withdrawal reflexes are regulated by the endogenous opioid peptides that may suppress the ipsilateral and contralateral segmental reflexes (*Clarke et al., 1992*; *Duarte et al., 2019*; *Jankowska and Schomburg, 1998*; *Schmidt et al., 1991*). Opioid receptors are expressed both in the dorsal and ventral horns of the spinal cord (*Kononenko et al., 2017*; *Wang et al., 2018*) and also in the periphery, in primary afferents including low-threshold mechanoreceptors that modulate cutaneous mechanosensation (*Bardoni et al., 2014*; *Snyder et al., 2018*). The left-right-specific endocrine mechanism may mediate the effects of UBI on the withdrawal reflexes through targeting peripherally or centrally located opioid receptors.

A large fraction of patients with stroke and cerebral palsy do not relax their muscles – they are tonically contracted without any voluntary command (*Baude et al., 2019*; *Gracies, 2005*; *Lorentzen et al., 2018*; *Sheean and McGuire, 2009*; *Trompetto et al., 2019*). This phenomenon is called 'spastic dystonia' and has a central mechanism that does not depend on afferent input (*Gracies, 2005*; *Lorentzen et al., 2018*; *Sheean and McGuire, 2009*). In the HL-PA analysis, no nociceptive stimulation is applied and tactile stimulation is negligible [this study and *Zhang et al., 2020*]. Stretch and postural limb reflexes are abolished immediately after complete spinal cord transection (*Frigon et al., 2011*; *Miller et al., 1996*; *Musienko et al., 2010*) and strongly inhibited by anesthesia (*Fuchigami et al., 2011*; *Zhou et al., 1998*). Therefore, in anesthetized rats with transected spinal cord, a role of nociceptive and stretch reflexes in the UBI-induced HL-PA formation may be limited. The finding that HL-PA is resistant to bilateral lumbar dorsal rhizotomy (*Zhang et al., 2020*) further supports this notion and suggests that the HL-PA and the clinical 'spastic dystonia' may be mechanistically similar.

Classical hyperreflexia develops during several weeks after the impact and is considered as pathology of the corticospinal tract (*Williams et al., 2017*). In contrast, the HL-PA is formed within 30 min after the UBI in rats with transected spinal cord. Whether the endocrine signaling has a role

in exacerbation of stretch reflex after damage to the corticospinal tract would be interesting to study.

Single injection of either naloxone or SSR-149415 inhibited HL-PA asymmetry formation. The vasopressin receptor V1B is largely expressed in the anterior pituitary by corticotrophs producing proopiomelanocortin (*Roper et al., 2011*). A plausible scenario is that Arg-vasopressin released from neurohypophysis activates the V1B receptor and stimulates secretion of proopiomelanocortin-derived β-endorphin (*Roper et al., 2011*) that induces the HL-PA. Alternatively, Arg-vasopressin and β-endorphin may produce synergistic effects acting through the complex of the vasopressin receptor V1B and μ-opioid receptor that integrates two signaling pathways (*Koshimizu et al., 2018*).

The general opioid antagonists naloxone and naltrexone may normalize neurological functions that are impaired in animals and human patients after unilateral cerebral ischemia (*Baskin and Hosobuchi, 1981*; *Baskin et al., 1984*; *Baskin et al., 1994*; *Hans et al., 1992*; *Hosobuchi et al., 1982*; *Jabaily and Davis, 1984*; *Namba et al., 1986*; *Skarphedinsson et al., 1989*; *Wang et al., 2019*), and also reduce spasticity in patients with primary progressive multiple sclerosis (*Gironi et al., 2008*). Our findings suggest that the efficacy of pharmacological treatment may depend on topographical correspondence between the side of neuronal deficits and the side that is preferentially targeted by neurohormones and their antagonists.

## Lateralized features of the spinal cord

A number of anatomical, functional, and molecular studies revealed left–right asymmetry in the spinal cord organization (*de Kovel et al., 2017*; *Deliagina et al., 2000*; *Hultborn and Malmsten, 1983a*; *Hultborn and Malmsten, 1983b*; *Knebel et al., 2018*; *Kononenko et al., 2017*; *Malmsten, 1983*; *Nathan et al., 1990*; *Ocklenburg et al., 2017*; *Zhang et al., 2020*). Three-quarters of cervical spinal cords are asymmetric with a larger right side (*Nathan et al., 1990*). Spinal-muscular systems are asymmetric in human fetuses, and the asymmetry correlates with lateralized gene transcription (*de Kovel et al., 2017*; *Ocklenburg et al., 2017*). Mono- and polysynaptic segmental reflexes evoked by stimulation of the dorsal roots and recorded in the ventral roots in intact rats and cats display higher activity on the right side (*Hultborn and Malmsten, 1983a*; *Hultborn and Malmsten, 1983b*; *Malmsten, 1983*). Similarly, EMG recordings of hindlimb withdrawal reflexes evoked by electrical stimulation in control rats display higher activity on the right side (*Zhang et al., 2020*). On this asymmetric background, neural circuits controlling the left and right limbs may be differently regulated by opioid peptides and Arg-vasopressin (*Bakalkin et al., 1981*; *Bakalkin et al., 1986*; *Chazov et al., 1981*; *Klement'ev et al., 1986*). The left-right side-specific neurohormonal effects may be mediated through lateralized receptors. In the rat spinal cord, the expression of three opioid receptors is lateralized to the left, and their proportions and co-expression patterns are different between the left and right sides (*Kononenko et al., 2017*; *Watanabe et al., 2021*). The asymmetry may be a critical feature of the spinal cord allowing translation of the 'non-directional' hormonal messages into the left-right side-specific response.

## Limitations

The side-specific endocrine signaling was revealed in anaesthetized animals with transected spinal cords that were studied up to 180 min post UBI. Its biological and pathophysiological relevance has not been determined. Pathways from the injured brain area to the hypothalamic-pituitary system, neurohormones mediating effects of the right side injury, peripheral or central targets for the left- and right-side-specific endocrine messengers, and afferent, central or efferent mechanisms of the asymmetry formation have not been investigated. The study did not analyze forelimb postural asymmetry. It was not induced by lesion of the hindlimb sensorimotor cortex (*Zhang et al., 2020*) but it did develop after injury of the forelimb area (unpublished data).

The strategy was to selectively disable the neural and endocrine mechanisms by surgical means. In this approach, dissection of neural pathways does not allow us to assess a contribution of each pathway to asymmetric postural and motor deficits in awake animals. On the contrary, analysis of the UBI effects in the hypophysectomized rats with intact spinal cords may uncover a function of the left – right side-specific endocrine signaling. Rats with intact (*Figure 1—figure supplement 1C,D*) and transected (*Figure 1—figure supplements 3* and *4*) spinal cords developed HL-PA during the first 30 min following UBI. It would be worthwhile to analyze in more details whether the time course

of development of asymmetry differs between these groups; and also to ascertain whether signals mediated by the neural and endocrine pathways are additive, synergistic, or even antagonistic with respect to each other during the initial impairment phase and the recovery period using naive animals and the hypophysectomized rats with intact spinal cords.

This study does not focus on clinical correlates and mechanisms of postural deficits. The withdrawal reflexes and hindlimb posture were studied as readouts of the UBI because they are regulated by neurohormones and may be analyzed after spinal cord transection. Furthermore, they are directed along the left-right axis, and, therefore, can reveal whether the endocrine system conveys the side-specific signals. At the same time, the HL-PA and withdrawal reflexes model several features of the brain injury-induced sensorimotor and postural deficits in humans. First, the changes induced by UBI have a contra-ipsilesional pattern. Second, the HL-PA is not dependent on the afferent input (*Zhang et al., 2020*) and in this regard it may be similar to 'spastic dystonia', a tonic muscle overactivity that contributes to 'hemiplegic posture' (*Gracies, 2005*; *Lorentzen et al., 2018*; *Sheean and McGuire, 2009*). Third, asymmetric exacerbated withdrawal reflexes that lead to flexor spasms in patients (*Bussel et al., 1989*; *Dietz et al., 2009*; *Lavrov et al., 2006*; *Schouenborg, 2002*) are similarly developed in rats.

TBI and stroke cause dysfunction of the hypothalamic–pituitary system and hypopituitarism manifested as changes in secretion of pituitary hormones (*Bondanelli et al., 2005*; *Emelifeonwu et al., 2020*; *Klose and Feldt-Rasmussen, 2018*; *Lillicrap et al., 2018*). Ischemic stroke activates the hypothalamus-pituitary-adrenal axis (*Anne et al., 2007*), while a unilateral ablation of sensorimotor cortex elevates the level of circulating ACTH and induces morphological changes in the pituitary corticotrophs that produce ACTH and β-endorphin (*Lavrnja et al., 2014*). Neurobiological mechanisms underlying these effects may be similar with those of the UBI-induced endocrine signaling described in this study. These mechanisms and anatomical pathways involved have not been identified. However, cortical projections to the hypothalamus that may potentially mediate effects of focal brain injury on secretion of pituitary hormones have been described (*Jeong et al., 2016*).

## Conclusion

This study describes the left-right side-specific endocrine mechanism that, in addition to descending neural tracts, may mediate asymmetric effects of a unilateral brain injury on hindlimb postural asymmetry and spinal reflexes (*Figure 5*). Identification of features and the proportion of asymmetric sensorimotor deficits transmitted by neurohormonal signals *vs.* those mediated by neural pathways may be essential for understanding of stroke and TBI mechanisms.

# Materials and methods

**Key resources table**

| Reagent type (species) or resource | Designation | Source or reference | Identifiers | Additional information |
|---|---|---|---|---|
| Strain, strain background (*Rattus norvegicus*) | Wistar Hannover rat, male | Charles River Laboratories, Spain | | |
| Strain, strain background (*Rattus norvegicus*) | Sprague Dawley rat, male | Taconic, Denmark and Charles River Laboratories, France | | |
| Chemical compound, drug | Isoflurane | Abbott Laboratories, Norway | NDC 0044-5260-03 | Anesthesia agent |
| Chemical compound, drug | Lidocaine hydrochloride | Merck Group, Germany | PHR1257 | Anesthetic |
| Chemical compound, drug | Paraformaldehyde | Sigma-Aldridge, USA | Cat#: 158127 | Perfusion |
| Chemical compound, drug | SSR-149415 | Tocris Bioscience, United Kingdom | Cat#: 6195 | vasopressin V1B antagonist |
| Chemical compound, drug | Naloxone | Tocris Bioscience, United Kingdom | Cat#: 0599 | Opioid antagonist |

*Continued on next page*

*Continued*

| Reagent type (species) or resource | Designation | Source or reference | Identifiers | Additional information |
|---|---|---|---|---|
| Commercial assay or kit | Giemsa Stain | Sigma-Aldridge, USA | Cat#: 32884 | Nissl staining |
| Commercial assay or kit | RNasy Plus Mini kit | Qiagen, CA, USA | Cat#: 74136 | Total RNA extraction |
| Commercial assay or kit | iScript cDNA Synthesis Kit | Bio-Rad Laboratories, USA | Cat#: 1708891 | cDNA Synthesis |
| Commercial assay or kit | iTaq Universal Probes supermix | Bio-Rad Laboratories, USA | Cat# 1725131 | Real-Time PCR reagent |
| Peptide, recombinant protein | ß-Endorphin | Bachem, Switzerland | Cat# H-2814 | Neurohormone |
| Peptide, recombinant protein | Arg-vasopressin | Bachem, Switzerland | Cat# H-1780 | Neurohormone |
| Antibody | Anti-Dynorphin B (rabbit, polyclonal) | *Nguyen et al., 2005*; *Nylander et al., 1997*; *Yakovleva et al., 2006* | | RIA 1: 350000 |
| Antibody | Anti-Leu-enkephalin-Arg (rabbit, polyclonal) | *Nguyen et al., 2005*; *Nylander et al., 1997*; *Yakovleva et al., 2006* | | RIA 1: 35000 |
| Antibody | Anti-Met-enkephalin-Arg-Phe (rabbit, polyclonal) | *Nguyen et al., 2005*; *Nylander et al., 1997*; *Watanabe et al., 2015* | | RIA 1: 15750 |
| Sequence-based reagent | *Actb* (*Rattus norvegicus*) | Bio-Rad Laboratories, USA | qRnoCIP0050804 | PrimePCR Probe assay |
| Sequence-based reagent | *Arc* (*Rattus norvegicus*) | Bio-Rad Laboratories, USA | qRnoCEP0027389 | PrimePCR Probe assay |
| Sequence-based reagent | *Avpr1a* (*Rattus norvegicus*) | Bio-Rad Laboratories, USA | qRnoCEP0023750 | PrimePCR Probe assay |
| Sequence-based reagent | *Bdnf* (*Rattus norvegicus*) | Bio-Rad Laboratories, USA | qRnoCEP0026843 | PrimePCR Probe assay |
| Sequence-based reagent | *cFos* (*Rattus norvegicus*) | Bio-Rad Laboratories, USA | qRnoCEP0024078 | PrimePCR Probe assay |
| Sequence-based reagent | *Dlg4* (*Rattus norvegicus*) | Bio-Rad Laboratories, USA | qRnoCIP0026242 | PrimePCR Probe assay |
| Sequence-based reagent | *Egr1* (*Rattus norvegicus*) | Bio-Rad Laboratories, USA | qRnoCEP0022872 | PrimePCR Probe assay |
| Sequence-based reagent | *Gap43* (*Rattus norvegicus*) | Bio-Rad Laboratories, USA | qRnoCIP0027599 | PrimePCR Probe assay |
| Sequence-based reagent | *Gapgh* (*Rattus norvegicus*) | Bio-Rad Laboratories, USA | qRnoCIP0050838 | PrimePCR Probe assay |
| Sequence-based reagent | *GluR1* (*Rattus norvegicus*) | Bio-Rad Laboratories, USA | qRnoCIP0030725 | PrimePCR Probe assay |
| Sequence-based reagent | *Grin2a* (*Rattus norvegicus*) | Bio-Rad Laboratories, USA | qRnoCIP0025244 | PrimePCR Probe assay |
| Sequence-based reagent | *Grin2b* (*Rattus norvegicus*) | Bio-Rad Laboratories, USA | qRnoCIP0023973 | PrimePCR Probe assay |
| Sequence-based reagent | *Homer-1* (*Rattus norvegicus*) | Bio-Rad Laboratories, USA | qRnoCEP0023985 | PrimePCR Probe assay |
| Sequence-based reagent | *Oprd1* (*Rattus norvegicus*) | Bio-Rad Laboratories, USA | qRnoCEP0029668 | PrimePCR Probe assay |
| Sequence-based reagent | *Oprk1* (*Rattus norvegicus*) | Bio-Rad Laboratories, USA | qRnoCIP0029310 | PrimePCR Probe assay |
| Sequence-based reagent | *Oprm1* (*Rattus norvegicus*) | Bio-Rad Laboratories, USA | qRnoCEP0024902 | PrimePCR Probe assay |

*Continued*

| Reagent type (species) or resource | Designation | Source or reference | Identifiers | Additional information |
|---|---|---|---|---|
| Sequence-based reagent | *Pdyn* (*Rattus norvegicus*) | Bio-Rad Laboratories, USA | qRnoCEP0025357 | PrimePCR Probe assay |
| Sequence-based reagent | *Penk* (*Rattus norvegicus*) | Bio-Rad Laboratories, USA | qRnoCEP0029455 | PrimePCR Probe assay |
| Sequence-based reagent | *Pcsk6* (*Rattus norvegicus*) | Bio-Rad Laboratories, USA | qRnoCIP0045340 | PrimePCR Probe assay |
| Sequence-based reagent | *Nfkbia* (*Rattus norvegicus*) | Bio-Rad Laboratories, USA | qRnoCEP0026759 | PrimePCR Probe assay |
| Sequence-based reagent | *Syt4* (*Rattus norvegicus*) | Bio-Rad Laboratories, USA | qRnoCIP0029728 | PrimePCR Probe assay |
| Sequence-based reagent | *Tgfb1* (*Rattus norvegicus*) | Bio-Rad Laboratories, USA | qRnoCIP0031022 | PrimePCR Probe assay |
| Software, algorithm | Spike 2 | CED, UK | | EMG recording |
| Software, algorithm | Offline Sorter | Plexon, USA | version 3 | |
| Software, algorithm | NeuroExplorer | Nex Technologies, USA | | |
| Software, algorithm | Stan | *Carpenter et al., 2017* | v 2.19.2 | Scaling the data |
| Software, algorithm | R program | *R Development Core Team, 2018* | v 3.6.1 | |
| Software, algorithm | *Brms* | *Burkner, 2017* | v 2.9.6 | Interface for R |
| Software, algorithm | *Emmeans* | *Searle et al., 2012* | v 1.4 | R package |
| Other | TissuDura | Baxter, Germany | Cat#: 0600096 | Covering material |

## Animals

Male Wistar Hannover rats (Charles River Laboratories, Spain), with body weight of 150–200 g were used in behavioral, HL-PA and molecular experiments (*Figures 1*, *3* and *4B–E*; *Figure 1—figure supplement 1*, *3*, *4F-I*; *Figure 3—figure supplement 1*, *2*, *3*; *Figure 4—figure supplement 1G–O*, *2*). Male Sprague Dawley rats were used for analysis of HL-PA (*Figure 1—figure supplements 2*, *4B–E* and *5*), in electrophysiological experiments (*Figure 2*, *Figure 2—figure supplement 1*, *2*, *3*) (Taconic, Denmark; 150–400 g body weight) and for hypophysectomy (*Figure 4A* and *Figure 4—figure supplement 1B–E*) (Charles River Laboratories, France; 115–125 g body weight). The animals received food and water ad libitum and were kept in a 12 hr day-night cycle at a constant environmental temperature of 21°C (humidity: 65%). Approval for animal experiments was obtained from the Malmö/Lund ethical committee on animal experiments (No.: M7-16), and the ethical committee of the Faculty of Medicine of Porto University and Portuguese Direção-Geral de Alimentação e Veterinária (No. 0421/000/000/2018).

## Surgery

The animals were anesthetized with sodium pentobarbital (I.P.; 60 mg/kg body weight, as an initial dose and then 6 mg/kg every hour). If needed, the anesthesia was reinforced with ≈1.5% isoflurane (IsoFlo, Abbott Laboratories, Norway) in a mixture of 65% nitrous oxide-35% oxygen. Core temperature of the animals was controlled using a feedback-regulated heating system. In the experiments involving electrophysiological recordings, the rats were ventilated artificially via a tracheal cannula and the expiratory $CO_2$ and mean arterial blood pressure (65–140 mmHg) was monitored continuously in the right carotid artery.

## Aspiration brain injury and spinal cord transection

The experimental design included rats with either the UBI alone or the UBI which was preceded by a complete spinal cord transection. In the UBI-only experiments, anesthetized rats were placed on a surgery platform with stereotaxic head holder. The rat head was fixed in a position in which the bregma and lambda were located at the same horizontal level. After local injection of lidocaine (Xylocaine, 3.5 mg/ml) with adrenaline (2.2 µg/ml), the scalp was cut open and a piece of the parietal

bone located 0.5–4.0 mm posterior to the bregma and 1.8–3.8 mm lateral to the midline (*Paxinos and Watson, 2007*) was removed. The part of the cerebral cortex located below the opening that includes the hind-limb representation area of the sensorimotor cortex (HL-SMC) was aspirated with a metallic pipette (tip diameter 0.5 mm) connected to an electrical suction machine (Craft Duo-Vec Suction unit, Rocket Medical Plc, UK). Care was taken to avoid damaging the white matter below the cortex. After the ablation, bleeding was stopped with a piece of Spongostone and the bone opening was covered with a piece of TissuDura (Baxter, Germany). For sham operations, animals underwent the same anesthesia and surgical procedures, but the cortex was not ablated.

In the experiments in which UBI was preceded by the spinal cord transection, the anaesthetized animals were first placed on a surgery platform and the skin of the back was incised along the midline at the level of the superior thoracic vertebrae. After the back muscles were retracted to the sides, a laminectomy was performed at the T2 and T3 vertebrae. The spinal cord between the two vertebrae then was completely transected using a pair of fine scissors. A piece of Spongostan (Medispon MDD sp. zo.o., Toruń, Poland) was placed between the rostral and caudal stumps of the spinal cord. The completeness of the transection was confirmed by (i) inspecting the cord during the operation to ensure that no spared fibers bridged the transection site and that the rostral and caudal stumps of the spinal cord were completely retracted; and (ii) examining the spinal cord in all animals after termination of the experiment. To further ensure the completeness of transection, a 3–4 mm spinal cord segment was dissected and removed after laminectomy at the T2-T4 level in a subset of rats. Inspection under the microscope demonstrated that the transection was complete. Following the surgery, the rats were mounted onto the stereotaxic frame and the UBI was performed as described above. After completion of all surgical procedures, the wounds were closed with the 3–0 suture (AgnTho's, Sweden) and the rat was kept under an infrared radiation lamp to maintain body temperature during monitoring of postural asymmetry (up to 3 hr) and during EMG recordings.

To verify the UBI site the rats were perfused with 4% paraformaldehyde and the brains were removed from the skulls. Following post fixation overnight in the same fixative, the brains were soaked in phosphate-buffered saline for 2 days, dissected into blocks and the blocks containing the lesion area were cut into 50 μm sections using a freezing microtome. Every fourth section was mounted on slides and stained for Nissl with modified Giemsa solution (Sigma-Aldridge, USA; 1:5 dilution). The left drawing in *Figure 1A* shows the location of the right hindlimb representation area on the rat brain surface (adapted from *Frost et al., 2013*).

## Hypophysectomy

The hypophysectomy was performed at the Charles River Laboratories (France) site and all the surgery-related procedures, including postoperative care and transportation of animals, were performed according to the ethical recommendations of that company. The procedure for transauricular hypophysectomy, performed under isoflurane anesthesia, was described elsewhere (*Koyama, 1962*). Briefly, a hypodermic needle fitted to a plastic syringe was introduced into the external acoustic meatus until its tip reached the medial wall of the tympanic cavity. The needle was then pushed slightly further, so that its tip pierced the bone and entered the pituitary capsule. The hypophysis was then sucked into the syringe. The success of hypophysectomy was assessed by visual inspection of the hypophyseal region of the skull under a microscope following animal sacrifice and removal of the brain. Only data obtained in rats in which complete hypophysectomy was confirmed were included in the analysis. Sham-operated rats underwent an identical procedure except that the needle was not introduced into the pituitary capsule. Following the hypophysectomy, the animals were given a 3-week recovery period before initiating the UBI experiments.

## Behavioral tests

Experiments were performed between 10:00 and 15:00 hr over the course of the week preceding surgeries (pre-training) and one-day post-surgery (testing).

### Beam-walking test (BWT)(*Feeney et al., 1982*)

The BWT apparatus consisted of a horizontally placed wooden beam, 130 cm long and 1.4 cm wide, which was elevated 55 cm above the surface of a table. One end of the beam was free, while another was connected to a square platform (10 x 10 cm, with the floor and two sidewalls painted in

black) leading to the rat's home cage. The rats were trained to walk along the beam from its free end to their home cages. Training continued during two consecutive days preceding the brain surgery, with three daily sessions and six trials per session. On the first, second, and third trials, the rat was placed on the beam close to the platform, at the midpoint, and at the starting point (free end of the beam), respectively. The rat was considered trained if it performed the task within 80 s on the second training day. On the day following the UBI/Sham surgery, rats were given one session consisting of three trials (runs). Each run was video-recorded for further offline analysis by an observer blind to the treatment groups. The number of times the left and the right hindlimbs slipped off the beam were recorded and averaged across all runs of a given session.

### Ladder Rung Walking Test (LRWT) (*Metz and Whishaw, 2002*)

The horizontal LRWT apparatus, 100 cm long × 20 cm high, consisted of sidewalls made of clear Plexiglas and metal grid floor. The width of the apparatus was approximately 12 cm. The floor was composed of removable stainless steel bars (rungs), 3 mm in diameter, spaced 1 cm apart (center-to-center). The ladder was placed 30 cm above the surface of a table and was connected to the animal's home cage at one of its ends. Its opposite end was open and served as a starting point. On each trial, the rat was placed on the starting point and allowed to cross the ladder to enter the home cage. The width of the apparatus was adjusted to the size of the animal in order to prevent the animal from running in the reverse direction. During training (one session consisting of five trials), every second bar was removed, so that the rungs were spaced regularly at 2 cm intervals. During post-surgery testing (one session of five trials), the rung spacing pattern was modified in order to increase the difficulty of the task. Five distinct irregular spacing patterns were implemented in the testing session: 001101, 011010100, 1010011100, 1000011010, and 10001011000, where one denotes a rung and 0 a missing rung. However, the rung spacing patterns and the order of their presentation were the same for all rats. Each ladder run was video-recorded for further offline analysis by an observer blind to the surgery type. A total number of steps made by the left and right hindlimbs during each run and the number of times the limbs slipped between the rungs were recorded. The limb slips/total steps ratio was averaged across the five testing trials and was used as an error score.

## Analysis of postural asymmetry by the hands-on and hands-off methods

Analysis of the HL-PA by the hands-on method was described previously (*Bakalkin and Kobylyansky, 1989*; *Zhang et al., 2020*). Briefly, the postural asymmetry measurement was performed under pentobarbital (60 mg/kg, I.P.) anesthesia, or isoflurane anesthesia when rats with UBI were analyzed one or more days after the surgery. The level of anesthesia was characterized by a barely perceptible corneal reflex and a lack of overall muscle tone. The rat was placed in the prone position on 1 mm grid paper, and the hip and knee joints were straightened by gently pulling the hindlimbs back for 5–10 mm to reach the same level. Then, the hindlimbs were released and the magnitude of postural asymmetry was measured in millimeters as the length of the projection of the line connecting symmetric hindlimb distal points (digits 2–4) on the longitudinal axis of the rat. The procedure was repeated six times in immediate succession, and the mean HL-PA value for a given rat was calculated and used in statistical analyses. The measurements were performed 0.5, 1, and 3 hr after the brain injury, or at other time points as shown on figures. In a separate group of rats (*Figure 1—figure supplement 1C,D*), HL-PA was assessed 1, 4, 7, and 14 days after the UBI or sham surgery under isoflurane anesthesia (1.5% isoflurane in a mixture of 65% nitrous oxide and 35% oxygen). The rat was regarded as asymmetric if the magnitude of HL-PA exceeded the 1 mm threshold (see statistical section). The limb displacing a shorter projection was considered to be flexed.

In a subset of the rats with UBI or sham surgery (n = 11 and 10, respectively), the hindlimbs were stretched by gently pulling two threads glued to the nails of the middle three toes of both legs. In another subset (n = 6), the skin of the hindlimbs including and distal to the ankle joints was fully anesthetized by a topical application of 5% lidocaine cream 10 min before the assessments of HL-PA in rats with UBI. The absence of the pedal withdrawal reflexes following lidocaine application was confirmed in awake rats by pinching the skin between the toes with blunt forceps. None of these two procedures affected the resulting HL-PA suggesting that HL-PA formation does not dependent on tactile input from the hind paw.

For analysis in the supine position, the rat was placed in a V-shaped trough, a 90° - angled frame located on a leveled table surface with the 1 mm grid sheet; otherwise, the procedure was the same as for the prone position. The HL-PA values and the probability to develop asymmetry ($P_A$) were essentially the same for both positions.

In the hands-off analysis (*Figure 1—figure supplement 2*), the anesthetized rat was placed on the bench in prone position. Silk threads were glued to the nails of the middle three toes of both hindlimbs, and their other ends were tied to hooks attached to the movable platform that was operated by a micromanipulator. To reduce potential friction between the hindlimbs and the surface with changes in their position during stretching and after releasing them, the bench under the rat was covered with plastic sheet and the movable platform was raised up to form a 10° angle between the threads and the bench surface. Stretching was initiated at the 'natural' hindlimb position that was either symmetric or asymmetric, and performed for the 2 cm distance at a rate of 2 cm/s (Variant 1, V1; *Figure 1—video 1*, episodes 1 and 2). Alternatively, the limbs were adjusted to an approximately symmetric position by gently pulling the thread on the flexed limb and then stretching it at a rate of 2 cm/sec for 1.5 cm (Variant 2, V2; *Figure 1—video 1*, episode 3). The threads then were relaxed, the limbs were released and the resulting HL-PA was photographed. The procedure was repeated six times in succession, and the mean value of postural asymmetry for a given rat was calculated and used in statistical analyses. Both variants 1 and 2 (V1 and V2) of the hands-off method, and the hands-on method produced virtually the same results; no differences (p > 0.40) in the magnitude and its direction were revealed between them (*Figure 1—figure supplement 2*).

The postural asymmetry analysis was blind to the observer excluding the analysis combined with the EMG. The 'reverse design' HL-PA results shown on *Figure 1F* were replicated by two groups in different laboratories (*Figure 1—figure supplement 4B–E and F–L*, respectively) and by both the hands-on and hands-off methods (*Figure 1—figure supplement 5*).

The HL-PA was measured in mm with negative and positive HL-PA values that are assigned to rats with the left and right hindlimb flexion, respectively. This measure shows the flexion side and HL-PA value. However, it does not show the proportion of the animals with asymmetry in each group; we could not see whether all or a small fraction of animals display the asymmetry. Furthermore, its interpretation may not be straightforward for groups with the similar number of left or right flexion; in this case, the HL-PA value would be about zero. Data are also presented as the probability of postural asymmetry (PA) that shows the proportion of animals exhibiting HL-PA at the imposed threshold (> 1 mm). The PA does not show flexion side and flexion size. These two measures are obviously dependent; however, they are not redundant and for this reason, both are required for data presentation and characterization of the HL-PA.

## EMG experiments

### Electromyography recordings

Core temperature was maintained between 36°C and 38°C using a thermostatically controlled, feedback-regulated heating system. The EMG activity of the extensor digitorum longus (EDL), interrossi (Int), peroneaus longus (PL), and semitendinosus (ST) muscles of both hindlimbs were recorded as described previously (*Schouenborg et al., 1992*; *Weng and Schouenborg, 1996*). EMG recordings were initiated approximately 3 hr after spinalization (i.e.; 2 hr and 20 min after UBI). Recordings were performed using gauge stainless steel disposable monopolar electrodes (DTM-1.00F, The Electrode Store, USA). The electrodes were insulated with Teflon except for ~200 μm at the tip. The impedance of the electrodes was from 200 to 1000 kΩ. For EMG recordings, a small opening was made in the skin overlying the muscle, and the electrode was inserted into the mid-region of the muscle belly. A reference electrode was inserted subcutaneously in an adjacent skin region. The electrode position was checked by passing trains (100 Hz, 200 ms) of cathodal pulses (amplitude < 30 μA, duration 0.2 ms). The EMG signal was recorded with Spike two program (CED, UK) with a sampling rate of 5000 Hz. Low and high pass filter was set at 50,000 Hz and 500 respectively. Generally, the EMG activity of three or four pairs of hind limb muscles was recorded simultaneously in each experiment / rat.

## Cutaneous stimulation

Stimulation sites were decided according to each muscle's receptive field as described previously (*Schouenborg et al., 1992*; *Weng and Schouenborg, 1996*). After searching the muscle receptive field in responding to pinch stimulation, a pair of stimulation electrodes that were the same as the recording electrodes were inserted subcutaneously into the center of each muscle's receptive field. The same pairs of digits (i.e. 2, 3, 4, and 5 of both limbs) were stimulated to induce ipsilateral reflex responses (*Schouenborg et al., 1992*). To detect the stimulation intensity that induce the maximal reflex in each muscle, graded current pulses (1 ms, 0.1 Hz) were used ranging mostly from 1 to 30 mA, and occasionally up to 50 mA. The reflex threshold was defined as the lowest stimulation current intensity evoking a response at least in 3 out of 5 stimulations. If a muscle response was induced by stimulation at more than one site, the lowest current was taken as a threshold value. For EMG data collection, the current level that induced submaximal EMG responses from both legs, usually ranging from 5 to 20 mA, was chosen. This was usually two to three times higher than the threshold currents. The same current level was used on symmetrical points from the most sensitive area on both paws. For each site, EMG responses from 18 to 20 stimulations at 0.1 Hz frequency were collected. No visible damage of the skin or marked changes in response properties at the stimulation sites were detectable at these intensities.

## EMG data analysis

The spikes from Spike2 EMG data files were sorted with Offline Sorter (version 3, Plexon, USA). The EMG amplitude (spike number) from different muscles was calculated with NeuroExplorer (Nex Technologies, USA). To avoid stimulation artifacts, spikes from the first one or two stimulations were removed from further analysis. The number of spikes was calculated from 16 consecutive stimulations thereafter. The EMG thresholds and responses registered from 0.2 to 1.0 s were analyzed. In this animal preparation, characterized in our previous studies, the withdrawal reflexes evoked by innocuous stimulation were very weak compared to those evoked by noxious stimulation (*Schouenborg, 2002*; *Schouenborg et al., 1992*; *Weng and Schouenborg, 1998*; *Zhang et al., 2020*).

## Criteria for comparison of two hindlimbs

Strict criteria were applied to ensure data comparability between two hindlimbs for (i) the experimental procedures including the symmetry of stimulation and recording conditions between the two sides; (ii) application of electrodes with similar resistance for analysis of symmetric muscles; (iii) selection of the reflex features for the analysis; and (iv) statistical analysis. The core criteria were similar to those described by Hultborn and Malmsten (*Hultborn and Malmsten, 1983a*; *Hultborn and Malmsten, 1983b*; *Malmsten, 1983*). Pairs of stimulation and recording electrodes were positioned as symmetrically as possible. Stimulation electrodes were inserted into the center of the receptive fields of the left and right muscles (*Schouenborg et al., 1992*; *Weng and Schouenborg, 1996*), and recording electrodes into the middle portion of the muscle belly. This was performed by an experienced investigator. The same stimulation patterns were used for stimulation of pairs of digits to induce reflexes in symmetric muscles. The same threshold level was used for the left and right muscles. In general, the current level that was applied exceeded the higher threshold recorded for each pair of muscles at a given stimulation site by 2–3-fold. Data recorded with stimulation of more than one site (digits 2, 3, 4, and 5) were processed as replicates to decrease experimental error. Only ipsilateral responses were recorded. Only data for pairs of left and right muscles of the same animal were included in the analysis. The sample size was sufficiently large (n = 11 rats in sham, and n = 18 rats in UBI groups) to ensure statistical power sufficient for analysis of responses typically recorded in this model (*Schouenborg et al., 1992*; *Weng and Schouenborg, 1996*; *Zhang et al., 2020*).

To minimize inter-individual variations that may be caused by differences in physiological and experimental conditions, including depth of anesthesia, and circulatory and respiratory states, the asymmetry index calculated for each animal but not the absolute values of the reflex size (i.e. reflex amplitude, thresholds and the number of spikes), was analyzed. Comparison between the two sides using the asymmetry index was based on the assumption that one side was a reference for the other side in each animal, and this approach largely diminished a contribution of the inter-individual variations (*Hultborn and Malmsten, 1983a*; *Hultborn and Malmsten, 1983b*; *Malmsten, 1983*).

Analysis of the asymmetry index allowed double assessment, first, within the UBI and control (sham) groups that identified the asymmetric *vs.* symmetric pattern, and, second, between the animal groups that revealed UBI-induced changes in the asymmetry. Analysis of the asymmetry index in control group established whether the observed distribution was close to the expected symmetric pattern (the size of variations around the symmetry point was assessed), and, therefore, demonstrated the validity of the approach.

Because multiple responses were measured for the same animal, including two limbs, four muscles, and differing stimulation conditions, and because they were analyzed within animal groups and between the groups, we applied mixed-effects models using Bayesian inference. To avoid bias in the acquisition of experimental observations that may be imposed by intermediate data analyses, the data processing and statistical analysis were performed after the completion of experiments. Only strong and significant UBI effects (in the study they were from 2.8- to 54-fold) were considered as biologically relevant. At the same time, the background reflex asymmetry in the control group that had a smaller effect size (1.7-fold difference from the symmetry), in its magnitude and direction was in an agreement with results that we published previously (*Zhang et al., 2020*) and that were published by other groups (*Hultborn and Malmsten, 1983a*; *Hultborn and Malmsten, 1983b*; *Malmsten, 1983*).

## Effects of serum, neurohormones, and antagonists of opioid (naloxone) and vasopressin V1B (SSR-149415) receptors on HL-PA development

Serum was collected from three animals in each group of rats with transected spinal cord 3 hr after the UBI or sham surgery, pooled, kept at −80°C until use, and administered intravenously (0.3 mL / rat) to rats under pentobarbital anesthesia 10 min after complete spinal cord transection.

Serum and Arg-vasopressin were administered into the cisterna magna (intracisternal route; five microliters/rat) (*Ramos et al., 2019*; *Xavier et al., 2018*) of intact rats under pentobarbital anesthesia, which was followed by the spinal cord transection 10 min later. HL-PA was analyzed at the 0.5, 1 and 3 hr time points after injection while the animals remained under pentobarbital anesthesia.

SSR-149415 (5 mg/ml/kg, n=12) dissolved in a mixture of DMSO (10%) and saline (90%), or vehicle alone (n = 8) was administered I.P. 10 min before spinal cord transection. This was followed by either the left-side UBI (SSR-149415: n = 6; vehicle: n = 5) or intravenous administration of serum from UBI rats (SSR-149415: n = 6; vehicle: n = 3). In rodents, the effects of SSR149415 develop within 0.5–1 hr and last for 4–6 hr after administration (*Ramos et al., 2006*; *Serradeil-Le Gal et al., 2002*). Naloxone (5 mg/ml/kg in saline) or saline alone was injected I.P. 2 hr after delivering the UBI (naloxone: n = 6; saline: n = 6) or after injecting the UBI serum (naloxone: n = 6; saline: n = 3) to rats with transected spinal cord.

## Analysis of mRNA levels by quantitative RT-PCR (qRT-PCR)

Total RNA was purified by using RNasy Plus Mini kit (Qiagen, Valencia, CA, USA). RNA concentrations were measured with Nanodrop (Nanodrop Technologies, Wilmington, DE, USA). RNA (1 μg) was reverse-transcribed to cDNA with the iScript cDNA Synthesis Kit (Bio-Rad Laboratories, CA, USA) according to manufacturer's protocol. cDNA samples were aliquoted and stored at –20°C. TagMan assay in 384-well format was applied. cDNAs were mixed with PrimePCR Probe assay and iTaq Universal Probes supermix (Bio-Rad) for qPCR with a CFX384 Touch Real-Time PCR Detection System (Bio-Rad Laboratories, CA, USA) according to manufacturer's instructions. TagMan probes used are listed in Key resources table.

All procedures were conducted strictly in accordance with the established guidelines for the qRCR based analysis of gene expression, consistent with the minimum information for publication of quantitative real-time PCR experiments guidelines (MIQE) (*Bustin et al., 2009*; *Taylor et al., 2019*). The raw qRT-qPCR data were obtained by the CFX Maestro Software for CFX384 Touch Real-Time PCR Detection System (Bio-Rad Laboratories, CA, USA). mRNA levels of genes of interest were normalized to the geometric mean of expression levels of two reference genes *Actb* and *Gapdh* selected out of 10 genes (*Actb, B2m, Gapdh, Gusb, Hprt, Pgk, Ppia, Rplpo13a, Tbp,* and *Tfrc*) using the geNorm program [https://genorm.cmgg.be/ and *Kononenko et al., 2017*; *Vandesompele et al., 2002*]. The expression stability of candidate reference genes was computed for four sets of samples that were the left and right halves of the lumbar spinal cord obtained from

the left-sided sham surgery group and the left-sided UBI group and was as follows (from high to low): *Actb, Gapdh, Tbp, Rplpo13a, Hprt, Pgk, B2m, Tfrc, Ppia,* and *Gusb*. In each experiment, the internal reference gene-stability measure M did not exceed 0.5 at the threshold value imposed by the MIQE equal to 1.5. The number of reference genes was optimized using the pairwise stability measure (V value) calculated by the geNorm program. The V value for *Actb* and *Gapdh*, the top reference genes was 0.12 that did not exceed the 0.15 threshold demonstrating that analysis of these two genes is sufficient for normalization.

## Neuroplasticity-related and neurohormonal genes

*Arc*, activity-regulated cytoskeletal gene implicated in numerous plasticity paradigms; *Bdnf*, brain-derived neurotrophic factor regulating synaptogenesis; *cFos*, a neuronal activity dependent transcription factor; *Dlg4* gene codes for PSD95 involved in AMPA receptor-mediated synaptic plasticity and post NMDA receptor activation events; *Egr1* regulating transcription of growth factors, DNA damage, and ischemia genes; *Gap-43* coding for growth-associated protein Gap-43 that regulates axonal growth and neural network formation; *GluR1* and *Grin2b* coding for the glutamate ionotropic receptor AMPA Type Subunit one and NMDA receptor subunit, respectively, both involved in glutamate signaling and synaptic plasticity; *Grin2a* subunit of the glutamate receptors that regulates formation of neural circuits and their plasticity; *Homer-1* giving rise to Homer Scaffold Protein 1, a component of glutamate signaling involved in nociceptive plasticity; *Nfkbia* (I-Kappa-B-Alpha) that inhibits NF-kappa-B/REL complexes regulating activity-dependent inhibitory and excitatory neuronal function; *Syt4* (Synaptotagmin 4) playing a role in dendrite formation and synaptic growth and plasticity; and *Tgfb1* that gives rise to Transforming Growth Factor β1 regulating inflammation, expression of neuropeptides and glutamate neurotoxicity, were selected as representatives of neuroplasticity-related genes (*Adkins et al., 2006*; *Anderson and Winterson, 1995*; *Buisson et al., 2003*; *Dolan et al., 2011*; *Epstein and Finkbeiner, 2018*; *Grasselli and Strata, 2013*; *Harris et al., 2016*; *Hayashi et al., 2000*; *Joynes et al., 2004*; *Larsson and Broman, 2008*; *O'Mahony et al., 2006*; *Santibañez et al., 2011*; *Tappe et al., 2006*; *Vavrek et al., 2006*; *Won et al., 2016*; *You et al., 2004*).

The prodynorphin (*Pdyn*) and proenkephalin (*Penk*) genes, and genes coding for opioid δ (*Oprd1*), κ (*Oprk1*) and μ (*Oprm1*) receptors, and arginine vasopressin receptor 1A (*Avpr1a*) were analyzed. Proopiomelanocortin (*Pomc*), arginine vasopressin receptor 1B (*Avpr1B*), arginine vasopressin receptor 2 (*Avpr2*), and arginine vasopressin (*Avp*) genes were found to be expressed at low levels in the spinal cord and were not included in the analysis.

## Radioimmunoassay (RIA)

The procedure was described elsewhere (*Christensson-Nylander et al., 1985*; *Merg et al., 2006*). Briefly, 1 M hot acetic acid was added to finely powdered frozen tissues, and samples were boiled for 5 min, ultrasonicated, and centrifuged. Tissue extract was run through a SP-Sephadex ion exchange C-25 column, and peptides were eluted and analyzed by RIA. Anti-Dynorphin B antiserum showed 100% molar cross-reactivity with big dynorphin, 0.8% molar cross-reactivity with Leu-morphine (29 amino acid C-terminally extended Dynorphin B), and <0.1% molar cross-reactivity with Dynorphin A (1–17), Dynorphin A (1–8), α-neoendorphin, and Leu-enkephalin (*Yakovleva et al., 2006*). Cross-reactivity of Leu-enkephalin-Arg antiserum with Dynorphin B and Leu- and Met-enkephalin was <0.1% molar, with α-neoendorphin 0.5% molar, with Dynorphin A (1–8) 0.7% molar, with Met-enkephalin-Arg-Phe 1% molar and with Met-enkephalin-Arg 10% molar. Cross-reactivity of Met-enkephalin-Arg-Phe antiserum with Met-enkephalin, Met-enkephalin-Arg, Met-enkephalin-Arg-Gly-Leu, Leu-enkephalin and Leu-enkephalin-Arg was <0.1% molar (*Nylander et al., 1997*). Our variant of RIA readily detected Dynorphin B and Leu-enkephalin-Arg in wild-type mice (*Nguyen et al., 2005*) but not in *Pdyn* knockout mice; thus the assay was highly specific and not sensitive to the presence of contaminants. The peptide content is presented in fmol/mg tissue.

## Statistical analysis

Processing and statistical analysis of the HL-PA, EMG and molecular data was performed after completion of the experiments by the statisticians (DS and VG), who were not involved in execution of

experiments. Therefore, the results of intermediate statistical analyses could not affect acquisition of experimental data that otherwise might be biased.

## Postural asymmetry and withdrawal reflexes

Predictors and outcomes were centered and scaled before Bayesian Regression Models were fitted by calling Stan 2.19.2 (*Carpenter et al., 2017*) from *R* 3.6.1 (*R Development Core Team, 2018*) using *brms* 2.9.6 (*Burkner, 2017*) interface. To reduce the influence of outliers, models used Student's *t* response distribution with identity link function. Models had no intercepts with indexing approach to predictors (*McElreath, 2019*). Default priors were provided by *brms* according to Stan recommendations (*Gelman, 2019*). Residual SD and group-level SD were given weakly informative prior student_t(3, 0, 10). An additional parameter ν of Student's distribution representing the degrees of freedom was given wide gamma prior gamma(2, 0.1). Group-level effects were given generic weakly informative prior normal(0, 1). Four Markov chain Monte Carlo (MCMC) chains of 40,000 iterations were simulated for each model, with a warm-up of 20,000 runs to ensure that effective sample size for each estimated parameter exceeded 10,000 (*Kruschke, 2015*), producing stable estimates of 95% highest posterior density continuous intervals (HPDCI). MCMC diagnostics were performed according to Stan manual.

The contrast in HL-PA between the groups [designated as $\Delta$HL-PA$_{(group\ 1\ -\ group\ 2)}$ on the figures] was a simple pairwise main effect (contrast), that is the difference between estimated marginal means of the groups computed by the R package emmeans (see https://cran.r-project.org/web/packages/emmeans/vignettes/comparisons.html#pairwise) given the fitted Bayesian model.

Median of the posterior distribution, 95% HPDCI and adjusted P-values were reported as computed by R package *emmeans* 1.4 (*Searle et al., 2012*). Adjusted P-values were produced by frequentist summary in *emmeans* using the multivariate t distribution with the same covariance structure as the estimates. The asymmetry and contrast between groups were defined as significant if the corresponding 95% HPDCI interval did not include zero and, simultaneously, the adjusted p-value was $\leq 0.05$. R scripts are available upon request.

The 99th percentile of the HL-PA magnitude in rats after sham surgery (n = 36) combined at the 2 or 3 hr time points was 1.1 mm. Therefore, the rats with HL-PA magnitude > 1 mm threshold were defined as asymmetric. The probability $P_A$ to develop HL-PA above 1 mm in magnitude was modelled with Bernoulli response distribution and logit link function. The UBI effects remained significant for models with thresholds of 2 or 3 mm.

In the EMG analysis, asymmetry in stimulation threshold (Thr) and a spike number (SN) in EMG responses for each pair of hindlimb muscles analyzed was assessed using the Contra-/Ipsilesional asymmetry indexes $AI_{Thr}$ ($AI_{Thr}$ = log$_2$ [Thr$_{contra}$ / Thr$_{ipsi}$]) and $AI_{SN}$ ($AI_{SN}$ = log$_2$ [(1+SN$_{contra}$)/(1 +SN$_{ipsi}$)]), where *contra* and *ipsi* designate the Contralateral and Ipsilesional sides relative to the brain injury side, or the Left-/Right asymmetry indexes. *Operation type* (UBI *vs.* sham) was the factor of interest analyzed for each muscle (EDL stimulated at D3, D4 and D5; Int at D2, D3, D4, and D5; PL at D4 and D5; and ST at the heel). Data recorded at stimulation of more than one site were processed as replicates for a given muscle. The number of rats for each pair of muscles in each UBI and sham group is shown in *Figure 2—figure supplement 2*.

The $AI_{SN}$ was fitted using linear multilevel model with fixed effects of *muscle* (EDL, Int, PL and ST) interacting with *operation type* (UBI vs. sham) and log-transformed *recording current*. The *sweep's number* was a fixed effect confounder with non-significant effect, showing that the $AI_{SN}$ was not significantly affected by wind-up. The $AI_{Thr}$ was fitted using the similar linear multilevel model without the *recording current* and *sweep's number* factors. To get rid of No-U-Turn Sampler warnings, parameters adapt_delta=0.999 and max_treedepth=13 were used.

## Gene expression and opioid peptide analyses

First, the mRNA levels of 14 neuroplasticity-related genes (*Arc*, *Bdnf*, *cFos*, *Dlg4*, *Egr1*, *Homer-1*, *Gap43*, *GluR1*, *Grin2a*, *Grin2b*, *Nfkbia*, *Pcsk6*, *Syt4*, and *Tgfb1*), and six opioid and vasopressin genes (*Penk*, *Pdyn*, *Oprm1*, *Oprd1*, *Oprk1*, and *Avpr1a*), along with the levels of opioid peptides Dynorphin B, Leu-enkephalin-Arg, and Met-enkephalin-Arg-Phe were compared separately for the left and right halves of the lumbar spinal cord between the left UBI (n = 12) and left sham (n = 11) rat groups. Only *Avpr1a* out of four genes of the vasopressin system (*Avpr1a*, *Avpr1b*, *Avpr2*, and

*Avp*) was found to be expressed in the lumbar spinal cord and therefore included in the statistical analyses. The mRNA and peptide levels were compared between the animal groups for the left and right spinal halves separately using Mann-Whitney test followed by Bonferroni correction for a number of tests (correction factor for mRNAs was 40, for peptides 6). Results were considered significant if the p value corrected for multiple comparisons ($P_{adjusted}$) did not exceed 0.05. Log fold change (logFC) was computed as a difference of median $\log_2$-scaled expression values.

Second, the expression asymmetry index (AI) defined as $\log_2$-scaled ratio of expression levels in Contra and Ipsilesional spinal halves ($\log_2$[Contra/Ipsi]), was compared between the rat groups. Comparison of AI was performed using a Mann-Whitney test followed by a Bonferroni correction for multiple tests (correction factor was 20).

Heatmaps of expression levels and AI were constructed using data (0,1)-standardized for each gene by subtraction of the median value and division by an inter-quartile range. In the analysis of expression levels, standardization was applied to $\log_2$-scaled expression levels pooled for the left and the right spinal halves.

### Intra- and inter-regional gene coexpression patterns

Spearman's rank correlation coefficient was calculated for all gene pairs in each area (n = 190) and between the areas (n = 400). To circumvent the effects of differences in a number of animals between the groups (caused by differences in the number of rats or by missing values) on outcome of statistical analyses, the following procedure was applied. For a given pair of genes, animals with missing expression levels were excluded from calculations. For the group with smaller number of remaining animals (let *N* denote this number), the correlation coefficient was calculated in a standard way. For the other group, the correlation coefficient was calculated for all subsets consisting of *N* animals, and the median was taken. The procedure was applied separately for each pair of genes in each analysis. Significance of correlation coefficients was assessed using Spearman R package with precomputed null distribution (i.e. *approximation* parameter set to '*exact*').

Statistical comparison of gene-gene coordination strength between the animal groups was performed by applying Wilcoxon signed-rank test to the set of absolute values of all correlation coefficients and, separately, to the set of absolute values of significant correlation coefficients (i.e. having associated p-value not exceeding 0.05 for at least one animal group). As the comparison of gene-gene coordination strength ignored correlation signs, a separate analysis was performed to assess differences in the proportion of positive and negative correlations between animal groups. This assessment was performed separately for the sets of (i) all and (ii) significant correlation coefficients using the Fisher's Exact test with two×two contingency table.

## Acknowledgements

This paper is dedicated to Professors Boris I Klement'ev and Genrich A Vartanian for their outstanding postural asymmetry studies, which are mostly unknown to the Western scientific community and which set the stage for the present investigation. We are grateful to Dr. Michael Ossipov, research professor emeritus at the University of Arizona College of Medicine for discussion and manuscript editing processing, Dr. Aleh Yahorau for help with biochemical assays and figure preparation, Dr. Jonas Thelin for help with electrophysiology experiments and discussion, and Dr. Gisela Maia for technical support. The study was supported by the Swedish Science Research Council (Grants K2014-62X-12190-19-5 and 2019-01771-3), PO Zetterling foundation, Uppsala University, and grants of the Government of the Russian Federation (14 .W03.31.0031) and the Russian Scientific Foundation (17-14-01338).

## Additional information

### Competing interests

Vladimir Galatenko: Vladimir Galatenko is affiliated with Evotec International GmbH. The author has no other competing interests to declare. The other authors declare that no competing interests exist.

## Funding

| Funder | Grant reference number | Author |
| --- | --- | --- |
| Vetenskapsrådet | K2014-62X-12190-19-5 | Georgy Bakalkin |
| Vetenskapsrådet | 2019-01771-3 | Georgy Bakalkin |
| Uppsala Universitet | | Georgy Bakalkin |
| Vetenskapsrådet | 2016-06195 | Jens Schouenborg |
| Skåne County Council's Research and Development Foundation | F2018/1490 | Jens Schouenborg |
| P.O. Zetterling Foundation | | Olga Kononenko |
| Government Council on Grants, Russian Federation | 14.W03.31.0031 | Vladimir Galatenko |
| Russian Science Foundation | 17-14-01338 | Alex Tonevitsky |

The funders had no role in study design, data collection and interpretation, or the decision to submit the work for publication.

## Author contributions

Nikolay Lukoyanov, Conceptualization, Data curation, Supervision, Investigation, Methodology, Writing - original draft, Writing - review and editing; Hiroyuki Watanabe, Data curation, Validation, Investigation, Methodology; Liliana S Carvalho, Validation, Investigation; Olga Kononenko, Data curation, Formal analysis, Investigation, Visualization; Daniil Sarkisyan, Data curation, Formal analysis, Validation, Writing - original draft; Mengliang Zhang, Formal analysis, Validation, Investigation, Methodology, Writing - review and editing; Marlene Storm Andersen, Elena A Lukoyanova, Igor Bazov, Tatiana Iakovleva, Investigation; Vladimir Galatenko, Data curation, Formal analysis, Methodology, Writing - original draft; Alex Tonevitsky, Resources, Data curation, Formal analysis; Jens Schouenborg, Conceptualization, Funding acquisition, Methodology, Writing - review and editing; Georgy Bakalkin, Conceptualization, Supervision, Funding acquisition, Writing - original draft, Project administration, Writing - review and editing

## Author ORCIDs

Olga Kononenko http://orcid.org/0000-0002-1332-7067
Igor Bazov http://orcid.org/0000-0003-4388-1656
Georgy Bakalkin https://orcid.org/0000-0002-8074-9833

## Ethics

Animal experimentation: Approval for animal experiments was obtained from the Malmö/Lund ethical committee on animal experiments (No.: M7-16), and the ethical committee of the Faculty of Medicine of Porto University and Portuguese Direção-Geral de Alimentação e Veterinária (No. 0421/000/000/2018).

## Decision letter and Author response

Decision letter https://doi.org/10.7554/eLife.65247.sa1
Author response https://doi.org/10.7554/eLife.65247.sa2

# Additional files

## Supplementary files

• Transparent reporting form

## Data availability

All data generated or analysed during this study are included in the manuscript and supporting files. Source data files have been provided for Figures 1–4.

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
