## [Decision Letter]

**Acceptance summary:**

This contribution is as important as it is unexpected. A wide variety of experiments makes a strong case for humoral signaling of forebrain damage to spinal circuits. The fundamental and clinical interest of these studies cannot be overestimated.

**Decision letter after peer review:**

[Editors’ note: the authors submitted for reconsideration following the decision after peer review. What follows is the decision letter after the first round of review.]

Thank you for submitting your work entitled "Endocrine signaling mediates asymmetric motor deficits after unilateral brain injury" for consideration by *eLife*. Your article has been reviewed by 4 peer reviewers, including X as the Reviewing Editor and Reviewer #1, and the evaluation has been overseen by a Senior Editor. The following individuals involved in review of your submission have agreed to reveal their identity: Simon M Danner (Reviewer #3); John H. Martin (Reviewer #4).

Our decision has been reached after consultation between the reviewers. Based on these discussions and the individual reviews below, we regret to inform you that your work will not be considered further for publication in *eLife*.

There was much about this manuscript that intrigued the reviewers. However, as the endpoint is an overthrow of more than a century old dogma, the bar of proof is raised. Thus, as editor, while I am appreciative of the opportunity to examine this interesting work, it is my opinion that this study does not meet the elevated bar needed.

The concerns are:

The lack of blinding.

Concerns about whether the transections were complete. Given the organization of the rodent spinal cord, even the sparing of a reticulospinal pathway is in the ventral column could account for cortical control of motor function in the distal spinal cord in a rodent.

Concern over papers from 30 years ago, from the same lab and never replicated, that form a critical piece of this story.

There is no information about the role of deafferentation. Is a background of deafferentation necessary?

There is no gold standard (UBI alone) for the reflex analysis.

Far more addressable but still worrisome is that the authors do not directly tackle the issue of the human correlate of postural asymmetry in the rodent.

*Reviewer #1:*

This is a study with the shocking message that several of the effects of a

somatomotor lesion on contralateral movement are mediated by humoral substances rather than by the loss of synaptic input. The experiments are thorough, showing that postural asymmetry results even if a thoracic transection is performed prior to the somatomotor cortical lesion. This does not happen for L brain injury after hypophysectomy but the effects of L brain injury can be recapitulated with vasopressin or b-endorphin. I have a few substantive comments.

The authors need to clearly state what they believe the postural asymmetry corresponds to in the human. Does this correspond to the hemiparetic stance, walk? Does it include the voluntary paralysis of the digits? I think it is easier to conceive of the former than the latter and when I first read the manuscript, I thought that was what the authors were postulating. However upon re-reading the introduction, it is less clear. The authors need to explicitly and clearly state their hypothesis in the introduction and support their interpretation on this point in the discussion.

I would like to see a comparison of the reflexes after L-UBI alone or L-UBI following thoracic transection. There is an effect in the dual manipulation but whether that mimics the L-UBI effect is not tested.

Secondly, as remarkable as the asymmetry in response to systemic peptides is the hindlimb preference (over the forelimb). One possibility for this is that there is a different set of molecules for the L hindlimb, R hl, L fl and R fl. Crazy-seeming but so is this result so at this point, the reader feels reluctant to rule anything out. A second possibility is that the deafferentation produced by the transection (or UBI in non-experimental conditions) is a necessary prerequisite for the humoral factors to work their magic. Related to this, do the authors have any data on whether VP, b-end do anything in the absence of a proceeding transection? These possibilities should be discussed.

The hyper- part of the reflexes is not analyzed, only the asymmetry. [actually there is mention of a decrease in threshold for the flexor reflexes but this is in the sum up and not in the figure or in the quantified text.] Why is this?

180-190 is written as though VP and b-endorphin are given together but the figure suggests they were given separately. Please make this clear. Did the authors try the two substances together? Same comments for the antagonists.

*Reviewer #2:*

This is an enormously interesting study. And, if its results are replicated, it is also an enormously important study. The belief that lateralized brain control of the somatic musculature is due entirely to lateralized neural pathways has been unquestioned truth for several thousand years. Other possibilities are seldom considered; when they are mentioned, they are quickly discounted (e.g., p. 568 in J Neurophysiol (1989) 61:563-576). This study indicates that this belief is not the whole story; that humoral factors also contribute to lateralized brain control. Given these fairly earth-shaking results (with really major scientific and clinical implications), the credibility of the study is critically important. In this regard, it is fortunate and impressive that the authors have addressed the issue in multiple ways – the basic postural study, the effects of specific agents, the effects of serum from other lesioned animals, the effects of hypophysectomy, the effects on gene expression. The Methods used for these experiments are not perfect, they raise a number of concerns of varying importance. Nevertheless, the multifaceted approach, and the fact that the results are largely consistent across experiments, is reassuring. In sum, the results as a whole are credible. They certainly need confirmation by other groups. Indeed, given their credibility and importance, they demand replication; they cannot be ignored.

It is good that the assessment of the postural asymmetry was confirmed by using alternative measurement techniques. However, it is unfortunate that the person doing the postural measurement was apparently not blinded to the rat type (i.e., Sham, RUBI, LUBI). Given the description of the measurement process (manually stretching the legs, etc.), it appears that there was room for unconscious R/L bias in the methods. I realize that blinding might have had practical problems (e.g., R or L post-surgical dressing or scarring on the head). Nevertheless, blinding could have been managed for at least a substantial subset of rats. (And it could presumably have been easily done for the serum-injection studies.) Furthermore, it would have been good to develop a hands-off method of stretching the legs, perhaps by attaching them to a bar perpendicular to the long axis of the rat and devising a simple mechanical method of ensuring bilaterally symmetrical stretch, with symmetrical grasp of the feet, symmetrical speed of stretch, etc. The lack of blinding and of hands-off stretch are major deficits. If the other experiments had not been conducted and yielded consistent results, these deficits would have made the results far less credible.

It would also have been preferable, in at least a subset of rats, to quantify EMG and force during the postural evaluation. The results show a marked postural asymmetry, but this is only in terms of limb displacement, not neural activity. While it is likely that the asymmetry reflects differences in neural drive to the limbs, it is conceivable that there was unconscious bias in the way the limbs were pulled back and released (or even to musculoskeletal differences caused by the rat lying or moving asymmetrically after the unilateral cerebral lesion). While the results do not include EMG during the asymmetry evaluation, this issue could be partially addressed by quantifying the number of spikes in the few seconds prior to the cutaneous stimulation test. This would evaluate the background EMG level, which presumably would be higher in the contralateral limb of the UBI rats (see Figures 2A and 2B) (and/or lower in the antagonist muscles). Lack of EMG asymmetry consistent with the postural asymmetry would be a very disturbing finding; it would suggest a non-neural origin for the postural asymmetry. It should be assessed.

I assume that the authors plan to continue experiments of the kind reported here. If that is the case, it is important, essentially imperative, that they markedly upgrade their methods for assessing the postural asymmetry. It is likely that many people will soon be looking at their studies, and groups with be trying to replicate them. The authors need to establish the best possible methods, both for their own credibility and for the future of the entire endeavor. Thus, the postural asymmetry should use an automated, hands-off device. The involvement of a mechanical engineer in its design and implementation is highly desirable. Also, the evaluation should be blinded, at least in substantial subsets of rats. And it should ideally include measurement of the right and left force and EMG associated with the stretch. In sum, relying simply on the amount of movement made by the limb after unblinded hand stretching is sloppy and inadequate. Blinded automated stretch of both legs together, with concurrent recording of right and left force and EMG at different lengths, is needed. The authors have shown that this research area is really important. They should now enable it to be done with the best possible, and thus most credible, methods.

In the methods (lines 600-604), they described how they verified that the spinal lesion was complete by: (1) visualizing the complete separation of the rostral and caudal sections of the spinal cord at the time that they made the incision; and (2) after termination of the experiment. The completeness of their spinal lesion is of the utmost importance for this manuscript to be believable. For (1), the lesion site tends to fill up with blood unless packed with Gelfoam or the like, and even if the bleeding is stopped, it can be difficult to be certain that the lesion is complete without some physical confirmation (e.g., passing a suture under the cord and then lifting it through the gap, or lifting one cut end and observing that the other cut end doesn't move). For (2), I was not sure if they just looked at it by eye or through a dissecting microscope, and/or did histology on sections and viewed it under a compound microscope. The latter approach is the only definitive way to say that there were no residual connections, but even this is not totally definitive--it is possible that there were descending connections present during the testing that were broken after the experiment has been completed; and thus the transection appeared to be complete in the post-experiment evaluation. Thus, the only way to guarantee that the transection is complete during the experiment is to show this before the experiment begins. It is likely that they completely transected the spinal cord, but they really haven't proved it. Given the potential importance of this paper, they should absolutely nail this issue down.

The analysis of the cutaneous stimulation-induced EMG responses appears to focus just on the number of post-stimulus spikes, without accounting for differences in the background activity. It would be nice to know if the increased response was due to a change in background activity (more motoneurons are closer to threshold), a change in afferent activation at the stimulation site (they would have needed to record the afferent volley in the dorsal root for this), or a change in processing of the afferent input. This is not a major issue.

The molecular biology raises several issues. First, heat maps are used to present the data rather than the CT values (or normalized expression values with associated error bars). This method tends to exaggerate very small differences in expression (i.e., the red to green transition is less that 2-fold and many genes show less than 1.5-fold change). It would be preferable to translate to differences in actual protein, at least for a few of the genes. This could allay the concern that the changes are not very big. The rationale for the gene selection is also not clear, at least to this reviewer. The reason these particular genes were chosen at all is a little confusing to me to begin with, but I might have missed something.

Another issue is the choice of control genes (actin and Gapdh). The authors do use a good tool for normalizing expression between samples (https://genorm.cmgg.be/). The approach is designed to minimize artifacts that crop up in QPCR when you use a limited pool of housekeeping genes to normalize sample to sample differences in expression. This tool would normally run more control genes (e.g., 6 or more for this many experimental genes) and then calculate a geometric mean of the group rather than a single gene normalization. However, this study used only two control genes, and compared them to over 20 experimental genes. Furthermore, it seems likely that the relative CT data would show that both control genes are expressed at far greater levels than many of the experimental genes. Thus, the signal used to normalize might be an order of magnitude greater than the results that are being normalized. Finally, the use of these particular control genes as house-keeping genes has been brought into question in recent years, since their expression has been shown to be altered by many neurophysiological challenges, including spinal cord injury ( https://www.ncbi.nlm.nih.gov/pmc/articles/PMC3614345/). The issue of appropriate QPCR controls for brain tissue is addressed in https://www.nature.com/articles/srep37116

At present, these two concerns limit the significance of the asymmetric gene expression results. They are probably fixable if the investigators add more controls to the QPCR study and provide the raw data with variance included for the RNA work.

*Reviewer #3:*

Lukoynov et al. show that unilateral brain injury (UBI) after spinal transection causes stereotypic asymmetrical posture and withdrawal reflexes. They further implicate that endocrine pathways play a vital role in this non-spinal lateralized response. This is a novel and very intriguing concept that is supported by ample evidence. The paper is a good fit for publication in *eLife*. My comments mainly concern the clarity of the presentation and several technical issues, mostly related to the electrophysiological data. None of my concerns invalidate the conclusions drawn by the authors.

1. My main concern is related to the clarity of the text. The article deals with unusual concepts and hence is difficult to follow. Some more guidance to the reader would be helpful. The concept that the left and right spinal cord are different (hormones, reflexes) should be stressed in the introduction and stressed in the parts of the results for which it is relevant (related to Figure 3 and 4). Often some results only become clear when looking up details in the methods. I highlighted more details in the minor comments.

2. The effect of UBI with spinal transection on the asymmetry has here only been shown up to 180 mins post UBI. Thus, no chronic effect has been shown. This should be explicitly stated. It is entirely possible that symmetry would reestablish in the chronic case. During acute spinalization many compensatory mechanisms will occur with which endochrine signaling could potentially interact with. Confirmation with chronic transection would be interesting and could be done in a followup (Research Advance) article.

3. All hormonal results seem to be only done with left UBI. Given the lateralization it would very interesting to see results from right UBI as well. I.e., confirm that and identify which neurohormones can induce the phenotype of right UBI. This is beyond the scope of the manuscript, but as a follow up would add evidence for the observed phenomena.

4. There are several potential issues with the withdrawal reflex data.

a. Stimulation thresholds and response amplitude strongly depend on the stimulation conditions (electrode positions, resistance, etc.) which are difficult to replicate on the contralateral side. Hultborn and Malmsten 1983 for example recorded the incoming afferent volley.

b. Stimulation intensities are not clearly reported (only in ranges) thus it is not clear if all responses are comparable. It seems to me that a certain stimulation condition was established on one side and then repeated on the other. This procedure has a high probability of error. Ideally, experimenters should have been blinded. This is a clear limitation and should be mentioned but, given the other results, it is likely that the conclusions drawn are valid.

c. There is no good reason to assume that C-fibers or nociception is involved in these responses. Stimulation intensities were around 2x the threshold and only one stimulation pulse was applied (5T and preceding subthreshold pulses are common) and responses were not validated to be similar to mechanical stimulation. Further, long lasting responses could also be explained with persistent inward currents. I recommend just referring to them as withdrawal reflexes.

d. Reflex results are reported as ipsi-/contralateral and were obtained from left and right UBI. HL-PA results were reported separately for left and right UBI and not together. This inconsistency is somewhat confusing for the reader. Furthermore, since asymmetry should be present in the sham condition (Hultborn and Malmsten 1983; Zhang et al. 2020), it would be preferable to also show the individual results of left and the right UBI.

*Reviewer #4:*

This study examines an intriguing phenomenon in which an aspect of the signs of unilateral lesion of the hind limb area of rat motor cortex (termed UBI, for unilateral brain injury) can be replicated by a presumed hypophysial factor in the serum. Importantly, serum taken from rats after receiving right UBI produces left (contralateral) hind leg flexion, whereas serum after left UBI produces right hind leg flexion. The effects of UBI are present in T3-4 full transection animals, implying descending motor pathways or intrinsic spinal circuits are not mediating the effect. The study seems to be motivated by the last author's earlier work that different systemically- or intrathecally-administered drugs can produce lateralized hind leg effects (e.g., left flexion produce by met-Enk; right flexion produced by Arg-vasopressin). Lesion of the hind limb area of motor cortex produces contralateral limb impairment, including hind limb flexion. As is often the case in motor impairment studies, the authors seem to be using the hindlimb flexion response as a proxy for the constellation of motor control impairments produced by the lesion.

This is a very carefully conducted study. I do not have any major criticisms about the procedures and analyses. The measurement of the extent of flexion is adequate, as is the asymmetry index. The EMG analysis provides some explanation for the unilateral flexion and, together with the supporting literature (i.e., early studies of Bakalkin lab; spinal asymmetry studies), provide strong evidence for a segmental/propriospinal locus for the effect triggered by a circulating molecule. I agree that currently there is no evidence to support a highly lateralized and muscle-specific projection from the hind leg area of motor cortex to the rostral/connected spinal sympathetic circuits. The laterality-specific sera experiment, a classical physiological demonstration, is clear. This leaves the endocrine-related explanation for the flexor phenomenon. My major concern rests in the functional and clinical significance of the contralateral hind limb flexor sign and whether, indeed, a "paradigm in neurology" has been successfully challenged.

1) Does hind limb asymmetry after UBI have a behavioral consequence other than hind limb asymmetry in the anesthetized state? The statement that this study questions "a paradigm in neurology" (Abstract) and that endocrine-based therapeutics might be helpful in ameliorating motor signs (Discussion) implies that the flexor asymmetry is part of the contralateral hemiplegia that is characteristic of a motor stroke. In my opinion, more experiments tackling whether they are studying a significant motor sign would need to be conducted. They show that there is an endocrine basis for the sign.

2) Is the asymmetry functional meaningful for limb control? There is no discussion of the effect of this asymmetry. Possibly the early pharmacology studies can address this; although the drugs do have complex behavioral effects, impacting more than the spinal cord. What is the manifestation of the flexor response in the awake animal?

3) What triggers the neuroendocrine response? I think it is necessary to identify this, at some level, to help calibrate the functional significance of the effect. Is the asymmetric hindlimb flexor response produced by any cortical lesion of similar size? Is there a neural basis or is that too, mediated by circulating molecules?

4) What is the cellular/biochemical target of the effect of hypophysectomy? The authors have clear neurochemical targets and demonstrate lateralized effects with drug action. Can this be leveraged histologically (in situ) to identify responding cell classes or biochemical changes in the pituitary?

5) What is the time course or persistence of the asymmetric flexor response after UBI. It seems that the focus was the very short-term. Although, fourteen days was shown in the supplementary material, but is this the same phenomenon as the short-term event (see next point)?

6) The asymmetric flexor response can occur at a very short time after the lesion. This is not like the classical hyperreflexia seen after cortical (or spinal) injuries, which can take weeks to develop. Is this short induction-time event a prodrome for hyperreflexia or a different process? The supplemental figure 1-1 shows effect strengthening over 3 hours and a persistence of 2 weeks. This makes me concerned that there may be changes at different joints or different processes. This needs to be clarified.

[Editors’ note: further revisions were suggested prior to acceptance, as described below.]

Thank you for choosing to send your work entitled "Endocrine signaling mediates asymmetric motor deficits after unilateral brain injury" for consideration at *eLife*. Your letter of appeal has been considered by a Senior Editor and Peggy Mason as Reviewing editor, and we are prepared to consider a revised submission with no guarantees of acceptance.

In addition to the changes and clarifications you have outlined in your appeal, we ask that you also address very specifically and thoroughly two questions:

Please expand very specifically and clearly on the possibility of connectivity and humoral signaling combining to explain the hindlimb assymetry as opposed to either mechanism alone.

What do the authors propose is the human correlate of postural asymmetry in the rodent? Is it hemiparalysis in toto or is it limited to gait and posture assymetries?

[Editors’ note: further revisions were suggested prior to acceptance, as described below.]

Thank you for choosing to send your work entitled "Endocrine signaling mediates asymmetric motor responses to unilateral brain injury" for consideration at *eLife*. Your article and your letter of appeal have been considered by Peggy Mason as Senior Editor, and we regret to inform you that we are upholding our original decision.

The title and first sentence of the abstract reveal a motivation for this study that is unwarranted. The title purports to demonstrate endocrine signaling in the asymmetric motor responses to unilateral brain injury. Then the first sentence makes clear that the asymmetric responses referred to are hemiplegia and hemiparesis. But in fact the authors go on to liken a one-sided flexion (we are not told which muscle acting at which joint is involved in this flexion) to the hemiparetic posture in humans that includes ankle inversion, plantar flexion, adduction at the hip and extension of the knee along with a decerebrate upper limb posture.

As stated in the initial review, the bar for upending a more than century-long neurological truth is high. This manuscript makes claims that surpass the evidence as nothing that the authors present contradicts the idea that the interruption of neural pathways is responsible for hemiplegia after motor cortex damage.

[Editors’ note: further revisions were suggested prior to acceptance, as described below.]

Thank you for submitting your article "Left-right side-specific endocrine signaling complements neural pathways to mediate asymmetric effects of brain injury" for consideration by *eLife*. Your article has been reviewed by 3 peer reviewers, including Peggy Mason as the Reviewing Editor and Reviewer #1, and the evaluation has been overseen by Christian Büchel as the Senior Editor. The following individual involved in review of your submission has agreed to reveal their identity: Simon M Danner (Reviewer #2).

The reviewers have discussed their reviews with one another, and the Reviewing Editor has drafted this to help you prepare a revised submission. We have also prepared public reviews of your work, which are designed to transform your unrefereed author manuscript into a publicly accessible refereed preprint (read more about this in the "Posting public reviews" section below).

*Reviewer #1 (Recommendations for the authors (required)):*

The authors look at a potential hormonal influence on postural symmetry of the hind limb and at flexion WD reflexes following lesion of the hindlimb somatomotor cortex on the left. They find that the flexion of the hindlimb contralateral to the cortical lesion persists even if the lesion is imposed after full spinal cord transection. This indirect evidence for a hormonal mechanism is supported by its dependence on an intact pituitary; its recapitulation with injection of serum from a lesioned animal into a transected animal; and the reversal of the effects by administration of antagonists of vasopressin or β-endorphin receptors.

Muscle relaxants are a broad pharmacological group and their antagonism of the flexion is weak evidence for any particular mechanism.

Either the authors are using language loosely or I am missing something. Eg on lines 98-99, "The HL-PA along with contralesional flexion…" What is the difference between HL-PA and the flexion? I understood them to be the same. Also see line 116.

No correlation is shown, no stats given although the authors state that a correlation exists between balance beam and gait on one hand and flexion on the other. Show the correlation or omit this statement.

What added info does Pa (probability of flexion) provide over the mm flexion metric shown? I can see none. And there are a large number of figures and supplemental figures; it would be lovely to cut down. Please either justify Pa's inclusion or omit it.

How is the difference – δ HL-PA – calculated?

In line 484, the authors state that animals were studied for up to 3 hours after brain injury. Yet Figure 1 Suppl 1 shows flexion measurements at 1, 7, and 14 days post lesion. Explain.

Two points in the response to reviewers are worth noting. First it is true that these results have never been replicated. It is also true that no one has published a failure to replicate. Given that 30 years have passed since this line of research began, the lack of replication may speak to a profound lack of interest or to an inability to replicate combined with the common tendency to not publish negative results. This observation is simply notable and more concerning than reassuring.

Second, there is no reflex analysis after UBI alone (without spinal transection) that I could find. The quoted lines and figure describe the hindlimb flexion analysis, NOT the withdrawal reflex. Thus the lack of a gold standard for the WD reflex changes remains a concern.

*Reviewer #2 (Recommendations for the authors (required)):*

The authors show that unilateral brain injury leads to an asymmetry in the spinal neural circuitry that is mediated by hormonal signaling. They show that asymmetric posture and reflex responses can be caused by unilateral brain injury in fully-spinalized animals. These effects could be replicated by transfusion of serum from animals with to animals without a brain injury. They further investigate and characterize the endochrine signaling involved. These are very surprising and exciting findings that (given independent replication) significantly add to our understanding of the mechanisms involved in the asymmetric effects of unilateral brain injury.

I have reviewed this paper before and it has since gone through several iterations with the editors. There were concerns about the methodology used for comparing the reflexes, blinding of the experimenters, relationship of postural asymmetry measures in rodents to the human, and how to best interpret the results. These concerns were of great importance because of the significance of the presented findings and their potential impact. The authors have addressed all my concerns (and I believe also those of raised by the other reviewers). I don't have any additional concerns and recommend the paper for publication.

*Reviewer #3 (Recommendations for the authors (required)):*

This manuscript reports remarkable results of very high scientific and possibly clinical importance. A fundamental and time-hallowed assumption in experimental and clinical neuroscience is that the lateralized deficits caused by a hemispheric lesion are due to pathological asymmetry in the activity of neural pathways that connect the brain and spinal cord. The results in this manuscript indicate that this is not the whole story; that vascular/humoral mechanisms also underlie the lateralized deficits. For basic and clinical neuroscientists, this is an essentially earthshaking finding, with huge implications.

The manuscript has been substantially improved by the revisions. We have no major problems with the current version. At the same time, we do think some additional revisions are desirable.

Lines 198-207: The authors seem to be saying that the afferent response to stimulation itself was different. Couldn't threshold be lower due to differences in presynaptic inhibition or intrinsic motoneuron properties?

Lines 238-242: While I don't think that the authors necessarily intended this, the statement could be interpreted as inferring causality from the results, which would not be appropriate. The results show that changes in gene expression correlate with ipsi/contra asymmetry. The text should be changed accordingly.

Lines 489-498: As indicated in the Discussion, further studies of the specific contributions of neural-pathway and vascular/humoral contributions to lateralized motor deficits after sensorimotor cortex lesions are certainly needed. In this context, Lines 489-498 are problematic. The statement "This strategy did not allow us to assess a role of these two mechanisms in the asymmetry formation in animals with intact spinal cord" is puzzling. In the future, why not examine UBI impact in hypophysectomized rats with intact spinal cords? Why not examine the effect of Arg-vasopressin, β-endorphin, opioid peptides on UBI impact in hypophysectomized rats with intact spinal cords (including differences for right vs. left UBI)? A future series of valuable studies would be possible with the same methods used very effectively in this paper. Such straightforward studies should be conducted first, and might reduce the need for genetic studies. Genetic manipulations introduce a host of potential complications in regard to interpretation (given their inevitable wider effects). In short, the Discussion should explicate the possible further extensions of the current well-established methods.

The present methods could also enable future explorations of the interactions of neural and vascular mechanisms underlying the laterality of the effects of unilateral cortical lesions. Their similarity in timing invite further questions about the extent of their mechanistic overlap, and about whether their effects are additive or even synergistic, or conversely, might saturate so that they add little to each other.

Finally, the Discussion might amplify consideration of the possible clinical implications of the findings in regard to inter-individual differences in stroke effects (particularly related to individual differences in functional lateralization), and in regard to possible novel therapeutic approaches.

This manuscript reports remarkable results of high scientific and possibly clinical importance. A fundamental and time-hallowed assumption in experimental and clinical neuroscience is that the lateralized deficits caused by a hemispheric lesion are due to pathological asymmetry in the activity of neural pathways that connect the brain and spinal cord. The results in this manuscript indicate that this is not the whole story; that vascular/humoral mechanisms also underlie the lateralized deficits. For basic and clinical neuroscientists, this is an essentially earthshaking finding, with huge implications.

The manuscript has been substantially improved by the revisions. We have no major problems with the current version. At the same time, we do think some additional revisions are desirable.

Lines 198-207: The authors seem to be saying that the afferent response to stimulation itself was different. Couldn't threshold be lower due to differences in presynaptic inhibition or intrinsic motoneuron properties?

Lines 238-242: While we don't think that the authors necessarily intended this, the statement could be interpreted as inferring causality from the results, which would not be appropriate. The results show that changes in gene expression correlate with ipsi/contra asymmetry. The text should be changed accordingly.

Lines 489-498: As indicated in the Discussion, further studies of the specific contributions of neural-pathway and vascular/humoral contributions to lateralized motor deficits after sensorimotor cortex lesions are certainly needed. In this context, Lines 489-498 are problematic. The statement "This strategy did not allow us to assess a role of these two mechanisms in the asymmetry formation in animals with intact spinal cord" is puzzling. In the future, why not examine UBI impact in hypophysectomized rats with intact spinal cords? Why not examine the effect of Arg-vasopressin, β-endorphin, opioid peptides on UBI impact in hypophysectomized rats with intact spinal cords (including differences for right vs. left UBI)? A future series of valuable studies would be possible with the same methods used very effectively in this paper. Such straightforward studies should be conducted first, and might reduce the need for genetic studies. Genetic manipulations introduce a host of potential complications in regard to interpretation (given their inevitable wider effects). In short, the Discussion should explicate the possible further extensions of the current well-established methods.

The present methods could also enable future explorations of the interactions of neural and vascular mechanisms underlying the laterality of the effects of unilateral cortical lesions. Their similarity in timing invite further questions about the extent of their mechanistic overlap, and about whether their effects are additive or even synergistic, or conversely, might saturate so that they add little to each other.

Finally, the Discussion might amplify consideration of the possible clinical implications of the findings in regard to inter-individual differences in stroke effects (particularly related to individual differences in functional lateralization), and in regard to possible novel therapeutic approaches.

[Editors’ note: further revisions were suggested prior to acceptance, as described below.]

Thank you for resubmitting your work entitled "Left-right side-specific endocrine signaling complements neural pathways to mediate asymmetric effects of brain injury" for further consideration by *eLife*. Your revised article has been evaluated by Christian Büchel (Senior Editor) and a Reviewing Editor.

The authors have responded to the exact critiques of the previous review. Here what we are looking for is a thorough re-examination of this manuscript using the spirit of the critiques and going above and beyond their specifics. We ask that the authors try this one more time:

Emblematic of the problems are the examples below. This is NOT an exhaustive list. Consequently the authors should comb through the entire manuscript with care and deliberate consideration.

– Intro sentence "Brain injury-induced sensorimotor deficits typically develop on

45 the contralateral side of the body. They include reduced voluntary control and muscle strength, 46 lack of dexterity, spasticity, asymmetric postural limb reflexes and abnormal posture."

"Reduced voluntary control" is euphemistic. A cut pyramidal tract yields voluntary paralysis. Reduced muscle strength is a poor way to put it. The muscle is only affected once atrophy occurs way down the time-line. Motor weakness would be a more accurate way to articulate the result.

– In the second paragraph of the Intro, there is talk of cerebellar lesions. Why? This manuscript is long (see more below on this) and complex enough already, without adding in completely ancillary points.

In the Intro it is stated that the strategy is to spinally transect and THEN damage somatomotor cortex, Figure 1 starts out with only damage to somatomotor cortex. Fine, the findings need to be anchored. But then "The extra-spinal mechanism of the HL-PA formation induced by brain lesion was tested in rats 131 that had complete transection of the spinal cord at the T2-3 level before the UBI was performed 132 (Figure 1F; Figure 1—figure supplement 3, Figure 1—figure supplement 4 and Figure 1—figure supplement 5 showing data of three replication experiments)" but Figure 1F is the wrong reference. It should be 1H.

This type of error should not be occurring on the nth submission of a manuscript.

There is even a typo in line 91 "effects of UBI on the on contra-ipsilesional …" This manuscript needs to be free of this sort of thing.

The manuscript is also extremely long, as in monograph-length. If this is necessary, fine. But this editor suspects that not all the supplementary figures are necessary. For example, you clearly performed calibration experiments on multiple methods for measuring hindlimb extension. There is no need to include such calibrations in the manuscript.

In sum, look beyond the problems stated in the review. Look at the manuscript afresh and tighten and clean it up. Use discipline: careful writing that uses words precisely and in deliberate and consistent ways.

[Editors’ note: further revisions were suggested prior to acceptance, as described below.]

Thank you for resubmitting your work entitled "Left-right side-specific endocrine signaling complements neural pathways to mediate asymmetric effects of brain injury" for further consideration by *eLife*. Your revised article has been evaluated by Christian Büchel (Senior Editor) and Peggy Mason as Reviewing Editor.

The manuscript has been improved but there are some remaining issues that need to be addressed, as outlined below:

I apologize for the delay in getting this decision back to you. This is a very interesting study and the authors have provided an excellent revision. Please address the following issues, send back and I will accept. That is, that I offer the bulk of my suggestions as strong suggestions but not requirements.

There are two exceptions to the suggestion over requirement and that is the first paragraph of the Results and Figure 1A-G. In these two areas you do two things: First you show that the time course of PA covers two weeks post UBI. Second you replicate the finding that PA persists after sc-tx is done on an animal with an existing UBI. Neither of these points is worthy of so much space (one paragraph, the bulk of one figure and 4 of 5 supplemental figures associated with that figure).

I ask that these results be telegraphed in one to two sentences because first, the rest of the manuscript deals with acute effects, 180 min and shorter; and second, the sc-tx after UBI experiments are not new. It would be fine to put a version of Figure 1 (without H-I) into the supplemental figures. To include time points up to 2 weeks gives the mistaken impression that is relevant to the paper. It is not.

Second, make clear in abstract and throughout results that effects are acute, 3 hr or less. Consider adding acute to the title as well.

---

## [Author Response]

[Editors’ note: The authors appealed the original decision. What follows is the authors’ response to the first round of review.]

There was much about this manuscript that intrigued the reviewers. However, as the endpoint is an overthrow of more than a century old dogma, the bar of proof is raised. Thus, as editor, while I am appreciative of the opportunity to examine this interesting work, it is my opinion that this study does not meet the elevated bar needed.The concerns are:The lack of blinding.

This might be a misunderstanding.

i) Most of hindlimb asymmetry experiments were blinded as stated in the manuscript (see please lines 697-699, 643-645, and 661-662). Specifically, brain surgeries were performed by a researcher (N.L. or L.C.) in a surgery room, whereas the asymmetry was measured by another researcher in a behavioral room (L.C. or N.L., respectively, in a reverse order) who was blind to the surgery type. The hypophysectomy / sham surgeries were performed in another institution and weight-matched codified animals were used as control, that ruled out any bias. The same is true for the blood serum and agonist / antagonist experiments; the samples were codified and therefore it was not possible to know whether serum was derived from UBI or sham-operated animals. For molecular analyzes, the surgeries were performed by a researcher (usually L.C.) while tissue dissection by other persons (N.L. or Gisela Maia – who due to only technical support was not included in the list of authors).

ii) Furthermore, the most critical experiments were repeated by two independent groups of investigators, in Lund and Porto, with no prior knowledge of the results of another group (please, see Figure 1F; Figure 1—figure supplement 2; Figure 1—figure supplement 3. Replication experiments 1 and 2).

iii) Processing and statistical analysis of the EMG and molecular data was performed after completion of the experiments. Thereby the eventual results could not affect the row data acquisition that otherwise might be biased (this is added to the corrected manuscript).

Concerns about whether the transections were complete. Given the organization of the rodent spinal cord, even the sparing of a reticulospinal pathway is in the ventral column could account for cortical control of motor function in the distal spinal cord in a rodent.

The transection procedure was performed with all controlling procedures typically implemented in numerous studies. Thus controls for complete transection included (a) controls with Spongostan that was pressed against the rostral and caudal stumps to separate the two ends 2-3 mm apart; and (b) examination after the transection using a microsurgery microscope and performed before the testing that showed complete separation of the two parts with no dura mater connecting the two ends, and with bony structure of anterior surface

of the vertebral canal that was clearly seen at the transection site.

The revised manuscript will be supplemented with further assessment of the completeness of the transection requested by reviewer 2.

On the other hand, the demonstration of the left-right specific humoral brain-to-spinal cord signaling does not require spinal cord transection. The compelling evidence for the endocrine signaling is a) that the blockage of this pathway by hypophysectomy

Concern over papers from 30 years ago, from the same lab and never replicated, that form a critical piece of this story.

Leaving aside the ethical question, we consider this statement as incorrect in four issues. First, there is no documented failure to replicate these findings, and there are no published attempts to repeat the experiments. We would appreciate receiving a

reference on this issue.

Second, effects of Arg-vasopressin were discovered in another laboratory, the I.P. Pavlov Department of Physiology, Institute of Experimental Medicine, St. Petersburg Klement'ev et al., 1986).

Third, the conceptual core emerged from the early findings, the lateralization of the opioid system in the animal and human CNS has recently received substantial molecular evidence from publications of our group of collaborators (J. Neurotrauma

2012, 29: 1785; Cerebral Cortex 2015, 25: 97; FASEB J. 2017, 2017 31: 1953; Brain Res. 2018, 1695: 78). Furthermore, the opioid lateralization hypothesis has received a direct evidence in the recent PET study from another laboratory (NeuroImage 2020, 217: 116922).

Fourth, conclusions in this manuscript are solely based on the results generated in the past 6 years and presented in the manuscript; the early findings have only laid basis for the hypothesis.

There is no information about the role of deafferentation. Is a background of deafferentation necessary?

These data (effects of UBI alone) are presented in Figure 1E, and Figure 1—figure

supplement 1B-D, and described in lines 69 – 74.

We also attach our paper recently accepted for publication (Zhang et al., 2020, BRAIN Communications) that provides evidence for spinal cord asymmetry and methodological background for the present study.

There is no gold standard (UBI alone) for the reflex analysis.

These data (effects of UBI alone in animals with intact spinal cord) are presented in Figure 1E-G, and Figure 1—figure supplement 1B-D, and described in lines 120 – 128.

Furthermore, effects of UBI alone on the hindlimb postural asymmetry and asymmetry of withdrawal reflexes are described in details in our recent paper (Zhang et al., 2020, *BRAIN Communications*).

Far more addressable but still worrisome is that the authors do not directly tackle the issue of the human correlate of postural asymmetry in the rodent.Reviewer #1:This is a study with the shocking message that several of the effects of asomatomotor lesion on contralateral movement are mediated by humoral substances rather than by the loss of synaptic input. The experiments are thorough, showing that postural asymmetry results even if a thoracic transection is performed prior to the somatomotor cortical lesion. This does not happen for L brain injury after hypophysectomy but the effects of L brain injury can be recapitulated with vasopressin or b-endorphin. I have a few substantive comments.The authors need to clearly state what they believe the postural asymmetry corresponds to in the human. Does this correspond to the hemiparetic stance, walk? Does it include the voluntary paralysis of the digits? I think it is easier to conceive of the former than the latter and when I first read the manuscript, I thought that was what the authors were postulating. However upon re-reading the introduction, it is less clear. The authors need to explicitly and clearly state their hypothesis in the introduction and support their interpretation on this point in the discussion..

Please see the response to the general comment 2. The animal model and the hypothesis are presented in the introduction and discussed in details in the discussion. We would like to emphasize that the focus of this phenomenological study is not on clinical correlates but on evidence for or against the side-specific humoral signaling, which pathophysiological role could be addressed addressed in future studies. Please also see Response to comment 1 of the fourth reviewer.

I would like to see a comparison of the reflexes after L-UBI alone or L-UBI following thoracic transection. There is an effect in the dual manipulation but whether that mimics the L-UBI effect is not tested.

Please see the response to the general comment 6. Requested data (effects of UBI alone in rats with intact spinal cord) are presented in Figure 1E-G, and Figure 1—figure supplement 1B-D, and described in lines 120 – 128. Furthermore, UBI effects on the hindlimb postural asymmetry and asymmetry of withdrawal reflexes developed in animals with intact spinal cord are described in details in our recent paper (Zhang et al., 2020).

Secondly, as remarkable as the asymmetry in response to systemic peptides is the hindlimb preference (over the forelimb). One possibility for this is that there is a different set of molecules for the L hindlimb, R hl, L fl and R fl. Crazy-seeming but so is this result so at this point, the reader feels reluctant to rule anything out. A second possibility is that the deafferentation produced by the transection (or UBI in non-experimental conditions) is a necessary prerequisite for the humoral factors to work their magic. Related to this, do the authors have any data on whether VP, b-end do anything in the absence of a proceeding transection? These possibilities should be discussed.

This is exciting idea to examine if Arg-vasopressin and opioid peptides may induce side-specific effects in the forelimbs, and, if so, whether the effects would be contra- or ipsilateral relative to those in hindlimbs.

In this paper we do not analyze effects of neurohormones on the forelimb asymmetry however it may be induced by the unilateral injury to the forelimb area of the cortex (unpublished data). Furthermore, the previous molecular study revealed the lateralized expression of opioid genes to the left, and asymmetric organization of their co-expression pattern in the cervical spinal cord (Kononenko et al., 2017) suggesting that opioid peptides may also induce side-specific changes in the forelimb posture and reflexes, and mediate effects of the unilateral brain lesions on the forelimbs.

In this and previous studies, postural asymmetry was induced by neurohormones only in animals with transected spinal cord. In intact animals the descending neural pathways may interfere with effects of neurohormones or mediate the compensatory processes.

At the moment, we do not have data on postural changes induced by opioid peptides and vasopressin in intact animals. However, an asymmetric organization and function of the opioid and vasopressin / oxytocin systems in animal and human CNS is possibly a general phenomenon that was demonstrated in a number of molecular, pharmacological and human imaging studies (see please Introduction and Discussion in the revised manuscript). For example, µ-opioid receptor and opioid peptides that induce euphoria and dysphoria are lateralized in the human brain (Watanabe et al., 2015; Kantonen et al., 2020), while κ-receptor mediates pain processing in the right but not left amygdala in the rats (Nation et al., 2018; Phelps at al., 2019).References and discussion on the issues have been added to Introduction and Discussion sections.

The hyper- part of the reflexes is not analyzed, only the asymmetry. [actually there is mention of a decrease in threshold for the flexor reflexes but this is in the sum up and not in the figure or in the quantified text.] Why is this?

The reflexes (the current thresholds and the spike numbers) were analyzed but due to variations among animals only data on the asymmetry coefficients that are less affected by the variability are presented in the paper.

Strict criteria for evaluation of asymmetric EMG responses were imposed and considered in Limitations and Materials and methods. Please, see response to point 4 of the third reviewer.

Data for UBI effects on the asymmetry index in the current threshold are shown in the result section and on Figure 2C,E. Thus, the threshold for the semitendinosus muscle in rats exposed to brain injury was lower 3.6-fold on the contra- vs. ipsilesional side (P = 0.015), and the asymmetry index for the threshold of the semitendinosus was decreased 4.0-fold after the UBI when compared to sham surgery (P = 0.04).

180-190 is written as though VP and b-endorphin are given together but the figure suggests they were given separately. Please make this clear. Did the authors try the two substances together? Same comments for the antagonists.

Thank you, text has been corrected. We did not treat animals with two agonists or two antagonists. Indeed, two agonists may produce additive effects, and this is interesting to evaluate.

Reviewer #2:This is an enormously interesting study. And, if its results are replicated, it is also an enormously important study. The belief that lateralized brain control of the somatic musculature is due entirely to lateralized neural pathways has been unquestioned truth for several thousand years. Other possibilities are seldom considered; when they are mentioned, they are quickly discounted (e.g., p. 568 in J Neurophysiol (1989) 61:563-576). This study indicates that this belief is not the whole story; that humoral factors also contribute to lateralized brain control. Given these fairly earthshaking results (with really major scientific and clinical implications), the credibility of the study is critically important. In this regard, it is fortunate and impressive that the authors have addressed the issue in multiple ways – the basic postural study, the effects of specific agents, the effects of serum from other lesioned animals, the effects of hypophysectomy, the effects on gene expression. The Methods used for these experiments are not perfect, they raise a number of concerns of varying importance. Nevertheless, the multifaceted approach, and the fact that the results are largely consistent across experiments, is reassuring. In sum, the results as a whole are credible. They certainly need confirmation by other groups. Indeed, given their credibility and importance, they demand replication; they cannot be ignored.It is good that the assessment of the postural asymmetry was confirmed by using alternative measurement techniques. However, it is unfortunate that the person doing the postural measurement was apparently not blinded to the rat type (i.e., Sham, RUBI, LUBI). Given the description of the measurement process (manually stretching the legs, etc.), it appears that there was room for unconscious R/L bias in the methods. I realize that blinding might have had practical problems (e.g., R or L post-surgical dressing or scarring on the head). Nevertheless, blinding could have been managed for at least a substantial subset of rats. (And it could presumably have been easily done for the serum-injection studies.)

Please, see the response to the general comment 3. Briefly, most of hindlimb asymmetry experiments were blinded as stated in the manuscript.

Furthermore, it would have been good to develop a hands-off method of stretching the legs, perhaps by attaching them to a bar perpendicular to the long axis of the rat and devising a simple mechanical method of ensuring bilaterally symmetrical stretch, with symmetrical grasp of the feet, symmetrical speed of stretch, etc. The lack of blinding and of hands-off stretch are major deficits. If the other experiments had not been conducted and yielded consistent results, these deficits would have made the results far less credible.

The hands-off method of stretching the legs has been developed (Figure 1—figure supplement 2) and applied for analysis of a subset of rats with transected spinal cord that were exposed to the unilateral brain injury (Figure 1—figure supplement 5).

It would also have been preferable, in at least a subset of rats, to quantify EMG and force during the postural evaluation. The results show a marked postural asymmetry, but this is only in terms of limb displacement, not neural activity. While it is likely that the asymmetry reflects differences in neural drive to the limbs, it is conceivable that there was unconscious bias in the way the limbs were pulled back and released (or even to musculoskeletal differences caused by the rat lying or moving asymmetrically after the unilateral cerebral lesion). While the results do not include EMG during the asymmetry evaluation, this issue could be partially addressed by quantifying the number of spikes in the few seconds prior to the cutaneous stimulation test. This would evaluate the background EMG level, which presumably would be higher in the contralateral limb of the UBI rats (see Figures 2A and 2B) (and/or lower in the antagonist muscles). Lack of EMG asymmetry consistent with the postural asymmetry would be a very disturbing finding; it would suggest a non-neural origin for the postural asymmetry. It should be assessed.I assume that the authors plan to continue experiments of the kind reported here. If that is the case, it is important, essentially imperative, that they markedly upgrade their methods for assessing the postural asymmetry. It is likely that many people will soon be looking at their studies, and groups with be trying to replicate them. The authors need to establish the best possible methods, both for their own credibility and for the future of the entire endeavor. Thus, the postural asymmetry should use an automated, hands-off device. The involvement of a mechanical engineer in its design and implementation is highly desirable. Also, the evaluation should be blinded, at least in substantial subsets of rats. And it should ideally include measurement of the right and left force and EMG associated with the stretch. In sum, relying simply on the amount of movement made by the limb after unblinded hand stretching is sloppy and inadequate. Blinded automated stretch of both legs together, with concurrent recording of right and left force and EMG at different lengths, is needed. The authors have shown that this research area is really important. They should now enable it to be done with the best possible, and thus most credible, methods.In the methods (lines 600-604), they described how they verified that the spinal lesion was complete by: (1) visualizing the complete separation of the rostral and caudal sections of the spinal cord at the time that they made the incision; and (2) after termination of the experiment. The completeness of their spinal lesion is of the utmost importance for this manuscript to be believable. For (1), the lesion site tends to fill up with blood unless packed with Gelfoam or the like, and even if the bleeding is stopped, it can be difficult to be certain that the lesion is complete without some physical confirmation (e.g., passing a suture under the cord and then lifting it through the gap, or lifting one cut end and observing that the other cut end doesn't move). For (2), I was not sure if they just looked at it by eye or through a dissecting microscope, and/or did histology on sections and viewed it under a compound microscope. The latter approach is the only definitive way to say that there were no residual connections, but even this is not totally definitive--it is possible that there were descending connections present during the testing that were broken after the experiment has been completed; and thus the transection appeared to be complete in the post-experiment evaluation. Thus, the only way to guarantee that the transection is complete during the experiment is to show this before the experiment begins. It is likely that they completely transected the spinal cord, but they really haven't proved it. Given the potential importance of this paper, they should absolutely nail this issue down.The analysis of the cutaneous stimulation-induced EMG responses appears to focus just on the number of post-stimulus spikes, without accounting for differences in the background activity. It would be nice to know if the increased response was due to a change in background activity (more motoneurons are closer to threshold), a change in afferent activation at the stimulation site (they would have needed to record the afferent volley in the dorsal root for this), or a change in processing of the afferent input. This is not a major issue.

We thank the reviewer for suggestions on how to address a mechanism of hindlimb postural asymmetry formation.

First, it should be emphasized that the asymmetry induced by the unilateral brain injury has a neurogenic origin because of pancuronium, a muscle relaxant abolished its formation (Zhang et al., BRAIN communications, 2020).

Second, examination of the background EMG that was recorded for 5 min before stimulation revealed virtually no spikes in sham rats and a few spike in rats with injury; no or only few of them were induced by stretching. This was because EMG was recorded when animals were under anesthesia, and within the “spinal chock” period after complete spinal cord transection that is characterized by the absence of most reflexes. EMG responses only may be evoked by strong sensory stimulation. At the same time, short periods of EMG activity, if any, at the initiation of limb stretching that trigger muscle contractions may contribute to formation of the hindlimb asymmetry observed in resting rats.

Third, the asymmetry with contralesional flexion may be caused by contraction of muscles that were not analyzed.

As described in Response to general comment 7, we have already analyzed i) postural asymmetry by the automated hands-off stretching device operated by a micromanipulator followed by photographic recording (Figure 1—figure supplement 2); as well as ii) the hindlimb stretching force that correlated with the postural asymmetry (Zhang et al., BRAIN communications, 2020).

In general terms, there are two phases in analysis of a phenomenon that are i) its discovery – the acquisition of primary evidence that is often accomplished with simple techniques as elegantly described by Hans Selye in his “in vivo” lectures; and ii) elaboration of the phenomenon, its mechanisms, biological role and clinical significance using advanced methods. Accordingly, this phenomenological manuscript presents the first evidence for the side-specific endocrine mechanism, and reserves the functional and mechanistic issues for further studies.

We agree that it is highly important to decipher afferent, central or efferent mechanisms of the asymmetry in electrophysiological experiments as proposed by the second reviewer; to identify a functional and pathophysiological role of the side-specific endocrine signaling in behavioral / functional experiments as proposed by the fourth reviewer; to reveal pathways from the injured brain area to the hypothalamic-pituitary system as suggested by the fourth reviewer; and to identify neurohormones selectively mediating effects of the left and right side injury by molecular techniques. However, all these are not trivial tasks and are beyond the scope of this first phenomenological study.

This is included in the Limitation section of Discussion.

The molecular biology raises several issues. First, heat maps are used to present the data rather than the CT values (or normalized expression values with associated error bars). This method tends to exaggerate very small differences in expression (i.e., the red to green transition is less that 2-fold and many genes show less than 1.5-fold change). It would be preferable to translate to differences in actual protein, at least for a few of the genes. This could allay the concern that the changes are not very big. The rationale for the gene selection is also not clear, at least to this reviewer. The reason these particular genes were chosen at all is a little confusing to me to begin with, but I might have missed something.Another issue is the choice of control genes (actin and Gapdh). The authors do use a good tool for normalizing expression between samples (https://genorm.cmgg.be/). The approach is designed to minimize artifacts that crop up in QPCR when you use a limited pool of housekeeping genes to normalize sample to sample differences in expression. This tool would normally run more control genes (e.g., 6 or more for this many experimental genes) and then calculate a geometric mean of the group rather than a single gene normalization. However, this study used only two control genes, and compared them to over 20 experimental genes. Furthermore, it seems likely that the relative CT data would show that both control genes are expressed at far greater levels than many of the experimental genes. Thus, the signal used to normalize might be an order of magnitude greater than the results that are being normalized. Finally, the use of these particular control genes as house-keeping genes has been brought into question in recent years, since their expression has been shown to be altered by many neurophysiological challenges, including spinal cord injury ( https://www.ncbi.nlm.nih.gov/pmc/articles/PMC3614345/). The issue of appropriate QPCR controls for brain tissue is addressed in https://www.nature.com/articles/srep37116At present, these two concerns limit the significance of the asymmetric gene expression results. They are probably fixable if the investigators add more controls to the QPCR study and provide the raw data with variance included for the RNA work.

Please see a response to the general comment 3.

Responses to specific points:

“First, heat maps are used to present the data ….”

We agree with the listed limitations of heatmap visualization and are sorry that the data were not described clearly that caused a misunderstanding. The manuscript has been corrected accordingly.

Heat maps were not used for presentation and statistical evaluation of differences in expression levels. Formal statistical analysis was applied to “heatmap-unscaled” expression levels and asymmetry indexes. The mRNA levels were shown as normalized values in log scale as boxplots with median and hinges (Figure 3—figure supplement 2A-F). The normalized data were also used for calculation of the asymmetry index presented as boxplots (Figure 3—figure supplement 2G-K). Due to subtle differences in both the mRNA levels and the asymmetry index between the animal groups (20-30%), that however were formally significant after adjustment for multiple comparisons, they were not considered as evidence for the UBI effects. At the same time, we note that “Subtle changes in gene expression can have important biological consequences in mammalian cells” (Michaels et al., *Nature Communications*: https://doi.org/10.1038/s41467-019-08777-y). We analyzed such differences to assess the direction of changes, either elevation or decrease in the expression levels (or in the asymmetry index) after brain injury. Our analysis was similar to the gene set enrichment analysis, where an importance of a pathway (or a gene set) is assessed using data for a sufficiently large number of genes with subtle individual changes in the expression. Heat maps were used to visualize the directions of changes in expression levels and the asymmetry index (Figure 3; Figure 3—figure supplement 3) for all analyzed genes in each animal, and for medians in groups of animals.

“It would be preferable to translate to differences in actual protein, at least for a few of the genes.”

We did not claim that differences in mRNA levels were significant and biologically relevant; therefore, there was no need to confirm these differences by analysis of respective proteins or peptides. Instead we identified robust changes in coordination of intra- and interregional gene expression patterns induced by the UBI (Figure 3; Figure 3—figure supplement 3). This transcriptional phenomenon could be hardly validated at the protein level due to nonlinear dependence between the mRNA and protein levels.

Nonetheless for characterization of molecular effects of the UBI on the opioid system that mediates UBI effects of postural asymmetry, we analyzed three opioid peptides using RIA, which is still a gold standard for quantitative analysis of peptides / proteins. The proenkephalin marker Met-enkephalin-Arg-Phe, and prodynorphin-derived dynorphin B and Leu-enkephalin-Arg were found to be substantially elevated in the left lumbar spinal cord of the rats with transected spinal cord exposed to the UBI (Figure 3E; Figure 3—figure supplement 4). In spite of recent technological advances in proteomics, there are substantial reservations for their application for quantification of subtle changes in the levels of proteins or peptides, and for analysis of low expressed proteins including numerous neuroplasticity-related proteins and neuropeptides. Western blotting is a semiquantitative method even in its best variants due to the problems with specificity and quantification (see e. g. our paper: Watanabe H et al., *Addict Biol. 2009*; 14: 294-297), while mass spectrometry should be further developed for quantitative protein analysis in multiple biological samples (lain Van Gool et al., *Expert Review of Proteomics*, 2020; 17, 257-273) including spinal cord tissue as described in our study (Sui et al. *J Proteome Res*. 2013; 12: 2245-2252).

“The rationale for the gene selection is also not clear, at least to this reviewer.”

There is no established view on how to categorize genes as neuroplasticity-related. Specifically, there are no lists of neuroplasticity-related genes consistent among the studies, and, consequently, there are no panels of neuroplasticity-related genes offered by biotech companies, providers of the platforms for mRNA analysis. Most companies (e. g. Thermofisher, Illumina, Nanostring) offer “Neurological” or “Neuropathology Research” panels that include neuroplasticity-related genes but are not designed for their targeted analysis.

Any selection is arbitrary and a set of selected genes could be different among the studies depending on the aims. At these circumstances, we selected genes as neuroplasticity related if they are considered as such in several major studies; the selection was justified for each gene by referring to these studies in Materials and methods, and was not biased. We do not claim that the gene set is comprehensive.

Importantly the main conclusion does not depend on a gene set analysed. The aim was not to identify neuroplasticity-related genes affected by the unilateral brain injury, but to analyse a gene transcription profile as a tool to assess whether the unilateral brain lesion produces asymmetric molecular changes in the lumbar spinal segments, that could provide a molecular evidence for the humoral pathway.

“Another issue is the choice of control genes.”

We are sorry that the geNorm procedure was not properly described. Additional information has been added and geNorm described in more details in Materials and methods in the re-submitted manuscript.

Thus, all procedures were conducted strictly in accordance with the established guidelines for the qRCR based analysis of gene expression, the minimum information for publication of quantitative real-time PCR experiments guidelines (MIQE) (Bustin et al. 2009; Taylor et al., 2019). Two reference genes *Actb* and *Gapdh* were selected out of ten candidate reference genes (*Actb*, *B2m*, *Gapdh*, *Gusb*, *Hprt*, *Pgk*, *Ppia*, *Rplpo13a*, *Tbp*, and *Tfrc*) using the geNorm program (https://genorm.cmgg.be/ and Vandesompele et al., 2002). The expression stability of candidate genes was computed for four sets of samples that were the left and right halves of the lumbar spinal cord obtained from the left-sided sham surgery group and the left-sided UBI group. All sample sets showed M value below 0.5 at the established limit with 1.5 as the maximum value. The V value was 0.12 that was lower than 0.15 imposed as the maximum established limit; thus two top reference genes (*Actb* and *Gapdh*) were sufficient for normalization.

The most stable genes are commonly highly expressed genes that have to be expressed in all or most cell types. According to the geNorm analysis, stability is prioritized over the expression level issue. However, this is not trivial question; housekeeping genes are characterized by high expression level whereas genes with low expression levels are generally cell type specific and adaptable, and their use as reference points may be compromised by differences in cell composition (the proportion of cells expressing and not expressing the genes) under transition between experimental conditions (for more details see our paper: Basov et al., 2017). Because the most stable reference genes were analyzed, and data were taken from the linear range of qPCR amplification, we could conclude that normalization was correct.

We agree with the reviewer that the expression of reference genes can differ among experimental conditions. For this reason, two reference genes with the stable expression levels were selected from ten candidate genes using geNorm before the main analysis (see, Materials and methods). Reference genes that demonstrated no difference in expression between the spinal sides and between the sham surgery and UBI groups were selected. The recommended study (https://www.nature.com/articles/srep37116) that compared a gene expression patterns among regions in the human brain, used the same reference genes, thus confirming that the geNorm was appropriate tool to evaluate the stability of reference genes.

“these two concerns limit the significance of the asymmetric gene expression results. They are probably fixable if the investigators add more controls to the QPCR study and provide the raw data with variance included for the RNA work”.

All requested controls have been performed and included in the revised manuscript, and normalized data with variance are provided as boxplots with medians and whiskers (Materials and methods; Figure 3E; Figure 3—figure supplement 2; and Figure 3—figure supplement 4).

Differences in the expression levels of individual genes and the asymmetry index between animal groups were significant but subtle and therefore these data were discarded as evidence for the UBI effects. However, subtle changes are biologically valid (Michaels et al., *Nature Communications*: https://doi.org/10.1038/s41467-019-08777-y) and may have a role in the revealed impressive changes in coordination of gene expression within and between the left and right spinal cord.

Reviewer #3:Lukoynov et al. show that unilateral brain injury (UBI) after spinal transection causes stereotypic asymmetrical posture and withdrawal reflexes. They further implicate that endocrine pathways play a vital role in this non-spinal lateralized response. This is a novel and very intriguing concept that is supported by ample evidence. The paper is a good fit for publication in eLife. My comments mainly concern the clarity of the presentation and several technical issues, mostly related to the electrophysiological data. None of my concerns invalidate the conclusions drawn by the authors.1. My main concern is related to the clarity of the text. The article deals with unusual concepts and hence is difficult to follow. Some more guidance to the reader would be helpful. The concept that the left and right spinal cord are different (hormones, reflexes) should be stressed in the introduction and stressed in the parts of the results for which it is relevant (related to Figure 3 and 4). Often some results only become clear when looking up details in the methods.

We are sorry about that; this was because the first variant of the manuscript was submitted as a brief report. The resubmitted manuscript has been supplemented with new background in Introduction, detailed presentation of the results and detailed discussion.

2. The effect of UBI with spinal transection on the asymmetry has here only been shown up to 180 mins post UBI. Thus, no chronic effect has been shown. This should be explicitly stated. It is entirely possible that symmetry would reestablish in the chronic case. During acute spinalization many compensatory mechanisms will occur with which endochrine signaling could potentially interact with. Confirmation with chronic transection would be interesting and could be done in a followup (Research Advance) article.

We thank the reviewer and have corrected the manuscript accordingly.

This is an excellent idea to examine i) whether the hindlimb postural asymmetry that is evident during first 3 h after the transection and brain lesion, could persist, or a symmetric patter could be reestablished in animals with chronically transected spinal cord; and ii) whether the side-specific neuroendocrine signaling would contribute to plastic changes in CPG and reflex transmission below the lesion, spasticity and the stepping recovery after the unilateral brain or spinal cord injury in the established models (Gossard et al., 2015; Tan et al., 2012).

3. All hormonal results seem to be only done with left UBI. Given the lateralization it would very interesting to see results from right UBI as well. I.e., confirm that and identify which neurohormones can induce the phenotype of right UBI. This is beyond the scope of the manuscript, but as a follow up would add evidence for the observed phenomena.

Thank you for this suggestion as well. We are working on it, and already have pharmacological evidence for a role of dynorphins in formation of the left hindlimb flexion after the right-side brain lesion in animals with intact spinal cord (manuscript in preparation).

4. There are several potential issues with the withdrawal reflex data.a. Stimulation thresholds and response amplitude strongly depend on the stimulation conditions (electrode positions, resistance, etc.) which are difficult to replicate on the contralateral side. Hultborn and Malmsten 1983 for example recorded the incoming afferent volley.b. Stimulation intensities are not clearly reported (only in ranges) thus it is not clear if all responses are comparable. It seems to me that a certain stimulation condition was established on one side and then repeated on the other. This procedure has a high probability of error. Ideally, experimenters should have been blinded. This is a clear limitation and should be mentioned but, given the other results, it is likely that the conclusions drawn are valid.

We are sorry that we did not clearly introduce the strict criteria that had been applied to ensure data comparability between two hindlimbs. They have been imposed for i) the experimental procedures including the symmetry of stimulation and recording conditions between the two sides; ii) application of electrodes with similar resistance for analysis of symmetric muscles; iii) selection of the reflex features for the analysis; and iv) statistical analysis. The core criteria were similar with those developed by Hultborn and Malmsten (Hultborn and Malmsten, 1983; Malmsten, 1983).

Specifically, in these experiments, stimulation and recording electrodes in pairs were positioned as symmetrically as possible. Stimulation electrodes were inserted into the center of receptive fields of the left and right muscles (Schouenborg et al., 1992; Weng and Schouenborg, 1996), and recording electrodes into the middle portion of the muscle belly. This was always done by an experienced investigator with knowledge in anatomy and physiology.

The same stimulation patterns were used for stimulation of pairs of digits to induce reflexes in symmetric muscles. The same threshold level was used for the left and right muscles. In general, the applied current was 2 to 3 times higher the recorded threshold for each pair of muscles at a given stimulation site. Data recorded with stimulation of more than one site (digits 2, 3, 4 and 5) were processed as replicates to decrease experimental error.

Only ipsilateral responses were recorded.

Only data for pairs of muscles of the same animal were included in the analysis. The sample size was sufficiently large (n = 11 in sham, and n = 18 in UBI groups) to ensure statistical power sufficient for analysis of responses typically registered in this model (Schouenborg et al., 1992; Weng and Schouenborg, 1996; Zhang et al., 2020).

To minimize inter-individual variations that may be caused by differences in physiological and experimental conditions including those in depth of anesthesia, and circulatory and respiratory state, the asymmetry index calculated for each animal but not absolute values of the reflex size including reflex amplitude, thresholds and the number of spikes, was analyzed. Comparison between the two sides using the asymmetry index was based on the assumption that one side was a reference for another side in each animal, and this approach largely diminished a contribution of the inter-individual variations (Hultborn and Malmsten, 1983; Malmsten, 1983).

Analysis of the asymmetry index allowed double assessment, first, within each the UBI and control (sham) groups that identified the asymmetric vs. symmetric pattern, and, second, between the animal groups that revealed UBI-induced changes in the asymmetry. Analysis of the asymmetry index in control group established whether the observed distribution was close to the expected symmetric pattern (the size of variations around the symmetry point was assessed), and, therefore, demonstrated the validity of the approach.

Because multiple responses were measured for the same animal, its two limbs, four muscles and stimulation conditions, and analyzed within the animal group and between the groups, we used mixed-effects models using Bayesian inference. To avoid bias in the acquisition of experiential observations that may be imposed by intermediate data analyses, the data processing and statistical analysis were performed after the completion of experiments. (The statistician (D.S.) who analyzed the data was not involved in execution of the experiments).

Only strong and significant UBI effects (they were from 2.8- to 54-fold) were considered as biologically relevant. At the same time, the background reflex asymmetry in the control group that had a smaller effect size (1.7-fold difference from the symmetry), in its magnitude and direction was in an agreement with previous results published by us (Zhang et al., 2020) and other groups (Hultborn and Malmsten, 1983; Malmsten, 1983).

In spite of strict criteria imposed and large size and high significance of UBI effects, the understanding of these effects due to complexity of the model requires more comprehensive mechanistic characterization of the UBI-induced reflex asymmetry e. g. by the complex assessment of system, cellular and synaptic potentials (for example see, Mahrous et al., 2019).

These issues have been added to Limitations and Materials and methods.

c. There is no good reason to assume that C-fibers or nociception is involved in these responses. Stimulation intensities were around 2x the threshold and only one stimulation pulse was applied (5T and preceding subtreshold pulses are common) and responses were not validated to be similar to mechanical stimulation. Further, long lasting responses could also be explained with persistent inward currents. I recommend just referring to them as withdrawal reflexes.

We thank the reviewer for the comment; the manuscript has been corrected accordingly. At the same time our previous studies demonstrated that while the withdrawal reflex could be evoked by innocuous stimulation in this experimental setting, these responses were weak compared to those evoked by noxious stimulation (Schouenborg et al., 1992; Weng and Schouenborg, 1998; Schouenborg, 2002; Zhang et al., 2020). This was added to section “EMG experiments” in Materials and methods.

d. Reflex results are reported as ipsi-/contralateral and were obtained from left and right UBI. HL-PA results were reported separately for left and right UBI and not together. This inconsistency is somewhat confusing for the reader. Furthermore, since asymmetry should be present in the sham condition (Hultborn and Malmsten 1983; Zhang et al. 2020), it would be preferable to also show the individual results of left and the right UBI.

Thank you for this comment. Effects of the left and right side brain injury were separately analyzed and the results were included in the manuscript (Figure 2—figure supplement 2; Figure 2—figure supplement 3). Virtually the same effects of brain injury on the spike number in rats with transected spinal cord were revealed, however not for all comparisons due to the lesser number of animals in the groups. In addition, the interosseous muscle was found to be asymmetric in the right UBI rats.

The withdrawal reflexes for all three muscles analyzed in the combined sham group consisting of the left and right sham animals displayed the left-right asymmetry in the number of spikes that was in the same direction (Left < Right) as described previously (Zhang et al., 2020; Hultborn and Malmsten, 1983). The asymmetry was substantial for the interosseous (median = -0.768, HPDCI = [-1.518, -0.013], fold difference = 1.7) and at the trend level for the extensor digitorum longus and semitendinosus due to the lesser number of observations compared to the preceding analysis (for each muscle, 7-10 *vs*. 14-15 in Zhang et al., 2020) (data are not included in the manuscript). The effect size for the background asymmetry (from 1.2- to 1.7-fold difference between the left and right hindlimbs) was much lower than the effects of the UBI (from 4- to 54-fold), and therefore the brain injury effects were readily identified on this background

Reviewer #4:This study examines an intriguing phenomenon in which an aspect of the signs of unilateral lesion of the hind limb area of rat motor cortex (termed UBI, for unilateral brain injury) can be replicated by a presumed hypophysial factor in the serum. Importantly, serum taken from rats after receiving right UBI produces left (contralateral) hind leg flexion, whereas serum after left UBI produces right hind leg flexion. The effects of UBI are present in T3-4 full transection animals, implying descending motor pathways or intrinsic spinal circuits are not mediating the effect. The study seems to be motivated by the last author's earlier work that different systemically- or intrathecally-administered drugs can produce lateralized hind leg effects (e.g., left flexion produce by met-Enk; right flexion produced by Arg-vasopressin). Lesion of the hind limb area of motor cortex produces contralateral limb impairment, including hind limb flexion. As is often the case in motor impairment studies, the authors seem to be using the hindlimb flexion response as a proxy for the constellation of motor control impairments produced by the lesion.This is a very carefully conducted study. I do not have any major criticisms about the procedures and analyses. The measurement of the extent of flexion is adequate, as is the asymmetry index. The EMG analysis provides some explanation for the unilateral flexion and, together with the supporting literature (i.e., early studies of Bakalkin lab; spinal asymmetry studies), provide strong evidence for a segmental/propriospinal locus for the effect triggered by a circulating molecule. I agree that currently there is no evidence to support a highly lateralized and muscle-specific projection from the hind leg area of motor cortex to the rostral/connected spinal sympathetic circuits. The laterality-specific sera experiment, a classical physiological demonstration, is clear. This leaves the endocrine-related explanation for the flexor phenomenon. My major concern rests in the functional and clinical significance of the contralateral hind limb flexor sign and whether, indeed, a "paradigm in neurology" has been successfully challenged.1) Does hind limb asymmetry after UBI have a behavioral consequence other than hind limb asymmetry in the anesthetized state? The statement that this study questions "a paradigm in neurology" (Abstract) and that endocrine-based therapeutics might be helpful in ameliorating motor signs (Discussion) implies that the flexor asymmetry is part of the contralateral hemiplegia that is characteristic of a motor stroke. In my opinion, more experiments tackling whether they are studying a significant motor sign would need to be conducted. They show that there is an endocrine basis for the sign.2) Is the asymmetry functional meaningful for limb control? There is no discussion of the effect of this asymmetry. Possibly the early pharmacology studies can address this; although the drugs do have complex behavioral effects, impacting more than the spinal cord. What is the manifestation of the flexor response in the awake animal?

We appreciate these comments; a functional role and clinical significance of the hindlimb postural asymmetry is also our concern.

The statement on “challenging” a “paradigm in neurology” has been omitted. We now hypothesize that the side-specific endocrine signaling may complement the descending neural pathways.

We should distinguish two consecutive phases in the analysis of a phenomenon: i) its discovery – the acquisition of primary evidence; and ii) its elaboration including identification of a functional role and pathophysiological significance that are very broad issues and generally require input from many studies and laboratories. Accordingly, this phenomenological manuscript presents the first evidence for the endocrine mechanism while leaving the second, functional and clinical issues for further studies.

The second phase requires development of novel pharmacological and genetic tools for selective inactivation of the endocrine mechanism in intact rats (e. g. antagonists that selectively block peripheral actions of the pituitary hormones when they are transported thought the blood; and transgenic animals in which this endocrine mechanism may be activated or inhibited). Development of both tool sets is not trivial.

Nevertheless, several findings discussed in the revised manuscript support the clinical relevance of the phenomenon. These findings suggest that the asymmetric changes in rats may recapitulate some clinical and pathophysiological features of the human upper motor neuron syndrome including the exaggerated asymmetric withdrawal reflexes and the asymmetric “hemiplegic posture”. Briefly, the asymmetrically exacerbated withdrawal reflexes, often leading to flexor spasms in patients, were developed in rats as shown in the study. Pathophysiological mechanisms of postural deficits are not well defined. Spastic dystonia, a tonic muscle overactivity of central origin that is developed without any trigger, may be a cause of “hemiplegic posture” with plantarﬂexion and inversion at the ankle, extension at the knee and associated ﬂexion at the elbow in patients (Lorentzen et al., 2018; Gracies, 2005; Sheean and McGuire, 2009), and the hindlimb postural asymmetry in rats as it was demonstrated in the previous study (Zhang et a., 2020).

Formation of the contralesional limb flexion correlates with impaired performance of this limb in the in the beam-walking and ladder rung tests (Figure 1). However, analysis of a role of the endocrine signaling vs. that of neural pathway in such impairment requires development of a novel model in which spinal mechanisms may be analyzed in isolation (not surgical) from the abnormal influences of the descending neural tracts in animals with intact spinal cord. This is not a simple task. We are not aware of any report that describes such a model.

To illustrate a potential of pharmacological amelioration of asymmetric motor deficits, in the revised manuscript we discuss the studies in which opioid antagonists reversed asymmetric neurological deficits secondary to unilateral cerebral ischemia (Baskin and Hosobuchi, 1981; Baskin et al., 1984; Baskin et al., 1994; Hosobuchi et al., 1982; Hans et al., 1992; Namba et al., 1986; Jabaily and Davis. 1984; Skarphedinsson et al., 1989; Gunnarsson et al., 1994), and reduce spasticity in patients with primary progressive multiple sclerosis (Gironi et al., 2008).

In summary, we agree that identification of a functional and pathophysiological role of the side-specific endocrine signaling in behavioral experiments, as proposed by the fourth reviewer, are highly important. We believe that deciphering the afferent, central or efferent mechanisms of the asymmetry in electrophysiological experiments, as proposed by the second reviewer, also merit consideration. Likewise, the identification of pathways from the injured brain area to the hypothalamic-pituitary system, which was brought up by the fourth reviewer, and the identification of neurohormones that selectively mediate the effects of the left and right side injury by molecular techniques, are highly important as well. However, we must point out to the reviewers that these are not at all trivial tasks and are beyond the scope of this first phenomenological study. We assure the reviewers that will keep these important points under consideration in further studies arising from this work.

3) What triggers the neuroendocrine response? I think it is necessary to identify this, at some level, to help calibrate the functional significance of the effect. Is the asymmetric hindlimb flexor response produced by any cortical lesion of similar size? Is there a neural basis or is that too, mediated by circulating molecules?

Hindlimb postural asymmetry is produced by the unilateral ablation injury of the cerebellum, the hindlimb area of sensorimotor cortex, large ablation in one of the hemispheres, and by the unilateral controlled cortical impact, a TBI model. The asymmetric effects were retained after complete spinal cord transection but a role of the endocrine mechanism has not been examined.

We included discussion of mechanisms of signaling from the injured cortex to the hypothalamic-pituitary system in the revised manuscript. Cortical projections to the hypothalamus that may mediate effects of focal brain injury on the secretion of pituitary hormones have been described, as have dysfunctions of the hypothalamic–pituitary system that cause changes in secretion of pituitary hormones after each stroke and TBI. However, neurobiological mechanisms underlying these effects have not been identified and anatomical pathways and neurotransmitter systems affected have not been revealed.

4) What is the cellular/biochemical target of the effect of hypophysectomy? The authors have clear neurochemical targets and demonstrate lateralized effects with drug action. Can this be leveraged histologically (in situ) to identify responding cell classes or biochemical changes in the pituitary?

We thank the reviewer for this question. This is important for understanding the side-specific endocrine mechanism and is discussed in the revised manuscript.

Arg-vasopressin and β-endorphin may mediate effects of the brain injury. Expression of both the vasopressin receptor V1B, which is activated by Arg-vasopressin and is blocked by its selective antagonist SSR-149415, and of proopiomelanocortin cleaved to β-endorphin occurs mostly in the pituitary gland, specifically in corticotrophs. A plausible scenario is that Arg-vasopressin released from neurohypophysis activates the V1B receptor in corticotrophs and stimulates secretion of β-endorphin that, by acting on the peripheral or central opioid receptors, induces the postural asymmetry. Analysis of gene expression in the pituitary demonstrated that the unilateral cortical injury robustly and selectively upregulated expression of the *Avpr1B* gene coding for the V1B vasopressin receptor. More detailed characterization of this effect is needed before its publication.

5) What is the time course or persistence of the asymmetric flexor response after UBI. It seems that the focus was the very short-term. Although, fourteen days was shown in the supplementary material, but is this the same phenomenon as the short-term event (see next point)?

Effects of the unilateral brain lesion that are mediated by the endocrine mechanism were evident at least for several hours after complete spinal cord transection. The approach was based on surgical dissociation of the neural and endocrine signaling, and did not allow for the analyses of these pathways separately in animals with intact spinal cord (see, please, Table 1).

We could not conclude whether hindlimb postural asymmetry recorded fourteen days after the injury in animals with intact spinal cord is mechanistically the same phenomenon. This issue is discussed in the limitation section and the conclusion part of the discussion.

6) The asymmetric flexor response can occur at a very short time after the lesion. This is not like the classical hyperreflexia seen after cortical (or spinal) injuries, which can take weeks to develop. Is this short induction-time event a prodrome for hyperreflexia or a different process? The supplemental figure 1-1 shows effect strengthening over 3 hours and a persistence of 2 weeks. This makes me concerned that there may be changes at different joints or different processes. This needs to be clarified.

Animals that showed the asymmetry during two weeks had either an intact spinal cord, or a spinal cord that was transected on the day of analysis. We could not rule out that the side-specific endocrine signaling may have a role in asymmetric motor deficits including exacerbated stretch reflex at later stages after the injury.

These issues are discussed in the revised manuscript.

[Editors’ note: what follows is the authors’ response to the second round of review.]

In addition to the changes and clarifications you have outlined in your appeal, we ask that you also address very specifically and thoroughly two questions:Please expand very specifically and clearly on the possibility of connectivity and humoral signaling combining to explain the hindlimb assymetry as opposed to either mechanism alone.

We have added such a discussion now to distinguish between the mechanisms. Briefly, the strategies applied to understand the contribution of neural *vs*. endocrine signaling to HL-PA development were, first, to selectively disable the former or the latter by surgical means; and, second, to assess a role of the latter by its activation in intact animals. The experimental procedures included spinal cord transection, hypophysectomy and administration of “pathological” serum that, respectively, disabled the neural and endocrine pathway, and turned on the endocrine mechanism (summarized in Table 1). These experiments provided evidence for the endocrine side-specific signaling from the injured brain in animals with transected spinal cord.

However, this approach did not allow for analyses of these pathways separately and in synergy in animals with intact spinal cord. Addressing these issues require the identification of the descending neural tracts that mediate formation of HL-PA, and development of pharmacological and genetic tools for selective inactivation of the neural pathways and the endocrine mechanism in intact rats. These are exciting tasks for future studies.

These issues are emphasized in the limitation section and the conclusion part of the discussion, and presented in Table 1 that has now been added for clarification.

What do the authors propose is the human correlate of postural asymmetry in the rodent? Is it hemiparalysis in toto or is it limited to gait and posture assymetries?

In this study the asymmetries of withdrawal reflexes and hindlimb posture were analyzed as a proxy or readouts for the effects polarized in the left-right direction with aim to investigate whether the endocrine system may convey the side-specific signals.

In the experimental part we did not focus on clinical correlates and mechanisms underlying posture and gait deficits. Nonetheless, the findings suggest that the pathological changes in rats may recapitulate several clinical and pathophysiological features of the human upper motor neuron syndrome including the exaggerated asymmetric withdrawal reflexes and the asymmetric “hemiplegic posture”. Briefly, the asymmetry in exacerbated reflexes, often leading to flexor spasms in patients, was similarly developed in rats as shown in the study. Pathophysiological mechanisms of postural deficits are not well defined. Spastic dystonia, a tonic muscle overactivity of central origin that is developed without any trigger, may contribute to “hemiplegic posture” with plantarﬂexion and inversion at the ankle, extension at the knee and associated ﬂexion at the elbow in human individuals (Lorentzen et al., 2018; Gracies, 2005; Sheean and McGuire, 2009), and to the hindlimb postural asymmetry in rats as we demonstrated in the previous study (Zhang et al., 2020).

The focus in this study was on the reflexes and postural asymmetry of the hindlimbs; the lesion of the hindlimb area of the sensorimotor cortex did not produce noticeable forelimb postural asymmetry (Zhang et al., 2020). The forelimb asymmetry may be induced by injury of the forelimb area (unpublished data). Undeniably a high rate of occurrence of the upper limb sensorimotor deficits in TBI and stroke patients requires the investigation of a role of the side-specific endocrine mechanism in these impairments.

All these issues are emphasized in the limitation section and throughout the manuscript.

[Editors’ note: what follows is the authors’ response to the second round of review.]

The title and first sentence of the abstract reveal a motivation for this study that is unwarranted. The title purports to demonstrate endocrine signaling in the asymmetric motor responses to unilateral brain injury. Then the first sentence makes clear that the asymmetric responses referred to are hemiplegia and hemiparesis. But in fact the authors go on to liken a one-sided flexion (we are not told which muscle acting at which joint is involved in this flexion) to the hemiparetic posture in humans that includes ankle inversion, plantar flexion, adduction at the hip and extension of the knee along with a decerebrate upper limb posture.As stated in the initial review, the bar for upending a more than century-long neurological truth is high. This manuscript makes claims that surpass the evidence as nothing that the authors present contradicts the idea that the interruption of neural pathways is responsible for hemiplegia after motor cortex damage.

[Editors’ note: what follows is the authors’ response to the second round of review.]

Reviewer #1 (Recommendations for the authors (required)):The authors look at a potential hormonal influence on postural symmetry of the hind limb and at flexion WD reflexes following lesion of the hindlimb somatomotor cortex on the left. They find that the flexion of the hindlimb contralateral to the cortical lesion persists even if the lesion is imposed after full spinal cord transection. This indirect evidence for a hormonal mechanism is supported by its dependence on an intact pituitary; its recapitulation with injection of serum from a lesioned animal into a transected animal; and the reversal of the effects by administration of antagonists of vasopressin or β-endorphin receptors.Muscle relaxants are a broad pharmacological group and their antagonism of the flexion is weak evidence for any particular mechanism.

Thank you. We did not use muscle relaxants in this study, and therefore now omit discussion of the effects of these substances from introduction and discussion (Lines 59-60).

Either the authors are using language loosely or I am missing something. Eg on lines 98-99, "The HL-PA along with contralesional flexion…" What is the difference between HL-PA and the flexion? I understood them to be the same.

This statement was perhaps inartfully made, and has been corrected (Line 95-96).

Also see line 116. No correlation is shown, no stats given although the authors state that a correlation exists between balance beam and gait on one hand and flexion on the other. Show the correlation or omit this statement.

Thank you for the remark. Two methods revealed that UBI effects were contralesional. Because a quantitative assessment of correlations was not necessary, we have removed the statement about correlation.

What added info does Pa (probability of flexion) provide over the mm flexion metric shown? I can see none. And there are a large number of figures and supplemental figures; it would be lovely to cut down. Please either justify Pa's inclusion or omit it.

We thank the reviewer for addressing this point and now justify the probability of postural asymmetry (P_A_) in the manuscript (Lines 112-114, and 1212-1221). The HL-PA was measured in mm with negative and positive HL-PA values that are assigned to rats with the left and right hindlimb flexion, respectively. This measure shows the flexion side and HL-PA value. However, it does not show the proportion of the animals with asymmetry in each group; we could not see whether all or a small fraction of animals display the asymmetry. Furthermore, its interpretation may not be straightforward for groups with the similar number of left or right flexion; in this case the HL-PA value would be about zero.

In contrast, the probability of postural asymmetry (P_A_) shows the proportion of animals exhibiting HL-PA at the imposed threshold (> 1 mm in this study). However, the P_A_ does not show flexion side and flexion size.

These two measures are obviously dependent; however, they are not redundant and for this reason, we believe that both are required for data presentation and characterization of the phenomenon.

How is the difference – δ HL-PA – calculated?

The contrast in HL-PA between the groups [designated as ∆HL-PA _(group 1 – group 2)_ on the figures] is a simple pairwise main effect (contrast), calculated as the difference between estimated marginal means of the groups computed by the R package emmeans (see https://cran.r-project.org/web/packages/emmeans/vignettes/comparisons.html#pairwise) given the fitted Bayesian model.

For example, in Figure 1—figure supplement 1, panel D, the ∆HL-PA_(UBI – Sh)_ is shown on the X-axis. This ∆HL-PA is computed as the mean HL-PA in the group "UBI rats, pre-treatment measurement" minus the mean HL-PA in the group "Sham surgery rats, pre-treatment measurement" that are shown in the upper two rows of panel C.

This explanation has been added to the statistical section (Lines 1415-1419).

In line 484, the authors state that animals were studied for up to 3 hours after brain injury. Yet Figure 1 Suppl 1 shows flexion measurements at 1, 7, and 14 days post lesion. Explain.

The asymmetry in rats with transected spinal cord was analyzed during the 3-hour time period after brain injury (Figure 1—figure supplement 3E-G). The asymmetry in rats with intact spinal cord that was studied for comparison (Figure 1—figure supplement 1C,D) was examined within 5 min, and at the 30, 60 and 180 min time points, and the 1, 7 and 14-day time points post lesion.

The sentence has been rewritten for clarity.

Two points in the response to reviewers are worth noting. First it is true that these results have never been replicated. It is also true that no one has published a failure to replicate. Given that 30 years have passed since this line of research began, the lack of replication may speak to a profound lack of interest or to an inability to replicate combined with the common tendency to not publish negative results. This observation is simply notable and more concerning than reassuring.

We respectfully disagree with the reviewer’s comments that our early findings “*have never been replicated*” for the following reasons.

First, the conceptual core emerged from the findings that i) the spinal cord asymmetry (de Kovel et al., 2017; Deliagina et al., 2000; Hultborn and Malmsten, 1983a, 1983b; Knebel et al., 2018; Kononenko et al., 2017; Malmsten, 1983; Nathan et al., 1990; Ocklenburg et al., 2017); and ii) the lateralization of the opioid / neuropeptide systems in the animal and human CNS; along with iii) the asymmetric and left / right side specific effects of neuropeptides in the brain and spinal cord all have received strong evidence in publications by our group of collaborators (J. Neurotrauma 2012, 29: 1785; Cerebral Cortex 2015, 25: 97; FASEB J. 2017, 2017 31: 1953; Brain Res. 2018, 1695: 78; Zhang et al., BRAIN Communications 2020; and Watanabe et al. BRAIN Communications 2020), and by others from animal experiments (Pilyavskii et al., Front Neurosci. 2013; Nation et al., 2018; Phelps at al., 2019) and human PET and pharmacological studies (Zink et al., 2011; Kantonen et al., 2020). These reports are discussed in the manuscript.

Second, the side-specific effects of Arg-vasopressin on formation of the right hindlimb flexion were reported by another laboratory, the I.P. Pavlov Department of Physiology, Institute of Experimental Medicine, St. Petersburg (Klement'ev et al., 1986). The vasopressin postural effects were replicated in the present study. The side-specific vasopressin actions are also supported by the observation that this neurohormone induces the left side response in the human brain (Zink et al., 2011).

We also would like to emphasize that there are many major and minor findings that received little or no attention at the time of their publication but were replicated or rediscovered many years afterwards, and thereupon attracting enormous interest. Such examples include “jumping genes”, chemiosmotic mechanism described by Mitchell, neural stem cells, and CRISPR by Mojica, and many others.

Second, there is no reflex analysis after UBI alone (without spinal transection) that I could find. The quoted lines and figure describe the hindlimb flexion analysis, NOT the withdrawal reflex. Thus the lack of a gold standard for the WD reflex changes remains a concern.

i) This is correct that “there is no reflex analysis after UBI alone (without spinal transection)”. In the present study, transection of the spinal cord was essential to disconnect the injured brain from the lumbar spinal cord for analysis of extra-spinal mechanism. Furthermore, the transection was necessary to examine spinal effects of the brain injury when descending influences that interfere with the reflexes are interrupted. Therefore, only sham injured rats with transected spinal cord may appropriately serve as a biological control. In this design, there is no need to analyze the withdrawal reflexes in animals with intact spinal cord as a control group, and such analysis was not the aim of the present study.

ii) This is also correct that “The quoted lines and figure describe the hindlimb flexion analysis, NOT the withdrawal reflex”. Analysis of postural asymmetry (Figures 1 and 4) did not include the withdrawal reflexes. The reflexes were a focus of the second part of the study (Figure 2; Figure 2—figure supplement 1; Figure 2—figure supplement 2), and were analyzed by the classic, well established method using EMG recording and electrical stimulation. The aim of the study did not include analysis of relationship between the behavioral postural asymmetry and withdrawal reflexes in animals with intact spinal cord, albeit they may be linked mechanistically.

The electrophysiological analysis of withdrawal reflexes used in the study, differs from behavioral analysis of these reflexes that is conducted in animals with intact spinal cord in pain studies (see, for example, our papers: Kononenko et al., 2018, Brain Res. 1695: 78-83; Ossipov et al., J Neurosci. 2007 27: 8226-37). In the present work, we focus on the withdrawal reflexes, while we plan to examine whether the unilateral brain injury could induce asymmetry in the hindlimb H-reflex and the spinal stretch reflex in future studies.

iii) We respectfully refer on the issue of “*the lack of a gold standard for the WD reflex changes*” to multiple previous studies including ours that provided well established standard for WRs analysis in animals with transected spinal cord (see, for example works by C.S. Sherrington a century ago, and by us and others: Schouenborg, 2002, Brain Res Rev, 40, 80-91; Clarke and Harris. Brain Res Rev. 2004, 46: 163-72; Schouenborg et al., Exp Brain Res, 1992, 90, 469-478; Weng and Schouenborg, 1996, J Physiol, 493, 253-265; Zhang et al., 2020, Brain Communications, 2(1), fcaa055). This level of quality of the WR analysis was attained in the present study.

Reviewer #3 (Recommendations for the authors (required)):This manuscript reports remarkable results of very high scientific and possibly clinical importance. A fundamental and time-hallowed assumption in experimental and clinical neuroscience is that the lateralized deficits caused by a hemispheric lesion are due to pathological asymmetry in the activity of neural pathways that connect the brain and spinal cord. The results in this manuscript indicate that this is not the whole story; that vascular/humoral mechanisms also underlie the lateralized deficits. For basic and clinical neuroscientists, this is an essentially earth-shaking finding, with huge implications.The manuscript has been substantially improved by the revisions. We have no major problems with the current version. At the same time, we do think some additional revisions are desirable.Lines 198-207: The authors seem to be saying that the afferent response to stimulation itself was different. Couldn't threshold be lower due to differences in presynaptic inhibition or intrinsic motoneuron properties?

We thank the reviewer for this remark. These sentences were not entirely clear and have been modified (Lines 213-215).

Lines 238-242: While I don't think that the authors necessarily intended this, the statement could be interpreted as inferring causality from the results, which would not be appropriate. The results show that changes in gene expression correlate with ipsi/contra asymmetry. The text should be changed accordingly.

We are afraid that inconsistencies in the line numbering among different outputs may hinder our understanding of this remark, and therefore we provide two responses.

First, if we understand it correctly, the reviewer points to the sentence “Changes in the expression asymmetry index were DUE TO decreased expression…”. To avoid it misinterpretation, this sentence has been modified (Lines 248-251).

Second, we would like to stress that Lines 248-251, and the rest of the manuscript do not infer any causality between gene expression and functional changes including those in postural asymmetry and reflexes. Lines 240-242 in the PDF of the previous variant of the manuscript introduce analysis of gene-gene correlations. We did not analyze correlations between molecular and functional parameters. Molecular changes may or may not be related to the asymmetric functional responses.

At the same time, we suggest that asymmetric molecular changes induced by the UBI in rats with transected spinal cord represent an independent from functional result molecular evidence for the extra-spinal mechanism. The molecular conclusion has also been rewritten (Lines 274-276). Furthermore, we have reviewed the manuscript to ascertain that we are not giving the impression that we are assigning causality, and made changes as appropriate.

Lines 489-498: As indicated in the Discussion, further studies of the specific contributions of neural-pathway and vascular/humoral contributions to lateralized motor deficits after sensorimotor cortex lesions are certainly needed. In this context, Lines 489-498 are problematic. The statement "This strategy did not allow us to assess a role of these two mechanisms in the asymmetry formation in animals with intact spinal cord" is puzzling. In the future, why not examine UBI impact in hypophysectomized rats with intact spinal cords? Why not examine the effect of Arg-vasopressin, β-endorphin, opioid peptides on UBI impact in hypophysectomized rats with intact spinal cords (including differences for right vs. left UBI)? A future series of valuable studies would be possible with the same methods used very effectively in this paper. Such straightforward studies should be conducted first, and might reduce the need for genetic studies. Genetic manipulations introduce a host of potential complications in regard to interpretation (given their inevitable wider effects). In short, the Discussion should explicate the possible further extensions of the current well-established methods.The present methods could also enable future explorations of the interactions of neural and vascular mechanisms underlying the laterality of the effects of unilateral cortical lesions. Their similarity in timing invite further questions about the extent of their mechanistic overlap, and about whether their effects are additive or even synergistic, or conversely, might saturate so that they add little to each other.

We are very thankful to the reviewer for these insightful and constructive comments, and have inserted these ideas in the revised discussion now (Lines 519-534).

Finally, the Discussion might amplify consideration of the possible clinical implications of the findings in regard to inter-individual differences in stroke effects (particularly related to individual differences in functional lateralization), and in regard to possible novel therapeutic approaches.

Discussion of possible clinical implications of the findings in regard to novel therapeutic approaches has been added to the manuscript (Lines 477-493).

[Editors’ note: what follows is the authors’ response to the second round of review.]

The authors have responded to the exact critiques of the previous review. Here what we are looking for is a thorough re-examination of this manuscript using the spirit of the critiques and going above and beyond their specifics. We ask that the authors try this one more time:Emblematic of the problems are the examples below. This is NOT an exhaustive list. Consequently the authors should comb through the entire manuscript with care and deliberate consideration.– Intro sentence "Brain injury-induced sensorimotor deficits typically develop on45 the contralateral side of the body. They include reduced voluntary control and muscle strength, 46 lack of dexterity, spasticity, asymmetric postural limb reflexes and abnormal posture.""Reduced voluntary control" is euphemistic. A cut pyramidal tract yields voluntary paralysis. Reduced muscle strength is a poor way to put it. The muscle is only affected once atrophy occurs way down the time line. Motor weakness would be a more accurate way to articulate the result.

We thank the editor for pointing this out. This part of introduction has been rewritten accordingly.

– In the second paragraph of the Intro, there is talk of cerebellar lesions. Why? This manuscript is long (see more below on this) and complex enough already, without adding in completely ancillary points.

Description of the effects of the cerebellar lesion on formation of hindlimb postural asymmetry and the respective references have been deleted.

In the Intro it is stated that the strategy is to spinally transect and THEN damage somatomotor cortex, Figure 1 starts out with only damage to somatomotor cortex. Fine, the findings need to be anchored. But then "The extra-spinal mechanism of the HL-PA formation induced by brain lesion was tested in rats 131 that had complete transection of the spinal cord at the T2-3 level before the UBI was performed 132 (Figure 1F; Figure 1—figure supplement 3, Figure 1—figure supplement 4 and Figure 1—figure supplement 5 showing data of three replication experiments)" but Figure 1F is the wrong reference. It should be 1H.This type of error should not be occurring on the nth submission of a manuscript.There is even a typo in line 91 "effects of UBI on the on contra-ipsilesional …" This manuscript needs to be free of this sort of thing.

Thank you. Typos have been corrected.

The manuscript is also extremely long, as in monograph-length. If this is necessary, fine. But this editor suspects that not all the supplementary figures are necessary. For example, you clearly performed calibration experiments on multiple methods for measuring hindlimb extension. There is no need to include such calibrations in the manuscript.

The manuscript has been substantially shortened. We have omitted two tables, Figure 1—figure supplement 3F-J, reduced the length of introduction and discussion, and reduced the number of references.

The revised manuscript does not contain “calibration experiments” but still contains results of experiments requested by the reviewers (e.g.; Figure 1—figure supplements 1, 2, 5; Figure 2—figure supplement 3). Other supplementary figures are essential to support the results presented in the main body of the manuscript. They show time course of asymmetry formation, results of EMG analysis of the EDL, Int and PL muscles that are not shown on Figure 2, and results of replication experiments with Wistar and Sprague Dawley rats that were used in different behavioral, molecular and electrophysiological studies.

At the same time our manuscript is well in range of the length of papers published by *eLife*. In five recent neuroscience papers taken as examples, the number of words in introduction, results and discussion varies from 4100 to 7300 while this number is 6231 in our manuscript. Method section generally consists of from 3200 to 9000 words and our manuscript has approximately 6000. There are only four figures with experimental data in the main body of our manuscript. The number of supplementary figures with multiple panels in our manuscript is similar to that of many recent *eLife* papers (e.g.; *eLife* 2021;10:e65228 and *eLife* 2021;10:e67262).

[Editors’ note: what follows is the authors’ response to the second round of review.]

I apologize for the delay in getting this decision back to you. This is a very interesting study and the authors have provided an excellent revision. Please address the following issues, send back and I will accept. That is, that I offer the bulk of my suggestions as strong suggestions but not requirements.There are two exceptions to the suggestion over requirement and that is the first paragraph of the Results and Figure 1A-G. In these two areas you do two things: First you show that the time course of PA covers two weeks post UBI. Second you replicate the finding that PA persists after sc-tx is done on an animal with an existing UBI. Neither of these points is worthy of so much space (one paragraph, the bulk of one figure and 4 of 5 supplemental figures associated with that figure).

We thank the editors for this suggestion. The first paragraph of the “Results” has been deleted while the second paragraph has been drastically modified according to the suggestions. It reads as follows:

“The hypothesis that a unilateral brain lesion may induce HL-PA through a pathway that bypasses the descending neural tracts was tested in rats that had complete transection of the spinal cord before the UBI was performed (Figure 1 A-E; Figure 1—figure supplements 1-5). The spinal cord was transected at the T2-T3 level and then the hindlimb representation area of the sensorimotor cortex was ablated (Figure 1A; Figure 1—figure supplements 1A). HL-PA was analyzed within 3 hours after the UBI by both the hands-on and hands-off methods of hindlimb stretching followed by photographic and / or visual recording of the asymmetry in animals under pentobarbital anesthesia (for details, see “Materials and methods” and Figure 1—figure supplement 2). HL-PA data are presented as the median values of HL-PA in mm (HL-PA size), and the probability to develop HL-PA (denoted as P_A_ on the figures) that depicts the proportion of rats with HL-PA above the 1-mm threshold. The analysis was generally blind to the observer (for details, see “Materials and methods”). Control experiments demonstrated that this injury produced HL-PA with contralesional hindlimb flexion within 3 hours after the UBI in rats with intact spinal cord (Figure 1F-H; Figure 1—figure supplement 1B-D), and contralesional hindlimb motor deficits in the beam-walking and ladder rung tests (Figure 1—figure supplement 1E,F).

Strikingly, in the rats with transected spinal cords the UBI also induced HL-PA (Figure 1C-E). The HL-PA developed within 3 hours after the brain injury. Its size and probability were much greater than in rats with sham surgery (Left UBI, n = 31; Right UBI, n = 15; sham surgery, n = 29). An unanticipated observation was that in rats with HL-PA, the hindlimb was flexed on the contralesional side. The left or right hindlimb flexion was induced by the right and left UBI, respectively (Figure 1D; Figure 1—figure supplement 3B,C; Figure 1—figure supplement 4B,C,F,G). Both Wistar rats (Figure 1D-E; Figure 1—figure supplements 3, 4F-I) and Sprague Dawley rats (Figure 1—figure supplements 4B-E, 5) that were used in further molecular and electrophysiological experiments, respectively, developed HL-PA with hindlimb flexion on the contralesional side. To ensure the completeness of the transection, a 3-4-mm spinal segment was excised at the T2-T3 level in a subset of rats (Figure 1—figure supplement 5). After the excision, the left-side UBI induced hindlimb postural asymmetry with the right limb flexion that replicated the other findings. The HL-PA size and probability, the time course of HL-PA development and formation of contralesional hindlimb flexion in rats with transected spinal cords that received UBI (Figure 1D,E; Figure 1—figure supplements 3,4) were similar to those of the UBI animals with intact spinal cords (Figure 1G,H; Figure 1—figure supplement 1C,D). We conclude that HL-PA formation in animals with transected spinal cord is mediated through a pathway that operates in parallel with the descending neural tracts and assures the development of contralesional flexion.”

Figure 1 has been also modified. This section and Figure 1 in the revised manuscript more focus on the key finding in rats with transected spinal cord that received brain injury (Figure 1C-E). Because comparison of the HL-PA between rats with transected and intact spinal cords was requested by the reviewers, data on the time course of HL-PA that was formed within three hours after the brain injury are presented for both rat groups (Figure 1 and Figure 1—figure supplement 1). These data are also new for the rats with intact spinal cord; the previous studies did not investigate the time period immediately after the UBI while analyzed HL-PA on day one and at later time points after the injury (Watanabe et al., 2021; Zhang et al., 2020).

There are five figure supplements to Figure 1. All of them present data requested by the reviewers and are necessary to support main findings. They describe and justify the brain injury model by showing the injury size (Figure 1—figure supplements 1A) and contralesional hindlimb responses (Figure 1—figure supplement 1), and demonstrate completeness of spinal cord transection (Figure 1—figure supplement 5), validity of HL-PA analysis by the hands-on and hands-off methods (Figure 1—figure supplement 2), and the effects of UBI on HL-PA formation in both Wistar rats (Figure 1H1D-JE; Figure 1—figure supplements 3, 4F-I) and Sprague Dawley rats (Figure 1—figure supplements 4B-E, 5) that were used in further molecular and electrophysiological experiments, respectively. They also allow the comparison of the time course of HL-PA formation in rats with intact (Figure 1—figure supplement 1C,D) and transected (Figure 1—figure supplements 3,4) spinal cord after the brain injury.

I ask that these results be telegraphed in one to two sentences because first, the rest of the manuscript deals with acute effects, 180 min and shorter; and second, the sc-tx after UBI experiments are not new. It would be fine to put a version of Figure 1 (without H-I) into the supplemental figures. To include time points up to 2 weeks gives the mistaken impression that is relevant to the paper. It is not.Second, make clear in abstract and throughout results that effects are acute, 3 hr or less. Consider adding acute to the title as well.

The title and abstract have been changed accordingly. They read as follows:

“Left-right side-specific endocrine signaling complements neural pathways to mediate acute asymmetric effects of brain injury“, and

“Brain injuries can interrupt descending neural pathways that convey motor commands from the cortex to spinal motoneurons. Here, we demonstrate that a unilateral injury of the hindlimb sensorimotor cortex of rats with completely transected thoracic spinal cord produces hindlimb postural asymmetry with contralateral flexion and asymmetric hindlimb withdrawal reflexes within three hours, as well as asymmetry in gene expression patterns in the lumbar spinal cord. The injury-induced postural effects were abolished by hypophysectomy and were mimicked by transfusion of serum from animals with brain injury. Administration of the pituitary neurohormones β-endorphin or Arg-vasopressin induced side-specific hindlimb responses in naïve animals, while antagonists of the opioid and vasopressin receptors blocked hindlimb postural asymmetry in rats with brain injury. Thus, in addition to the well-established involvement of motor pathways descending from the brain to spinal circuits, the side-specific humoral signaling may also add to postural and reflex asymmetries seen after brain injury“.

Furthermore, the 3 hr observation period is addressed repeatedly in “Results”, on figures and in figure legends, and it is emphasized in “Discussion”; please, see the first sentence in “Limitations”: “The side-specific endocrine signaling was revealed in anaesthetized animals with transected spinal cords that were studied up to 180 min post UBI”.